# Change Point Detection on A Separable Model for Dynamic Networks

**Yik Lun Kei**[*]                                                                 *ykei@ucsc.edu*
*Department of Statistics*
*University of California, Santa Cruz*

**Hangjian Li**[*]                                                         *hangjian.li@walmart.com*
*Walmart Global Tech*

**Yanzhen Chen**                                                           *imyanzhen@ust.hk*
*Department of Information Systems, Business Statistics and Operations Management*
*Hong Kong University of Science and Technology*

**Oscar Hernan Madrid Padilla**                                *oscar.madrid@stat.ucla.edu*
*Department of Statistics and Data Science*
*University of California, Los Angeles*

**Reviewed on OpenReview:** *https://openreview.net/forum?id=DSNJykzHF3*

## Abstract

This paper studies the unsupervised change point detection problem in time series of networks using the Separable Temporal Exponential-family Random Graph Model (STERGM). Inherently, dynamic network patterns are complex due to dyadic and temporal dependence, and change points detection can identify the discrepancies in the underlying data generating processes to facilitate downstream analysis. In particular, the STERGM that utilizes network statistics and nodal attributes to represent the structural patterns is a flexible and parsimonious model to fit dynamic networks. We propose a new estimator derived from the Alternating Direction Method of Multipliers (ADMM) procedure and Group Fused Lasso (GFL) regularization to simultaneously detect multiple time points where the parameters of a time-heterogeneous STERGM have shifted. Experiments on both simulated and real data show good performance of the proposed framework, and an R package `CPDstergm` is developed to implement the method.

## 1 Introduction

Networks are often used to describe relational phenomena that are not limited merely to the attributes of individuals as tabular data. In an investigation of the transmission of COVID-19, Fritz et al. (2021) used networks to represent human mobility and forecast disease incidents. The analysis of physical connections, beyond the health status of individuals, permits policymakers to implement preventive measures effectively and allocate healthcare resources efficiently. Yet relations by nature progress in time, and dynamic relational phenomena are occasionally aggregated into a static network for analysis. To this end, temporal models for dynamic networks are in high demand to study the evolution of relational phenomena.

In recent decades, a plethora of temporal models has been proposed for dynamic networks analysis. Snijders (2001), Snijders (2005), and Snijders et al. (2010) developed a Stochastic Actor-Oriented Model, which is driven by the actor's perspective to make or withdraw ties to other actors in a network. Hoff et al. (2002),

---

[*]Equal Contribution

Sarkar & Moore (2005), Sewell & Chen (2015), and Sewell & Chen (2016) presented latent space models, by assuming the edges between actors are more likely when they are closer in the latent Euclidean space. Matias & Miele (2017), Ludkin et al. (2018), and Pensky (2019) investigated the dynamic Stochastic Block Model (SBM), and Jiang et al. (2020) developed an autoregressive SBM to characterize the evolution of communities. Kolar et al. (2010) focused on recovering the latent time-varying graph structures of Markov Random Fields from serial observations of nodal attributes. Furthermore, the Exponential-family Random Graph Model (ERGM) that uses local forces to shape global structures (Hunter et al., 2008b) is a promising model for networks with dependent ties. Hanneke et al. (2010) defined a Temporal ERGM (TERGM), by conditioning on previous networks in the network statistics of an ERGM. Desmarais & Cranmer (2012b) proposed a bootstrap approach to maximize the pseudo-likelihood of the TERGM and assess uncertainty. In general, network evolution concerns the rate at which edges form and dissolve. Demonstrated in Krivitsky & Handcock (2014), these two factors can be mutually interfering, making the dynamic models used in the literature difficult to interpret. Posing that the underlying reasons that result in dyad formation are different from those that result in dyad dissolution, Krivitsky & Handcock (2014) proposed a Separable Temporal ERGM (STERGM) to dissect the entanglement with two conditionally independent models.

In time series analysis, change point detection plays a central role in identifying the discrepancies in the underlying data generating processes over time. In reality, network evolution is usually time-heterogeneous. Without taking the structural changes across dynamic networks into consideration, learning from the time series may not be meaningful, by confounding the network effects before and after a change occurs. As relational phenomena are studied in numerous domains, it is practical for researchers to first localize the change points, and then analyze the network effects within stable segments, rather than overlooking the time points where the network patterns have substantially changed.

There has also been an increasing interest in studying the unsupervised change point detection problem for dynamic networks. Wang et al. (2013) focused on the Stochastic Block Model time series, and Wang et al. (2021) studied a sequence of inhomogeneous Bernoulli networks. Larroca et al. (2021), Marenco et al. (2022), and Madrid Padilla et al. (2022) considered a sequence of Random Dot Product Graphs that are both dyadic and temporal dependent. Methodologically, Chen & Zhang (2015) and Chu & Chen (2019) developed a graph-based approach to delineate the distributional differences before and after a change point, and Chen (2019) utilized the nearest neighbor information to detect the changes in an online framework. Zhao et al. (2019) proposed a two-step approach that consists of an initial graphon estimation followed by a screening algorithm, Song & Chen (2024) exploited the features in high dimensions via a kernel-based method, and Chen et al. (2020a) employed embedding methods to detect both anomalous graphs and anomalous vertices. Chen et al. (2024) and Athreya et al. (2024) developed a model for network time series based on a latent position process, using spectral estimates of the Euclidean mirror to detect first-order change points. Zhang et al. (2024) combined Variational Graph Auto-Encoder and Gaussian Mixture Model, and Kei et al. (2025) focused on graph representation learning for change point detection in dynamic networks. Moreover, Liu et al. (2018) introduced an eigenvector-based method to reveal the change and persistence in the gene communities for a developing brain. Bybee & Atchadé (2018) focused on a Gaussian Graphical Model to detect the change points in the covariance structure of the Standard and Poor's 500. Ondrus et al. (2021) proposed a factorized binary search method to understand brain connectivity from the functional Magnetic Resonance Imaging time series data.

Allowing for interpretable network statistics to determine the structural changes for the detection, we make the following contributions in the proposed framework:

- To detect multiple change points from a sequence of networks, we fit a time-heterogeneous STERGM while penalizing the sum of Euclidean norms of the sequential parameter differences. Essentially, we impose a Group Fused Lasso (GFL) regularization on the model parameters, smoothing out minor variation and highlighting significant structural changes. An Alternating Direction Method of Multipliers (ADMM) procedure is derived to solve the resulting optimization problem.

- We exploit the practicality of STERGM, which manages dyad formation and dissolution separately, to capture the structural changes in network evolution realistically. The flexibility of STERGM and the extensive selection of network statistics also boost the power of the proposed method. Moreover,

we demonstrate the capability of including nodal attributes to detect change points, and an R package `CPDstergm` is developed to implement the proposed method.

- We simulate dynamic networks to imitate realistic social interactions, and our method can achieve greater accuracy on the networks that are both dyadic and temporal dependent. Furthermore, we punctually detect the winter and spring vacations with the MIT cellphone data (Eagle & Pentland, 2006), and we detect three significant financial events during the 2008 worldwide economic crisis from the stock market data analyzed by James & Matteson (2015).

The rest of the paper is organized as follows. In Section 2, we review the STERGM for dynamic networks. In Section 3, we present the likelihood-based objective function with Group Fused Lasso regularization, and we derive an Alternating Direction Method of Multipliers to solve the optimization problem. In Section 4, we discuss change points localization after parameter learning, along with model selection. In Section 5, we implement our method on simulated and real data. In Section 6, we conclude our work with a discussion on the limitation and potential future developments.

## 2 STERGM Change Point Model

### 2.1 Pseudo-likelihood of ERGM

For a node set $N = \{1, 2, \cdots, n\}$, we use a network $\boldsymbol{y} \in \mathcal{Y} = \{0,1\}^{n \times n}$ to represent the potential relations for all pairs $(i, j) \in \mathbb{Y} = N \times N$. The network $\boldsymbol{y}$ has dyad $\boldsymbol{y}_{ij} \in \{0, 1\}$ to indicate the absence or presence of a relation between node $i$ and node $j$. The relations in a network can be either directed or undirected, where an undirected network has $\boldsymbol{y}_{ij} = \boldsymbol{y}_{ji}$ for all $(i, j) \in \mathbb{Y}$.

The probabilistic formulation of an Exponential-family Random Graph Model (ERGM) is

$$P(\boldsymbol{y}; \boldsymbol{\theta}) = \exp[\boldsymbol{\theta}^\top \boldsymbol{g}(\boldsymbol{y}) - \psi(\boldsymbol{\theta})] \tag{1}$$

where $\boldsymbol{g}(\boldsymbol{y})$, with $\boldsymbol{g} : \mathcal{Y} \to \mathbb{R}^p$, is a vector of network statistics; $\boldsymbol{\theta} \in \mathbb{R}^p$ is a vector of parameters; $\exp[\psi(\boldsymbol{\theta})] = \sum_{\boldsymbol{y} \in \mathcal{Y}} \exp[\boldsymbol{\theta}^\top \boldsymbol{g}(\boldsymbol{y})]$ is the normalizing constant. The network statistics $\boldsymbol{g}(\boldsymbol{y})$ may also depend on the nodal attributes $\boldsymbol{x}$. For notational simplicity, we omit the dependence of $\boldsymbol{g}(\boldsymbol{y})$ on $\boldsymbol{x}$.

With a surrogate as in Besag (1974); Strauss & Ikeda (1990); Robins et al. (2007); Van Duijn et al. (2009); Desmarais & Cranmer (2012b); Hummel et al. (2012), the pseudo-likelihood of ERGM, a product of likelihood for each dyad $\boldsymbol{y}_{ij}$ conditional on the rest of the network $\boldsymbol{y}_{-ij}$, is

$$PL(\boldsymbol{\theta}) = \prod_{(i,j) \in \mathbb{Y}} P(\boldsymbol{y}_{ij} | \boldsymbol{y}_{-ij}; \boldsymbol{\theta}) = \prod_{(i,j) \in \mathbb{Y}} \left[ h\big(\boldsymbol{\theta} \cdot \Delta \boldsymbol{g}_{ij}(\boldsymbol{y})\big) \right]^{\boldsymbol{y}_{ij}} \cdot \left[ 1 - h\big(\boldsymbol{\theta} \cdot \Delta \boldsymbol{g}_{ij}(\boldsymbol{y})\big) \right]^{1 - \boldsymbol{y}_{ij}} \tag{2}$$

where $h(x) = 1/(1 + \exp(-x)) \in (0, 1)$ is the sigmoid function with $P(\boldsymbol{y}_{ij} = 1 | \boldsymbol{y}_{-ij}; \boldsymbol{\theta}) = h(\boldsymbol{\theta} \cdot \Delta \boldsymbol{g}_{ij}(\boldsymbol{y}))$. The change statistics $\Delta \boldsymbol{g}_{ij}(\boldsymbol{y}) \in \mathbb{R}^p$ denote the change in $\boldsymbol{g}(\boldsymbol{y})$ when $\boldsymbol{y}_{ij}$ changes from 0 to 1, while rest of the network $\boldsymbol{y}_{-ij}$ remains the same. Then the logarithm of the pseudo-likelihood is given by

$$l(\boldsymbol{\theta})_{\text{ERGM}} = \sum_{(i,j) \in \mathbb{Y}} \left\{ \boldsymbol{y}_{ij}[\boldsymbol{\theta} \cdot \Delta \boldsymbol{g}_{ij}(\boldsymbol{y})] - \log\left\{1 + \exp[\boldsymbol{\theta} \cdot \Delta \boldsymbol{g}_{ij}(\boldsymbol{y})]\right\} \right\}$$

which is helpful in ERGM parameter estimation. Moreover, the Hessian matrix of $-l(\boldsymbol{\theta})_{\text{ERGM}}$ is

$$H(\boldsymbol{\theta}) = \sum_{(i,j) \in \mathbb{Y}} h\big(\boldsymbol{\theta} \cdot \Delta \boldsymbol{g}_{ij}(\boldsymbol{y})\big) \cdot \left[1 - h\big(\boldsymbol{\theta} \cdot \Delta \boldsymbol{g}_{ij}(\boldsymbol{y})\big)\right] \cdot \left[\Delta \boldsymbol{g}_{ij}(\boldsymbol{y}) \Delta \boldsymbol{g}_{ij}(\boldsymbol{y})^\top\right].$$

Since $h(\boldsymbol{\theta} \cdot \Delta \boldsymbol{g}_{ij}(\boldsymbol{y})) \in (0, 1)$ and $\Delta \boldsymbol{g}_{ij}(\boldsymbol{y}) \Delta \boldsymbol{g}_{ij}(\boldsymbol{y})^\top \in \mathbb{R}^{p \times p}$ is positive semi-definite, the Hessian $H(\boldsymbol{\theta}) = \nabla^2_{\boldsymbol{\theta}} - l(\boldsymbol{\theta})_{\text{ERGM}}$ is also positive semi-definite. In other words, the negative logarithm of the pseudo-likelihood is convex with respect to $\boldsymbol{\theta} \in \mathbb{R}^p$. Next, we introduce the Separable Temporal ERGM (STERGM) used in our change point detection model.

## 2.2 Pseudo-likelihood of STERGM

For network time series, network evolution concerns (1) incidence: how often new ties are formed, and (2) duration: how long old ties last since they were formed. Pointed out by Donnat & Holmes (2018), Goyal & De Gruttola (2020), and Jiang et al. (2020), modeling snapshots of networks often gives limited information about the transitions between consecutive networks. To address this concern, Krivitsky & Handcock (2014) designed two intermediate networks, formation network and dissolution network, to reflect the incidence and duration. In particular, the incidence can be measured by dyad formation, and the duration can be traced by dyad dissolution. Many applications on real-world data support the separable mechanism for dynamic networks (Broekel & Bednarz, 2018; Uppala & Handcock, 2020; Ando et al., 2025).

Let $\boldsymbol{y}^t \in \mathcal{Y}^t = \{0,1\}^{n \times n}$ be a network observed at a discrete time point $t$. The formation network $\boldsymbol{y}^{+,t} \in \mathcal{Y}^{+,t}$ is obtained by attaching the edges that formed at time $t$ to $\boldsymbol{y}^{t-1}$, and $\mathcal{Y}^{+,t} = \{\boldsymbol{y} \in \mathcal{Y}^t : \boldsymbol{y} \supseteq \boldsymbol{y}^{t-1}\}$. The dissolution network $\boldsymbol{y}^{-,t} \in \mathcal{Y}^{-,t}$ is obtained by deleting the edges that dissolved at time $t$ from $\boldsymbol{y}^{t-1}$, and $\mathcal{Y}^{-,t} = \{\boldsymbol{y} \in \mathcal{Y}^t : \boldsymbol{y} \subseteq \boldsymbol{y}^{t-1}\}$. We can also use the notation from Kei et al. (2023) to specify the respective formation and dissolution networks between time $t-1$ and time $t$ as

$$\boldsymbol{y}_{ij}^{+,t} = \max(\boldsymbol{y}_{ij}^{t-1}, \boldsymbol{y}_{ij}^t) \ \text{ and } \ \boldsymbol{y}_{ij}^{-,t} = \min(\boldsymbol{y}_{ij}^{t-1}, \boldsymbol{y}_{ij}^t)$$

for all $(i,j) \in \mathbb{Y}$. In summary, $\boldsymbol{y}^{+,t}$ and $\boldsymbol{y}^{-,t}$ incorporate the dependence on $\boldsymbol{y}^{t-1}$ through construction, and they can be considered as two latent networks recovered from both $\boldsymbol{y}^{t-1}$ and $\boldsymbol{y}^t$ to emphasize the network transition from time $t-1$ to time $t$.

Posing that the underlying factors that result in edge formation are different from those that result in edge dissolution, Krivitsky & Handcock (2014) proposed the Separable Temporal ERGM (STERGM) to dissect the evolution between consecutive networks. Assuming $\boldsymbol{y}^{+,t}$ is conditionally independent of $\boldsymbol{y}^{-,t}$ given $\boldsymbol{y}^{t-1}$, the time-heterogeneous STERGM for $\boldsymbol{y}^t$ conditional on $\boldsymbol{y}^{t-1}$ is

$$\prod_{t=2}^{T} P(\boldsymbol{y}^t | \boldsymbol{y}^{t-1}; \boldsymbol{\theta}^t) = \prod_{t=2}^{T} P(\boldsymbol{y}^{+,t} | \boldsymbol{y}^{t-1}; \boldsymbol{\theta}^{+,t}) \times P(\boldsymbol{y}^{-,t} | \boldsymbol{y}^{t-1}; \boldsymbol{\theta}^{-,t}) \tag{3}$$

with the respective formation and dissolution models:

$$P(\boldsymbol{y}^{+,t} | \boldsymbol{y}^{t-1}; \boldsymbol{\theta}^{+,t}) = \exp\left[\boldsymbol{\theta}^{+,t} \cdot \boldsymbol{g}^+(\boldsymbol{y}^{+,t}, \boldsymbol{y}^{t-1}) - \psi^+(\boldsymbol{\theta}^{+,t}, \boldsymbol{y}^{t-1})\right],$$
$$P(\boldsymbol{y}^{-,t} | \boldsymbol{y}^{t-1}; \boldsymbol{\theta}^{-,t}) = \exp\left[\boldsymbol{\theta}^{-,t} \cdot \boldsymbol{g}^-(\boldsymbol{y}^{-,t}, \boldsymbol{y}^{t-1}) - \psi^-(\boldsymbol{\theta}^{-,t}, \boldsymbol{y}^{t-1})\right].$$

The parameter $\boldsymbol{\theta}^t = (\boldsymbol{\theta}^{+,t}, \boldsymbol{\theta}^{-,t}) \in \mathbb{R}^p$ is a concatenation of $\boldsymbol{\theta}^{+,t} \in \mathbb{R}^{p_1}$ and $\boldsymbol{\theta}^{-,t} \in \mathbb{R}^{p_2}$ such that $p_1 + p_2 = p$. Notably, the normalizing constant in the formation model $\exp[\psi^+(\boldsymbol{\theta}^{+,t}, \boldsymbol{y}^{t-1})]$ is a sum over all possible networks in $\mathcal{Y}^{+,t}$, and the normalizing constant in the dissolution model $\exp[\psi^-(\boldsymbol{\theta}^{-,t}, \boldsymbol{y}^{t-1})]$ is a sum over all possible networks in $\mathcal{Y}^{-,t}$. Since measuring these normalizing constants is computationally intractable when the number of nodes $n$ is large (Hunter & Handcock, 2006), many parameter estimation methods exploit MCMC sampling (Geyer & Thompson, 1992; Krivitsky, 2017) or Bayesian inference (Caimo & Friel, 2011; Thiemichen et al., 2016) to circumvent the intractability of the normalizing constants.

However, for change point detection with a time-heterogeneous model, these parameter estimation methods remain computationally intensive, making them prohibitive for relatively long sequences of networks. Hence, extending from the pseudo-likelihood of ERGM in (2), the pseudo-likelihood of a time-heterogeneous STERGM in (3) is given by

$$\prod_{t=2}^{T} PL(\boldsymbol{y}^t | \boldsymbol{y}^{t-1}; \boldsymbol{\theta}^t) = \prod_{t=2}^{T} \prod_{(i,j) \in \mathbb{Y}} \left[h\big(\boldsymbol{\theta}^{+,t} \cdot \Delta \boldsymbol{g}_{ij}^+(\boldsymbol{y}^{+,t})\big)\right]^{\boldsymbol{y}_{ij}^{+,t}} \cdot \left[1 - h\big(\boldsymbol{\theta}^{+,t} \cdot \Delta \boldsymbol{g}_{ij}^+(\boldsymbol{y}^{+,t})\big)\right]^{1 - \boldsymbol{y}_{ij}^{+,t}} \times$$
$$\left[h\big(\boldsymbol{\theta}^{-,t} \cdot \Delta \boldsymbol{g}_{ij}^-(\boldsymbol{y}^{-,t})\big)\right]^{\boldsymbol{y}_{ij}^{-,t}} \cdot \left[1 - h\big(\boldsymbol{\theta}^{-,t} \cdot \Delta \boldsymbol{g}_{ij}^-(\boldsymbol{y}^{-,t})\big)\right]^{1 - \boldsymbol{y}_{ij}^{-,t}}$$

where $h(x) = 1/(1 + \exp(-x)) \in (0,1)$ is the sigmoid function. Since $\boldsymbol{y}^{+,t}$ and $\boldsymbol{y}^{-,t}$ inherit the dependence on $\boldsymbol{y}^{t-1}$ by construction, we use the implicit dynamic terms, $\boldsymbol{g}^+(\boldsymbol{y}^{+,t})$ with $\boldsymbol{g}^+ : \mathcal{Y}^{+,t} \to \mathbb{R}^{p_1}$ and $\boldsymbol{g}^-(\boldsymbol{y}^{-,t})$

with $\boldsymbol{g}^- : \mathcal{Y}^{-,t} \to \mathbb{R}^{p_2}$, as discussed in Krivitsky & Handcock (2014). The change statistics $\Delta\boldsymbol{g}_{ij}^+(\boldsymbol{y}^{+,t}) \in \mathbb{R}^{p_1}$ denote the change in $\boldsymbol{g}^+(\boldsymbol{y}^{+,t})$ when $\boldsymbol{y}_{ij}^{+,t}$ changes from 0 to 1, while rest of the $\boldsymbol{y}^{+,t}$ remains the same; the change statistics $\Delta\boldsymbol{g}_{ij}^-(\boldsymbol{y}^{-,t}) \in \mathbb{R}^{p_2}$ denote the change in $\boldsymbol{g}^-(\boldsymbol{y}^{-,t})$ when $\boldsymbol{y}_{ij}^{-,t}$ changes from 0 to 1, while rest of the $\boldsymbol{y}^{-,t}$ remains the same. Similar to (2), the pseudo-likelihood of STERGM is a product of conditional likelihood for each dyad in the respective formation and dissolution networks, given the rest of the networks and the previous network (Besag, 1975; Strauss & Ikeda, 1990; Cranmer & Desmarais, 2011; Schmid & Hunter, 2023). Moreover, the logarithm of the pseudo-likelihood of the time-heterogeneous STERGM is

$$l(\boldsymbol{\theta}) = \sum_{t=2}^{T} \sum_{(i,j)\in\mathbb{Y}} \left\{ \boldsymbol{y}_{ij}^{+,t}\left[\boldsymbol{\theta}^{+,t}\cdot\Delta\boldsymbol{g}_{ij}^+(\boldsymbol{y}^{+,t})\right] - \log\left\{1 + \exp\left[\boldsymbol{\theta}^{+,t}\cdot\Delta\boldsymbol{g}_{ij}^+(\boldsymbol{y}^{+,t})\right]\right\} + \right.$$
$$\left. \boldsymbol{y}_{ij}^{-,t}\left[\boldsymbol{\theta}^{-,t}\cdot\Delta\boldsymbol{g}_{ij}^-(\boldsymbol{y}^{-,t})\right] - \log\left\{1 + \exp\left[\boldsymbol{\theta}^{-,t}\cdot\Delta\boldsymbol{g}_{ij}^-(\boldsymbol{y}^{-,t})\right]\right\}\right\} \tag{4}$$

where $\boldsymbol{\theta} = (\boldsymbol{\theta}^2, \dots, \boldsymbol{\theta}^T)^\top \in \mathbb{R}^{\tau\times p}$ with $\tau = T - 1$.

Empirically, for change point detection, the pseudo-likelihood of STERGM is preferable to the true likelihood for the following three reasons. First, the dyadic dependency in $\boldsymbol{y}^{+,t}$ and $\boldsymbol{y}^{-,t}$ is mitigated by conditioning on the previous network $\boldsymbol{y}^{t-1}$ and by separately modeling through the formation and dissolution processes. Specifically, conditional on $\boldsymbol{y}_{ij}^{t-1} = 1$, the $\boldsymbol{y}_{ij}^{+,t} = \max(\boldsymbol{y}_{ij}^{t-1}, \boldsymbol{y}_{ij}^t)$ can only be 1; while conditional on $\boldsymbol{y}_{ij}^{t-1} = 0$, the $\boldsymbol{y}_{ij}^{-,t} = \min(\boldsymbol{y}_{ij}^{t-1}, \boldsymbol{y}_{ij}^t)$ can only be 0. This design explicitly restricts the states of dyads by partitioning network evolution into formation and dissolution processes, thereby reducing the dyadic dependence within $\boldsymbol{y}^{+,t}$ and $\boldsymbol{y}^{-,t}$. Hence, conditioning on the previous network which already captures the structural dependencies, the pseudo-likelihood of STERGM becomes a reasonable surrogate. Second, the primary objective in change point detection is to localize substantial structural changes over time, rather than to recover the coefficient estimates for network effect interpretation. In the former case, the pseudo-likelihood of STERGM remains adequate, as large parameter shifts can still be reliably identified, even if the estimates are subject to mild bias by using a common approximation to the true likelihood. Third, we adopt the logarithm of the pseudo-likelihood to particularly avoid using MCMC sampling or Bayesian inference, which is computationally challenging for the optimization problem defined in Section 3. Instead, the pseudo-likelihood of STERGM improves the scalability of the estimation procedure, and the sigmoid function that involves the pre-computed change statistics permits efficient calculation of the gradients and Hessians for iterative parameter updates. In summary, the pseudo-likelihood of STERGM provides both computational feasibility and effectiveness to facilitate change point detection in dynamic networks, an advantage that the true likelihood cannot offer at scale.

Now we consider the change points to be detected in terms of the parameters in STERGM. Let $\{B_k\}_{k=0}^{K+1} \subset \{2, \dots, T\}$ be a collection of ordered change points with $2 = B_0 < B_1 < \cdots < B_K < B_{K+1} = T$ such that

$$\boldsymbol{\theta}^{B_k} = \boldsymbol{\theta}^{B_k+1} = \cdots = \boldsymbol{\theta}^{B_{k+1}-1}, \quad k = 0, \dots, K,$$

$$\boldsymbol{\theta}^{B_k} \neq \boldsymbol{\theta}^{B_{k+1}}, \quad k = 0, \dots, K-1, \quad \text{and} \quad \boldsymbol{\theta}^{B_{K+1}} = \boldsymbol{\theta}^{B_K}.$$

Our goal is to recover the collection $\{B_k\}_{k=1}^{K}$ from a sequence of observed networks $\{\boldsymbol{y}^t\}_{t=1}^{T}$ where the number of change points $K$ is also unknown. Intuitively, the consecutive parameters $\boldsymbol{\theta}^t$ and $\boldsymbol{\theta}^{t+1}$ are similar when no change occurs, but one or more components in $\boldsymbol{\theta}^{B_{k+1}} \in \mathbb{R}^p$ can be different from $\boldsymbol{\theta}^{B_k} \in \mathbb{R}^p$ after a change happens. For this setting, we present our method in the next section.

# 3 STERGM with Group Fused Lasso

## 3.1 Optimization Problem

Inspired by Vert & Bleakley (2010) and Bleakley & Vert (2011), we propose the following estimator for our change point detection problem:

$$\hat{\boldsymbol{\theta}} = \arg\min_{\boldsymbol{\theta}} -l(\boldsymbol{\theta}) + \lambda \sum_{i=1}^{\tau-1} \frac{\|\boldsymbol{\theta}_{i+1,.} - \boldsymbol{\theta}_{i,.}\|_2}{\boldsymbol{d}_i} \tag{5}$$

where $l(\boldsymbol{\theta})$ is formulated by (4). The term $\lambda > 0$ is a tuning parameter for the Group Fused Lasso penalty, and the term $\boldsymbol{d} \in \mathbb{R}_+^{\tau-1}$ is a position dependent weight such that $\boldsymbol{d}_i = \sqrt{\tau/[i(\tau-i)]}$ for $i \in [1, \tau-1]$. Intuitively, the inverse of $\boldsymbol{d}_i$ assigns a greater weight at the time point that is far from the beginning and the end of a time span, as the end points are usually not of interest for change point detection.

The Group Fused Lasso penalty expressed as the sum of Euclidean norms encourages sparsity of the parameter differences, while enforcing multiple components in $\boldsymbol{\theta}_{i+1,j} - \boldsymbol{\theta}_{i,j}$ across $j = 1, \ldots, p$ to change at the same group $i$. This is a grouping effect that cannot be achieved with the $\ell_1$ penalty of the differences. Along with the user-specified network statistics in STERGM, the sequential parameter differences learned from the observed networks with (5) can reflect the magnitude of structural changes over time. By penalizing the sum of sequential differences between the STERGM parameters, the proposed framework focuses on capturing significant structural changes while smoothing out minor variations.

Figure 1 gives an overview of the proposed framework. The shaded circles on the top denote the sequence of observed networks as time passes from left to right. The dashed circles in the middle denote the sequences of formation networks $\boldsymbol{y}^{+,t}$ and dissolution networks $\boldsymbol{y}^{-,t}$ recovered from the observed networks. Note that each observed network is utilized multiple times to extract useful information that emphasizes the transition between consecutive time steps. We learn the parameters denoted by the dotted circles at the bottom, while monitoring the sequential parameter differences.

Sequence of $T$ observed networks

Figure 1: An illustration of change point model with STERGM.

To solve the problem in (5), we first introduce a slack variable $\boldsymbol{z} \in \mathbb{R}^{\tau \times p}$ and rewrite the original problem as a constrained optimization problem:

$$\hat{\boldsymbol{\theta}} = \arg\min_{\boldsymbol{\theta}} -l(\boldsymbol{\theta}) + \lambda \sum_{i=1}^{\tau-1} \frac{\|\boldsymbol{z}_{i+1,\cdot} - \boldsymbol{z}_{i,\cdot}\|_2}{\boldsymbol{d}_i} \tag{6}$$

$$\text{subject to } \boldsymbol{\theta} = \boldsymbol{z}.$$

Let $\boldsymbol{u} \in \mathbb{R}^{\tau \times p}$ be the scaled dual variable. The augmented Lagrangian can be expressed as

$$\mathcal{L}_\alpha(\boldsymbol{\theta}, \boldsymbol{z}, \boldsymbol{u}) = -l(\boldsymbol{\theta}) + \lambda \sum_{i=1}^{\tau-1} \frac{\|\boldsymbol{z}_{i+1,\cdot} - \boldsymbol{z}_{i,\cdot}\|_2}{\boldsymbol{d}_i} + \frac{\alpha}{2}\|\boldsymbol{\theta} - \boldsymbol{z} + \boldsymbol{u}\|_F^2 - \frac{\alpha}{2}\|\boldsymbol{u}\|_F^2 \tag{7}$$

where $\alpha \in \mathbb{R}_+$ is another penalty parameter for the augmentation term.

Fused Lasso regularization has been widely used for one-dimensional change point detection problems (Levy-leduc & Harchaoui, 2007; Rojas & Wahlberg, 2014; Lin et al., 2017). Following Bleakley & Vert (2011), we make the change of variables $(\boldsymbol{\gamma}, \boldsymbol{\beta}) \in \mathbb{R}^{1 \times p} \times \mathbb{R}^{(\tau-1) \times p}$ to formulate the augmented Lagrangian in (7) as a Group Lasso regression problem (Yuan & Lin, 2006; Friedman et al., 2010; Alaíz et al., 2013), where

$$\boldsymbol{\gamma} = \boldsymbol{z}_{1,\cdot} \quad \text{and} \quad \boldsymbol{\beta}_{i,\cdot} = \frac{\boldsymbol{z}_{i+1,\cdot} - \boldsymbol{z}_{i,\cdot}}{\boldsymbol{d}_i} \quad \forall i \in [1, \tau-1]. \tag{8}$$

Reversely, the matrix $\boldsymbol{z} \in \mathbb{R}^{\tau \times p}$ can also be collected by

$$\boldsymbol{z} = \boldsymbol{1}_{\tau,1}\boldsymbol{\gamma} + \boldsymbol{X}\boldsymbol{\beta}$$

where $\boldsymbol{X} \in \mathbb{R}^{\tau \times (\tau-1)}$ is a designed matrix with $\boldsymbol{X}_{i,j} = \boldsymbol{d}_j$ for $i > j$ and 0 otherwise. Plugging $\boldsymbol{\gamma}$ and $\boldsymbol{\beta}$ into (7), we have

$$\mathcal{L}_\alpha(\boldsymbol{\theta}, \boldsymbol{\gamma}, \boldsymbol{\beta}, \boldsymbol{u}) = -l(\boldsymbol{\theta}) + \lambda \sum_{i=1}^{\tau-1} \|\boldsymbol{\beta}_{i,\cdot}\|_2 + \frac{\alpha}{2}\|\boldsymbol{\theta} - \boldsymbol{1}_{\tau,1}\boldsymbol{\gamma} - \boldsymbol{X}\boldsymbol{\beta} + \boldsymbol{u}\|_F^2 - \frac{\alpha}{2}\|\boldsymbol{u}\|_F^2. \tag{9}$$

Thus we derive the following Alternating Direction Method of Multipliers (ADMM) procedure to solve (6):

$$\boldsymbol{\theta}^{(a+1)} = \arg\min_{\boldsymbol{\theta}} -l(\boldsymbol{\theta}) + \frac{\alpha}{2}\|\boldsymbol{\theta} - \boldsymbol{z}^{(a)} + \boldsymbol{u}^{(a)}\|_F^2, \tag{10}$$

$$\boldsymbol{\gamma}^{(a+1)}, \boldsymbol{\beta}^{(a+1)} = \arg\min_{\boldsymbol{\gamma}, \boldsymbol{\beta}} \lambda \sum_{i=1}^{\tau-1} \|\boldsymbol{\beta}_{i,\cdot}\|_2 + \frac{\alpha}{2}\|\boldsymbol{\theta}^{(a+1)} - \boldsymbol{1}_{\tau,1}\boldsymbol{\gamma} - \boldsymbol{X}\boldsymbol{\beta} + \boldsymbol{u}^{(a)}\|_F^2, \tag{11}$$

$$\boldsymbol{u}^{(a+1)} = \boldsymbol{\theta}^{(a+1)} - \boldsymbol{z}^{(a+1)} + \boldsymbol{u}^{(a)}, \tag{12}$$

where $a$ denotes the current ADMM iteration. Once the update (11) is completed within an ADMM iteration, we collect $\boldsymbol{z}^{(a+1)} = \boldsymbol{1}_{\tau,1}\boldsymbol{\gamma}^{(a+1)} + \boldsymbol{X}\boldsymbol{\beta}^{(a+1)}$ until the next decomposition of $\boldsymbol{z}$. We recursively implement the three updates until a convergence criterion is satisfied. By adapting the idea from Boyd et al. (2011), we have the following result for the proposed ADMM procedure:

**Proposition 1.** *Denote the respective primal and dual residuals at the $a$th ADMM iteration as*

$$r_{primal}^{(a)} = \sqrt{\frac{1}{\tau \times p} \sum_{i=1}^{\tau} \sum_{j=1}^{p} (\boldsymbol{\theta}_{ij}^{(a)} - \boldsymbol{z}_{ij}^{(a)})^2} \quad \text{and} \quad r_{dual}^{(a)} = \sqrt{\frac{1}{\tau \times p} \sum_{i=1}^{\tau} \sum_{j=1}^{p} (\boldsymbol{z}_{ij}^{(a)} - \boldsymbol{z}_{ij}^{(a-1)})^2}.$$

*Assume the updates (10) and (11) attain minimum at each ADMM iteration. Then the primal residual $r_{primal}^{(a)} \to 0$ and dual residual $r_{dual}^{(a)} \to 0$ as $a \to \infty$.*

The proof is provided in Appendix A. Next, we discuss the updates (10) and (11) in detail.

## 3.2 Updating $\theta$

In this section, we derive the Newton-Raphson method for learning $\boldsymbol{\theta}$ in the update (10). We choose to use the Newton-Raphson method because it is more efficient than gradient descent: the Newton-Raphson method utilizes second-order information to adaptively determine the step size, leading to quadratic convergence and more stable updates (Galántai, 2000). Specifically, to implement the Newton-Raphson method in a compact form, we vectorize $\boldsymbol{\theta} \in \mathbb{R}^{\tau \times p}$ as $\vec{\boldsymbol{\theta}} = \text{vec}_{\tau p}(\boldsymbol{\theta}) \in \mathbb{R}^{\tau p \times 1}$, and we construct the following matrices:

$$\boldsymbol{\Delta}^t = \begin{pmatrix} \boldsymbol{\Delta}^{+,t} & \\ & \boldsymbol{\Delta}^{-,t} \end{pmatrix} \quad \text{and} \quad \boldsymbol{H} = \begin{pmatrix} \boldsymbol{\Delta}^2 & & \\ & \ddots & \\ & & \boldsymbol{\Delta}^T \end{pmatrix}.$$

The matrices $\boldsymbol{\Delta}^{+,t} \in \mathbb{R}^{E \times p_1}$ and $\boldsymbol{\Delta}^{-,t} \in \mathbb{R}^{E \times p_2}$ abbreviate the respective change statistics $\Delta \boldsymbol{g}_{ij}^+(\boldsymbol{y}^{+,t})$ and $\Delta \boldsymbol{g}_{ij}^-(\boldsymbol{y}^{-,t})$ that are ordered by the dyads. The dimension of $\boldsymbol{\Delta}^t$ is thus $2E \times p$, where the quantity $E = n \times n$

is due to vectorization, and the double in the number of rows is due to the separability of STERGM. In practice, the matrix $\boldsymbol{H} \in \mathbb{R}^{2\tau E \times \tau p}$ that consists of the change statistics for $t = 2, \ldots, T$ is calculated before the implementation of ADMM.

For the Hessian matrix, we also need to calculate $\vec{\boldsymbol{\mu}} = h(\boldsymbol{H} \cdot \vec{\boldsymbol{\theta}}) \in \mathbb{R}^{2\tau E \times 1}$ where $h(x) = 1/(1 + \exp(-x))$ is the element-wise sigmoid function with $h'(x) = h(x)(1 - h(x))$. Furthermore, we construct the following matrices:

$$\boldsymbol{W}^t = \begin{pmatrix} \boldsymbol{W}^{+,t} & \\ & \boldsymbol{W}^{-,t} \end{pmatrix} \text{ and } \boldsymbol{W} = \begin{pmatrix} \boldsymbol{W}^2 & & \\ & \ddots & \\ & & \boldsymbol{W}^T \end{pmatrix}$$

where $\boldsymbol{W}^{+,t} = \operatorname{diag}(\boldsymbol{\mu}_{ij}^{+,t}(1 - \boldsymbol{\mu}_{ij}^{+,t})) \in (0, 1/4)^{E \times E}$ with $\boldsymbol{\mu}_{ij}^{+,t} = h(\boldsymbol{\theta}^{+,t} \cdot \Delta \boldsymbol{g}_{ij}^+(\boldsymbol{y}^{+,t})) \in (0, 1)$. The matrix $\boldsymbol{W}^{-,t} \in \mathbb{R}^{E \times E}$ is defined similarly except for notational difference.

The Newton-Raphson method to iteratively update the parameter $\vec{\boldsymbol{\theta}} \in \mathbb{R}^{\tau p \times 1}$ for the proposed framework is implemented as follows:

$$\vec{\boldsymbol{\theta}}_{c+1} = \vec{\boldsymbol{\theta}}_c - \left( \boldsymbol{H}^\top \boldsymbol{W} \boldsymbol{H} + \alpha \boldsymbol{I}_{\tau p} \right)^{-1} \cdot \left( -\boldsymbol{H}^\top (\vec{\boldsymbol{y}} - \vec{\boldsymbol{\mu}}) + \alpha(\vec{\boldsymbol{\theta}}_c - \vec{\boldsymbol{z}}^{(a)} + \vec{\boldsymbol{u}}^{(a)}) \right) \tag{13}$$

where $c$ denotes the current Newton-Raphson iteration. The derivations are provided in Appendix B. The diagonal matrix $\boldsymbol{W}$ with diagonal entries between 0 and 1, along with a quadratic form of the matrix $\boldsymbol{H}$, shows that the matrix $\boldsymbol{H}^\top \boldsymbol{W} \boldsymbol{H}$ is positive semi-definite. The identity matrix $\boldsymbol{I}_{\tau p}$ inherited from the augmentation term $\|\boldsymbol{\theta} - \boldsymbol{z} + \boldsymbol{u}\|_F^2$ in (10) ensures the Hessian matrix $\boldsymbol{H}^\top \boldsymbol{W} \boldsymbol{H} + \alpha \boldsymbol{I}_{\tau p}$ is not only invertible but also positive definite. Thus the objective function in (10) is strongly convex with respect to the parameter $\boldsymbol{\theta}$ and a unique global minimum is guaranteed to exist. Once the Newton-Raphson method is concluded within an ADMM iteration, we fold the updated vector $\vec{\boldsymbol{\theta}}$ back into a matrix as $\boldsymbol{\theta}^{(a+1)} = \operatorname{vec}_{\tau,p}^{-1}(\vec{\boldsymbol{\theta}}) \in \mathbb{R}^{\tau \times p}$ before implementing the update in (11), which is discussed next.

### 3.3 Updating $\gamma$ and $\beta$

In this section, we derive the update in (11), which is equivalent to solving a Group Lasso problem. We decompose the matrix $\boldsymbol{z}$ to work with $\boldsymbol{\gamma}$ and $\boldsymbol{\beta}$ instead, and the objective function is convex with respect to these parameters. With ADMM, the updates on $\boldsymbol{\gamma}$ and $\boldsymbol{\beta}$ do not require the network data and the change statistics, but the updates primarily rely on the $\boldsymbol{\theta}$ learned from the update (10).

By adapting the derivation from Vert & Bleakley (2010) and Bleakley & Vert (2011), the matrix $\boldsymbol{\beta} \in \mathbb{R}^{(\tau-1) \times p}$ can be updated in a block coordinate descent manner. Specifically, the block coordinate descent method to update $\boldsymbol{\beta}_{i,\cdot}$ for each block $i = 1, \ldots, \tau - 1$ is implemented as follows:

$$\boldsymbol{\beta}_{i,\cdot} \leftarrow \frac{1}{\alpha \boldsymbol{X}_{\cdot,i}^\top \boldsymbol{X}_{\cdot,i}} \left( 1 - \frac{\lambda}{\|\boldsymbol{s}_i\|_2} \right)_+ \boldsymbol{s}_i \tag{14}$$

where $(\cdot)_+ = \max(\cdot, 0)$ and

$$\boldsymbol{s}_i = \alpha \boldsymbol{X}_{\cdot,i}^\top \left( \boldsymbol{\theta}^{(a+1)} + \boldsymbol{u}^{(a)} - \mathbf{1}_{\tau,1} \boldsymbol{\gamma} - \boldsymbol{X}_{\cdot,-i} \boldsymbol{\beta}_{-i,\cdot} \right).$$

The derivations are provided in Appendix C, and the convergence of the procedure is monitored by the Karush-Kuhn-Tucker (KKT) conditions:

$$\lambda \frac{\boldsymbol{\beta}_{i,\cdot}}{\|\boldsymbol{\beta}_{i,\cdot}\|_2} - \alpha \boldsymbol{X}_{\cdot,i}^\top (\boldsymbol{\theta}^{(a+1)} + \boldsymbol{u}^{(a)} - \mathbf{1}_{\tau,1} \boldsymbol{\gamma} - \boldsymbol{X}\boldsymbol{\beta}) = \mathbf{0} \qquad \forall \boldsymbol{\beta}_{i,\cdot} \neq \mathbf{0},$$

$$\|-\alpha \boldsymbol{X}_{\cdot,i}^\top (\boldsymbol{\theta}^{(a+1)} + \boldsymbol{u}^{(a)} - \mathbf{1}_{\tau,1} \boldsymbol{\gamma} - \boldsymbol{X}\boldsymbol{\beta})\|_2 \leq \lambda \qquad \forall \boldsymbol{\beta}_{i,\cdot} = \mathbf{0}.$$

Subsequently, for any $\boldsymbol{\beta} \in \mathbb{R}^{(\tau-1) \times p}$, the minimum in $\boldsymbol{\gamma} \in \mathbb{R}^{1 \times p}$ is achieved at

$$\boldsymbol{\gamma} = (1/\tau) \mathbf{1}_{1,\tau} \cdot (\boldsymbol{\theta}^{(a+1)} + \boldsymbol{u}^{(a)} - \boldsymbol{X}\boldsymbol{\beta}).$$

Once the update (11) is concluded within an ADMM iteration, we collect $\boldsymbol{z} = \mathbf{1}_{\tau,1} \boldsymbol{\gamma} + \boldsymbol{X}\boldsymbol{\beta}$ and proceed to update the scaled dual variable $\boldsymbol{u} \in \mathbb{R}^{\tau \times p}$ with (12).

# 4 Model Selection and Change Point Localization

In this section, we discuss additional details for change point detection with STERGM. The proposed method uses network statistics including nodal attributes, and the estimator is consistent under certain assumptions. Moreover, we can use Bayesian information criterion to perform model selection, and we provide a data-driven threshold for change point localization.

## 4.1 Network Statistics and Nodal Attributes

As a probability distribution over dynamic networks, STERGM allows us to generate different networks that share similar structural patterns with the observed networks, by using a carefully designed MCMC sampling algorithm (Besag, 2001; Snijders, 2002; Krivitsky, 2017). Hence, in a dynamic network modeling problem with STERGM, network statistics are often chosen to signify the underlying process producing the observed networks or to capture important network effects interpreting for a research question.

In the change point detection problem with STERGM, network statistics are chosen to determine the types of structural changes that are searched for by the researchers. The R library `ergm` (Handcock et al., 2022) provides an extensive list of network statistics that boost the power of the proposed method. Since the underlying reasons that result in edge formation are usually different from those that result in edge dissolution, the choices of network statistics in the formation model can be different from those in the dissolution model. Moreover, we permit the inclusion of nodal attributes in network statistics, a capability that many change point detection methods for dynamic networks do not provide. For an in-depth discussion of network statistics in an ERGM framework, see Handcock et al. (2003), Hunter & Handcock (2006), Snijders et al. (2006), Hunter et al. (2008a), Morris et al. (2008), Robins et al. (2009), and Blackburn & Handcock (2022).

## 4.2 Error Bound under Structured Sparsity

For our change point detection framework with Group Fused Lasso regularization, we provide the following estimation error bounds by adapting the idea from Negahban et al. (2012).

**Proposition 2.** *Denote $\boldsymbol{\theta}^* \in \mathbb{R}^{\tau \times p}$ as the true parameter and suppose that $\|\boldsymbol{\theta}^*\|_\infty \leq M/2$ for some $M > 0$. Let $\hat{\boldsymbol{\theta}} \in \mathbb{R}^{\tau \times p}$ be the minimizer of the objective function in (5), subject to the constraint $\|\boldsymbol{\theta}\|_\infty \leq M/2$. Define the set of true change points as*

$$S = \left\{ i \in \{1, \dots, \tau - 1\} : \boldsymbol{\theta}^*_{i+1,.} \neq \boldsymbol{\theta}^*_{i,.} \right\}.$$

*Suppose the loss function $L(\boldsymbol{\theta}) := -l(\boldsymbol{\theta})$ satisfies the Restricted Strong Convexity condition:*

$$L(\boldsymbol{\theta}^* + \boldsymbol{\Delta}) \geq L(\boldsymbol{\theta}^*) + \langle \nabla L(\boldsymbol{\theta}^*), \boldsymbol{\Delta} \rangle + \frac{k}{2} \|\boldsymbol{\Delta}\|_F^2,$$

*for all perturbations $\boldsymbol{\Delta} \in \mathbb{R}^{\tau \times p}$ that satisfy the structured sparsity condition:*

$$\sum_{i \notin S} \frac{\|\boldsymbol{\Delta}_{i+1,.} - \boldsymbol{\Delta}_{i,.}\|_2}{\boldsymbol{d}_i} \leq \alpha \sum_{i \in S} \frac{\|\boldsymbol{\Delta}_{i+1,.} - \boldsymbol{\Delta}_{i,.}\|_2}{\boldsymbol{d}_i}. \tag{15}$$

*for some constants $k > 0$ and $\alpha > 0$. Also, assume that with probability at least $1 - \delta$, the restricted dual norm of the gradient satisfies*

$$\|\nabla L(\boldsymbol{\theta}^*)\|_* \leq \frac{\lambda}{2}$$

*where the restricted dual norm $\|\cdot\|_*$ is taken with respect to the Total Variation (TV) norm $\|\cdot\|_{TV}$ as*

$$\|\nabla L(\boldsymbol{\theta}^*)\|_* = \sup_{\boldsymbol{\Delta}:\|\boldsymbol{\Delta}\|_{TV} \leq 1, \|\boldsymbol{\Delta}\|_\infty \leq 1} \langle \nabla L(\boldsymbol{\theta}^*), \boldsymbol{\Delta} \rangle \quad and \quad \|\boldsymbol{\Delta}\|_{TV} = \sum_{i=1}^{\tau-1} \frac{\|\boldsymbol{\Delta}_{i+1,.} - \boldsymbol{\Delta}_{i,.}\|_2}{\boldsymbol{d}_i}.$$

*Then if the estimation error $\hat{\boldsymbol{\Delta}} := \hat{\boldsymbol{\theta}} - \boldsymbol{\theta}^* \in \mathbb{R}^{\tau \times p}$ also satisfies the structured sparsity condition in (15), it follows with probability at least $1 - \delta$ that the mean squared error is bounded as*

$$\frac{1}{\tau p}\|\hat{\boldsymbol{\theta}} - \boldsymbol{\theta}^*\|_F^2 \le \frac{1}{\tau p} \max \left\{ \left[ \frac{6\lambda}{k}(1+\alpha) \cdot \sqrt{\sum_{i \in S} \boldsymbol{d}_i^{-2}} \right]^2, \frac{2\lambda M}{k} \right\}.$$

The proof is provided in Appendix D. Specifically, the Restricted Strong Convexity assumption ensures that $L(\boldsymbol{\theta})$ exhibits sufficient curvature when the estimation error $\hat{\boldsymbol{\Delta}}$ has most of its total variation aligned with the true change points. The restricted dual norm condition on $\nabla L(\boldsymbol{\theta}^*)$ controls the influence of noise, preventing stochastic fluctuations from suggesting spurious jumps in the estimated parameters. Together, these conditions facilitate a bound that reflects the estimator's sensitivity to the signal structure and noise level. Moreover, when $\alpha, \kappa, M \asymp 1$, $\delta \to 0$, and

$$\frac{1}{\tau p} \left( \lambda^2 \sum_{i \in S} \boldsymbol{d}_i^{-2} + \lambda \right) \to 0,$$

we obtain that our estimator is consistent for estimating $\boldsymbol{\theta}^*$ in terms of the mean squared error:

$$\frac{1}{\tau p}\|\hat{\boldsymbol{\theta}} - \boldsymbol{\theta}^*\|_F^2 \to 0.$$

While previous works have developed error bounds for Total Variation and Fused Lasso estimators (Rojas & Wahlberg, 2014; Hütter & Rigollet, 2016; Lin et al., 2017), we focus on the Group Fused Lasso regularization applied to the parameters of STERGM for dynamic networks.

## 4.3 Model Selection

In practice, to determine the optimal set of change points over multiple STERGMs learned with different tuning parameter $\lambda$, we can use Bayesian information criterion (BIC) to perform model selection. Consider the STERGM with learned $\hat{\boldsymbol{\theta}}$ and fixed $\lambda$, we have

$$\text{BIC}(\hat{\boldsymbol{\theta}}, \lambda) = -2l(\hat{\boldsymbol{\theta}}) + \log(TN_{\text{net}}) \times p \times \text{Seg}(\hat{\boldsymbol{\theta}}, \lambda). \tag{16}$$

For a list of $\lambda$, we choose the set of change points obtained from the STERGM with the lowest BIC value.

Different from the number of nodes $n$, the network size $N_{\text{net}}$ is $\binom{n}{2}$ for an undirected network and $2 \times \binom{n}{2}$ for a directed network. In general, for a dyadic dependent network, the effective network size is often smaller than $N_{\text{net}}$ and it may be difficult to quantify the effective size (Hunter et al., 2008a). In a node clustering problem for a static network, Handcock et al. (2007) used the number of observed edges to quantify the effective network size. In this work, we use $N_{\text{net}}$ to consider a greater value as the network size, since the procedure is to select a model with the lowest BIC value. Furthermore, the term $\text{Seg}(\hat{\boldsymbol{\theta}}, \lambda)$ in (16) gives the number of segments between change points $\{\hat{B}_k\}_{k=0}^{\hat{K}+1}$ that are learned with a particular $\lambda$ value. In other words, $\text{Seg}(\hat{\boldsymbol{\theta}}, \lambda) = \hat{K} + 1$, where $\hat{K}$ is the number of detected change points.

## 4.4 Data-driven Threshold

Intuitively, the location of a change point is the time step where the parameter of STERGM at time $t$ differs from that at time $t - 1$. To this end, we can calculate the parameter difference between consecutive time points in $\hat{\boldsymbol{\theta}} \in \mathbb{R}^{\tau \times p}$ as

$$\Delta \hat{\boldsymbol{\theta}}_i = \|\hat{\boldsymbol{\theta}}_{i+1,\cdot} - \hat{\boldsymbol{\theta}}_{i,\cdot}\|_2 \quad \forall i \in [1, \tau - 1]$$

and declare a change point when a parameter difference is greater than a threshold.

Though researchers can choose an arbitrary threshold for $\Delta \hat{\boldsymbol{\theta}}$ based on the sensitivity of the detection, in this work we provide a data-driven threshold with the following procedures. First we standardize the parameter

differences $\Delta\hat{\boldsymbol{\theta}}$ as

$$\Delta\hat{\boldsymbol{\zeta}}_i = \frac{\Delta\hat{\boldsymbol{\theta}}_i - \text{median}(\Delta\hat{\boldsymbol{\theta}})}{\text{sd}(\Delta\hat{\boldsymbol{\theta}})} \quad \forall i \in [1, \tau - 1]. \tag{17}$$

The $\Delta\hat{\boldsymbol{\zeta}} \in \mathbb{R}^{\tau-1}$ in (17) can be considered as the change magnitude in networks over time. Then the threshold based on the parameters learned from the data is constructed as

$$\epsilon_{\text{thr}} = \text{mean}(\Delta\hat{\boldsymbol{\zeta}}) + \mathcal{Z}_{1-\alpha} \times \text{sd}(\Delta\hat{\boldsymbol{\zeta}}) \tag{18}$$

where $\mathcal{Z}_{1-\alpha}$ is the $(1-\alpha)\%$ quantile of the standard Normal distribution. We declare a change point $B$ when $\Delta\hat{\boldsymbol{\zeta}}_B > \epsilon_{\text{thr}}$. The data-driven threshold in (18) is intuitive, as the standardized parameter differences at the change points are greater than those values in between the change points, derived from the Group Fused Lasso penalty. When tracing in a plot over time, the values $\Delta\hat{\boldsymbol{\zeta}}$ can exhibit the magnitude of structural changes, in terms of the network statistics specified in the STERGM.

## 5 Simulated and Real Data Experiments

In this section, we evaluate the proposed method on simulated and real data. For real data where ground truth is unknown, we align the detected change points with real world events for interpretation. For simulated data where ground truth is known, we use the following metrics to compare the performance of the proposed and competitor methods. The first metric is the absolute error $|\hat{K} - K|$ where $\hat{K}$ and $K$ are the numbers of detected and true change points, respectively. The second metric is the one-sided Hausdorff distance:

$$d(\hat{\mathcal{C}}|\mathcal{C}) = \max_{c \in \mathcal{C}} \min_{\hat{c} \in \hat{\mathcal{C}}} |\hat{c} - c|,$$

where $\hat{\mathcal{C}}$ and $\mathcal{C}$ are the respective sets of detected and true change points. We also report the metric $d(\mathcal{C}|\hat{\mathcal{C}})$. When $\hat{\mathcal{C}} = \emptyset$, we define $d(\hat{\mathcal{C}}|\mathcal{C}) = \infty$ and $d(\mathcal{C}|\hat{\mathcal{C}}) = \infty$. The third metric described in van den Burg & Williams (2020) is the coverage of a partition $\mathcal{G}$ by another partition $\mathcal{G}'$, defined as

$$C(\mathcal{G}, \mathcal{G}') = \frac{1}{T} \sum_{\mathcal{A} \in \mathcal{G}} |\mathcal{A}| \cdot \max_{\mathcal{A}' \in \mathcal{G}'} \frac{|\mathcal{A} \cap \mathcal{A}'|}{|\mathcal{A} \cup \mathcal{A}'|}$$

with $\mathcal{A}, \mathcal{A}' \subseteq [1, T]$. The $\mathcal{G}$ and $\mathcal{G}'$ are collections of intervals between consecutive change points for the respective true and detected change points. These three metrics are chosen to reflect a progression in evaluation difficulty: the absolute error assesses the number of detected change points, the one-sided Hausdorff distance captures the largest deviation in change point location, and the coverage metric measures alignment over the full time span, requiring a comprehensive match between the true and detected segments.

### 5.1 Simulation Study

We simulate dynamic networks from three particular models to imitate realistic patterns. First, we use the Stochastic Block Model (SBM) to attain that participants with similar attributes tend to form communities, and we impose a time-dependent mechanism in the network generation process. Second, we simulate dynamic networks from STERGM, which separately takes into account how relations form and dissolve over time, as their underlying reasons are usually different. Third, we utilize the Random Dot Product Graph Model (RDPGM), where edge formation is driven by the similarity in latent positions between nodes, and we allow these latent positions to evolve over time.

For each specification, we let the time span $T = 100$ and the number of nodes $n = \{50, 100, 200\}$. The $K = 3$ true change points are located at $t = \{26, 51, 76\}$, and the $K + 1 = 4$ intervals in the partition $\mathcal{G}$ are $\mathcal{A}_1 = \{1, \ldots, 25\}$, $\mathcal{A}_2 = \{26, \ldots, 50\}$, $\mathcal{A}_3 = \{51, \ldots, 75\}$, and $\mathcal{A}_4 = \{76, \ldots, 100\}$. In each specification, we report the means and standard deviations over 15 Monte Carlo trials for different evaluation metrics. Specifically, to detect change points with the proposed method, we initialize the penalty parameter $\alpha = 10$, and we let the tuning parameter $\lambda = 10^b$ with $b \in \{0, 1, 2, 3, 4\}$. For each $\lambda$, we run $A = 200$ iterations of ADMM and the stopping criterion in (56) uses $\epsilon_{\text{tol}} = 10^{-7}$. Within each ADMM iteration, we run $C = 20$

iterations of the Newton-Raphson method, and $D = 20$ iterations for the Group Lasso update. The stopping criteria for the Newton-Raphson method is $\|\vec{\boldsymbol{\theta}}_{c+1} - \vec{\boldsymbol{\theta}}_c\|_2 < 10^{-3}$. To construct the data-driven threshold in (18), we use the 0.9 quantile of the standard Normal distribution. Additional details about the model specifications are provided in Appendix E. Throughout, the network statistics are calculated directly from the R library `ergm` (Handcock et al., 2022), and the formulations are provided in Appendix F.

We compare our proposed method against five competing methods: gSeg (Chen & Zhang, 2015), kerSeg (Song & Chen, 2024), CPDrdpg (Madrid Padilla et al., 2022), CPDnbs (Wang et al., 2021), and CPDker (Madrid Padilla et al., 2021). The gSeg method utilizes a graph-based scan statistic and the kerSeg method employs a kernel-based scan statistic to assess distributional differences between two segments before and after a candidate change point. The CPDrdpg method identifies change points by estimating the latent positions of nodes using a Random Dot Product Graph Model (RDPGM) and by constructing a nonparametric CUSUM statistic that accounts for temporal dependence. The CPDnbs method integrates sample splitting with wild binary segmentation (WBS) and detects change points by maximizing the inner product between two CUSUM statistics derived from the split samples. The CPDker method detects change points by using a kernel density estimation approach to identify distributional shifts in multivariate data sequences. In particular, the gSeg and kerSeg methods are available in the respective R libraries `gSeg` (Chen et al., 2020b) and `kerSeg` (Song & Chen, 2022). The CPDrdpg, CPDnbs, and CPDker methods are available in the R library `changepoints` (Xu et al., 2022).

For gSeg, we use the minimum spanning tree to construct the similarity graph, and we use the original edge-count scan statistic to test the null hypothesis that there is no change point within a time span. The significance level is set to $\alpha = 0.05$. For kerSeg, we use the kernel-based scan statistic fGKCP$_1$ and we set the significance level $\alpha = 0.001$. Since gSeg and kerSeg are general methods for change point detection, we use networks (nets.) and network statistics (stats.) as two types of input data for comparison. For CPDrdpg, we let the number of intervals for wild binary segmentation (WBS) be $W = 50$, and we let the number of leading singular values of an adjacency matrix in the scaled PCA algorithm be $d = 5$ to fit a RDPGM. For CPDnbs, we let the number of intervals for WBS be $W = 15$ and we set the threshold for detection to the order of $n \log^2(T)$ as suggested by Wang et al. (2021). For CPDker, we let the number of intervals for WBS be $W = 30$ and we set the bandwith to the order of $(\log(T)/T)^{1/p}$ for the Gaussian kernel as suggested by Madrid Padilla et al. (2021), where $p$ is the number of network statistics. Throughout, we use these chosen settings, since they produce higher coverage metrics $C(\mathcal{G}, \mathcal{G}')$ for the competitors across different scenarios on average. Changing the above settings can improve their performance on some specifications, while severely jeopardizing their performance on other specifications.

**Scenario 1: Stochastic Block Model (SBM)**

In this scenario, we use Stochastic Block Model (SBM) to generate dynamic networks. As in Madrid Padilla et al. (2022), we construct two probability matrices $\boldsymbol{P}, \boldsymbol{Q} \in [0, 1]^{n \times n}$ and they are defined as

$$\boldsymbol{P}_{ij} = \begin{cases} 0.5, & i, j \in \mathcal{B}_l, l \in [3], \\ 0.3, & \text{otherwise,} \end{cases} \quad \text{and} \quad \boldsymbol{Q}_{ij} = \begin{cases} 0.45, & i, j \in \mathcal{B}_l, l \in [3], \\ 0.2, & \text{otherwise,} \end{cases}$$

where $\mathcal{B}_1, \mathcal{B}_2, \mathcal{B}_3$ are evenly sized clusters that form a partition of $\{1, \ldots, n\}$. We then construct a sequence of matrices $\boldsymbol{E}^t$ for $t = 1, \ldots, T$ such that

$$\boldsymbol{E}_{ij}^t = \begin{cases} \boldsymbol{P}_{ij}, & t \in \mathcal{A}_1 \cup \mathcal{A}_3, \\ \boldsymbol{Q}_{ij}, & t \in \mathcal{A}_2 \cup \mathcal{A}_4. \end{cases}$$

Lastly, the networks are generated with $\rho \in \{0.0, 0.5, 0.9\}$ as a time-dependent mechanism. For any $\rho$ and $t = 1, \ldots, T - 1$, we let $\boldsymbol{y}_{ij}^1 \sim \text{Bernoulli}(\boldsymbol{E}_{ij}^1)$ and

$$\boldsymbol{y}_{ij}^{t+1} \sim \begin{cases} \text{Bernoulli}\big(\rho(1 - \boldsymbol{E}_{ij}^{t+1}) + \boldsymbol{E}_{ij}^{t+1}\big), & \boldsymbol{y}_{ij}^t = 1, \\ \text{Bernoulli}\big((1 - \rho)\boldsymbol{E}_{ij}^{t+1}\big), & \boldsymbol{y}_{ij}^t = 0. \end{cases}$$

When $\rho = 0$, the probability to draw an edge for dyad $(i, j)$ at time $t + 1$ remains the same. This imposes a time-independent condition for a sequence of generated networks. On the contrary, when $\rho > 0$, the probability to draw an edge for dyad $(i, j)$ becomes greater at time $t + 1$ when there exists an edge at time $t$, and the probability becomes smaller when there does not exist an edge at time $t$.

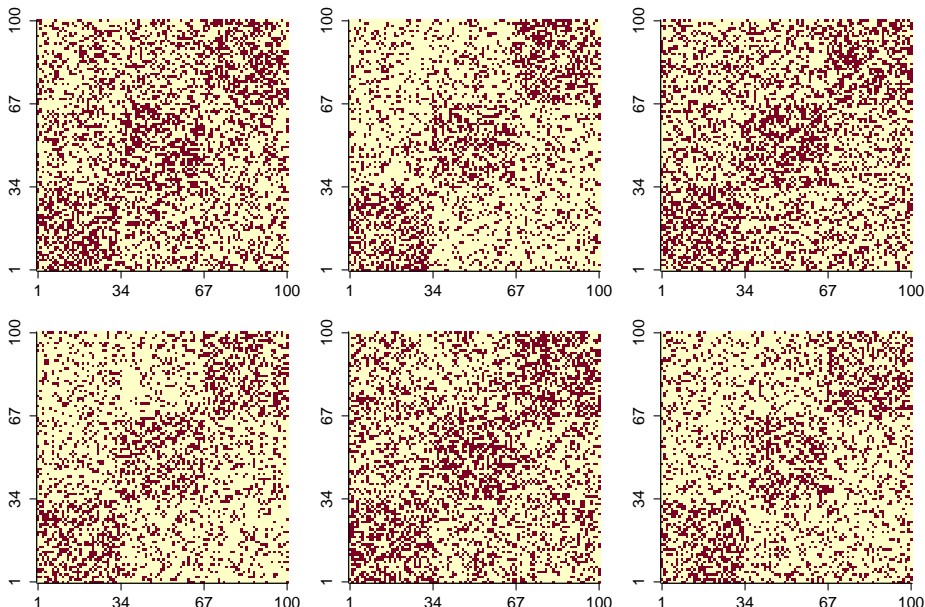

Figure 2: Examples of adjacency matrices generated from SBM with $\rho = 0.5$ and $n = 100$. In the first row, from left to right, each plot corresponds to the network at $t = 25, 50, 75$ respectively. In the second row, from left to right, each plot corresponds to the network at $t = 26, 51, 76$ respectively (the change points). In each display, a red dot indicates one and zero otherwise.

Figure 2 exhibits examples of generated networks at particular time points. Visually, Scenario 1 produces adjacency matrices with block structures, and mutuality is an important pattern in these networks. Hence, to detect the change points with our method, we use two network statistics, edge count and mutuality, in both formation and dissolution models. For gSeg and kerSeg, besides dynamic networks $\{\boldsymbol{y}^t\}_{t=1}^{T}$, we also use the two network statistics $\{\boldsymbol{g}(\boldsymbol{y}^t)\}_{t=1}^{T}$ as another specification. The CPDrdpg and CPDnbs methods directly use the networks as input data. Tables 1, 2, and 3 display the means and standard deviations of evaluation metrics for different specifications.

As expected, the CPDrdpg, CPDnbs, and kerSeg methods can achieve good performance in terms of the covering metric $C(\mathcal{G}, \mathcal{G}')$ when $\rho = 0$, since the time-independent setting aligns with the assumptions of these competitor methods. However, their performances are worsened when $\rho > 0$. In particular, when the networks in the sequence are time-dependent, the gSeg, kerSeg, and CPDnbs methods can effectively detect the true change points, as the one-sided Hausdorff distances $d(\hat{\mathcal{C}}|\mathcal{C})$ are small. Yet the reversed one-sided Hausdorff distance $d(\mathcal{C}|\hat{\mathcal{C}})$ and the absolute error $|\hat{K} - K|$ suggest that they tend to detect excessive number of change points as the sequences of networks become noisier under the time-dependent condition. Moreover, the kerSeg method can achieve a good performance when the temporal dependency is moderate with $\rho = 0.5$. The CPDrdpg method remains robust when the temporal dependency is strong with $\rho = 0.9$, and the performance improves as the number of nodes increases. Regardless of the temporal dependence, our CPDstergm method, on average, achieves smaller absolute error, smaller one-sided Hausdorff distances, and greater coverage of interval partitions, outperforming the competitor methods.

Another aspect worth mentioning is the usage of the network statistics in two competitor methods. The performance of gSeg and kerSeg methods, in terms of the covering metric $C(\mathcal{G}, \mathcal{G}')$, improves significantly when we change the input data from networks to network statistics, which demonstrates the potential of using network level summary statistics to represent the enormous amount of individual relations.

Table 1: Means (standard deviations) of evaluation metrics for dynamic networks simulated from the Stochastic Block Model with $\rho = 0.0$. The best coverage metric is bolded.

| $\rho$ | $n$ | Method | $\|\hat{K} - K\| \downarrow$ | $d(\hat{\mathcal{C}}\|\mathcal{C}) \downarrow$ | $d(\mathcal{C}\|\hat{\mathcal{C}}) \downarrow$ | $C(\mathcal{G}, \mathcal{G}') \uparrow$ |
|---|---|---|---|---|---|---|
| 0.0 | 50 | CPDstergm | 0.2 (0.6) | 0.8 (0.4) | 1.7 (2.7) | 95.99% |
| | | CPDrdpg | 0.3 (0.8) | 2.2 (1.2) | 3.6 (3.6) | 89.32% |
| | | CPDnbs | 0.1 (0.3) | 3.3 (5.8) | 1.8 (0.8) | 92.17% |
| | | CPDker | 0.0 (0.0) | 17.3 (7.0) | 10.7 (1.8) | 58.67% |
| | | gSeg (nets.) | 2.8 (0.4) | Inf (na) | Inf (na) | 9.08% |
| | | kerSeg (nets.) | 0.0 (0.0) | 0.0 (0.0) | 0.0 (0.0) | **100**% |
| | | gSeg (stats.) | 2.1 (0.4) | Inf (na) | Inf (na) | 43.68% |
| | | kerSeg (stats.) | 0.1 (0.3) | 0.1 (0.3) | 0.3 (0.8) | 99.67% |
| 0.0 | 100 | CPDstergm | 0.7 (1.3) | 0.8 (0.4) | 5.0 (6.4) | 91.34% |
| | | CPDrdpg | 0.3 (0.6) | 1.3 (0.6) | 2.3 (2.3) | 92.33% |
| | | CPDnbs | 0.1 (0.4) | 3.3 (5.7) | 2.5 (2.6) | 90.46% |
| | | CPDker | 0.1 (0.3) | 21.3 (10.6) | 10.5 (2.7) | 57.92% |
| | | gSeg (nets.) | 2.9 (0.3) | Inf (na) | Inf (na) | 3.20% |
| | | kerSeg (nets.) | 0.0 (0.0) | 0.0 (0.0) | 0.0 (0.0) | **100**% |
| | | gSeg (stats.) | 1.9 (0.6) | Inf (na) | Inf (na) | 45.55% |
| | | kerSeg (stats.) | 0.0 (0.0) | 0.0 (0.0) | 0.0 (0.0) | **100**% |
| 0.0 | 200 | CPDstergm | 0.2 (0.4) | 0.8 (0.4) | 2.7 (3.9) | 95.33% |
| | | CPDrdpg | 0.1 (0.3) | 1.0 (0.0) | 1.5 (1.8) | 92.85% |
| | | CPDnbs | 0.1 (0.3) | 3.3 (5.7) | 1.9 (0.4) | 91.40% |
| | | CPDker | 0.1 (0.3) | 29.2 (10.5) | 11.7 (2.4) | 48.40% |
| | | gSeg (nets.) | 2.9 (0.4) | Inf (na) | Inf (na) | 6.01% |
| | | kerSeg (nets.) | 0.0 (0.0) | 0.0 (0.0) | 0.0 (0.0) | **100**% |
| | | gSeg (stats.) | 1.9 (0.6) | Inf (na) | Inf (na) | 42.28% |
| | | kerSeg (stats.) | 0.0 (0.0) | 0.0 (0.0) | 0.0 (0.0) | **100**% |

Table 2: Means (standard deviations) of evaluation metrics for dynamic networks simulated from the Stochastic Block Model with $\rho = 0.5$. The best coverage metric is bolded.

| $\rho$ | $n$ | Method | $\|\hat{K} - K\| \downarrow$ | $d(\hat{\mathcal{C}}\|\mathcal{C}) \downarrow$ | $d(\mathcal{C}\|\hat{\mathcal{C}}) \downarrow$ | $C(\mathcal{G}, \mathcal{G}') \uparrow$ |
|---|---|---|---|---|---|---|
| 0.5 | 50 | CPDstergm | 0.0 (0.0) | 1.0 (0.0) | 1.0 (0.0) | **98.04**% |
| | | CPDrdpg | 1.3 (1.7) | 2.5 (1.4) | 7.5 (5.9) | 81.47% |
| | | CPDnbs | 1.6 (0.6) | 3.3 (3.2) | 11.4 (1.1) | 73.54% |
| | | CPDker | 0.2 (0.4) | 24.9 (9.7) | 10.7 (2.3) | 56.80% |
| | | gSeg (nets.) | 12.9 (1.8) | 0.0 (0.0) | 19.3 (0.7) | 27.20% |
| | | kerSeg (nets.) | 6.3 (1.4) | 0.0 (0.0) | 16.5 (2.6) | 45.87% |
| | | gSeg (stats.) | 1.7 (1.1) | 42.7 (20.2) | 7.9 (7.8) | 50.92% |
| | | kerSeg (stats.) | 0.7 (0.8) | 0.0 (0.0) | 5.3 (7.1) | 95.13% |
| 0.5 | 100 | CPDstergm | 0.0 (0.0) | 1.0 (0.0) | 1.0 (0.0) | **98.04**% |
| | | CPDrdpg | 0.3 (0.6) | 1.4 (0.5) | 2.8 (3.1) | 91.07% |
| | | CPDnbs | 1.8 (0.7) | 2.9 (2.9) | 12.2 (1.1) | 72.17% |
| | | CPDker | 0.1 (0.3) | 18.1 (8.1) | 10.3 (2.9) | 58.33% |
| | | gSeg (nets.) | 12.5 (1.1) | 0.0 (0.0) | 19.3 (0.7) | 27.60% |
| | | kerSeg (nets.) | 6.1 (1.0) | 0.0 (0.0) | 15.0 (2.1) | 46.40% |
| | | gSeg (stats.) | 1.7 (0.7) | Inf (na) | Inf (na) | 53.18% |
| | | kerSeg (stats.) | 0.9 (0.6) | 0.0 (0.0) | 8.1 (7.3) | 93.67% |
| 0.5 | 200 | CPDstergm | 0.0 (0.0) | 1.0 (0.0) | 1.0 (0.0) | **98.04**% |
| | | CPDrdpg | 0.0 (0.0) | 1.0 (0.0) | 1.0 (0.0) | 93.32% |
| | | CPDnbs | 1.8 (0.7) | 3.2 (3.6) | 11.8 (0.9) | 71.43% |
| | | CPDker | 0.1 (0.3) | 26.7 (12.7) | 11.9 (2.2) | 53.18% |
| | | gSeg (nets.) | 12.2 (0.6) | 0.0 (0.0) | 19.1 (0.5) | 27.87% |
| | | kerSeg (nets.) | 4.5 (0.7) | 0.0 (0.0) | 13.8 (1.7) | 52.00% |
| | | gSeg (stats.) | 1.6 (0.7) | Inf (na) | Inf (na) | 54.36% |
| | | kerSeg (stats.) | 0.5 (0.6) | 0.0 (0.0) | 4.4 (6.1) | 96.33% |

Table 3: Means (standard deviations) of evaluation metrics for dynamic networks simulated from the Stochastic Block Model with $\rho = 0.9$. The best coverage metric is bolded.

| $\rho$ | $n$ | Method | $|\hat{K} - K| \downarrow$ | $d(\hat{\mathcal{C}}|\mathcal{C}) \downarrow$ | $d(\mathcal{C}|\hat{\mathcal{C}}) \downarrow$ | $C(\mathcal{G}, \mathcal{G}') \uparrow$ |
|---|---|---|---|---|---|---|
| 0.9 | 50 | CPDstergm | 0.0 (0.0) | 1.0 (0.0) | 1.0 (0.0) | **98.04**% |
| | | CPDrdpg | 1.1 (1.4) | 2.5 (1.1) | 8.4 (5.7) | 83.13% |
| | | CPDnbs | 1.3 (0.6) | 2.7 (2.6) | 10.7 (2.6) | 76.38% |
| | | CPDker | 0.0 (0.0) | 16.9 (6.8) | 10.5 (2.2) | 60.43% |
| | | gSeg (nets.) | 12.4 (1.1) | 0.0 (0.0) | 19.1 (0.5) | 27.67% |
| | | kerSeg (nets.) | 11.1 (0.7) | 0.0 (0.0) | 18.9 (0.6) | 31.53% |
| | | gSeg (stats.) | 6.2 (3.8) | 6.6 (14.8) | 16.7 (4.8) | 57.38% |
| | | kerSeg (stats.) | 4.1 (1.7) | 0.0 (0.0) | 15.1 (3.4) | 72.67% |
| 0.9 | 100 | CPDstergm | 0.0 (0.0) | 1.0 (0.0) | 1.0 (0.0) | **98.04**% |
| | | CPDrdpg | 0.1 (0.3) | 1.6 (0.8) | 1.9 (1.6) | 92.14% |
| | | CPDnbs | 1.4 (0.6) | 2.5 (2.1) | 10.4 (2.5) | 76.44% |
| | | CPDker | 0.2 (0.6) | 21.1 (14.7) | 9.3 (3.1) | 60.52% |
| | | gSeg (nets.) | 12.4 (0.9) | 0.0 (0.0) | 19.0 (0.0) | 27.67% |
| | | kerSeg (nets.) | 11.9 (0.3) | 0.0 (0.0) | 19.0 (0.0) | 28.27% |
| | | gSeg (stats.) | 5.3 (2.3) | 3.8 (9.0) | 18.7 (3.0) | 62.81% |
| | | kerSeg (stats.) | 3.7 (1.2) | 0.0 (0.0) | 17.1 (3.5) | 73.27% |
| 0.9 | 200 | CPDstergm | 0.0 (0.0) | 1.0 (0.0) | 1.0 (0.0) | **98.04**% |
| | | CPDrdpg | 0.0 (0.0) | 1.0 (0.0) | 1.0 (0.0) | 93.19% |
| | | CPDnbs | 1.3 (0.6) | 2.5 (2.1) | 10.3 (2.5) | 77.38% |
| | | CPDker | 0.1 (0.3) | 22.5 (12.5) | 11.1 (2.5) | 54.72% |
| | | gSeg (nets.) | 12.3 (1.0) | 0.0 (0.0) | 19.1 (0.3) | 27.73% |
| | | kerSeg (nets.) | 12.0 (0.0) | 0.0 (0.0) | 18.9 (0.3) | 28.00% |
| | | gSeg (stats.) | 7.4 (2.4) | 0.9 (1.8) | 17.9 (3.8) | 57.61% |
| | | kerSeg (stats.) | 4.8 (1.7) | 0.0 (0.0) | 16.7 (4.9) | 66.87% |

**Scenario 2: Separable Temporal ERGM**

In this scenario, we employ time-homogeneous STERGMs (Krivitsky & Handcock, 2014) between change points to generate sequences of dynamic networks, using the R package `tergm` (Krivitsky & Handcock, 2022). For the following three specifications, we gradually increase the complexity of the network patterns, by adding more network statistics in the data generating process. First we use two network statistics, edge count and mutuality, in both formation and dissolution models to let $p = 4$. The parameters are

$$\boldsymbol{\theta}^{+,t}, \boldsymbol{\theta}^{-,t} = \begin{cases} -1, \ -2, \ -1, \ -2, & t \in \mathcal{A}_1 \cup \mathcal{A}_3, \\ -1, \ 1, \ -1, \ -1, & t \in \mathcal{A}_2 \cup \mathcal{A}_4. \end{cases}$$

Next, we include the number of triangles in both formation and dissolution models to let $p = 6$. The parameters are

$$\boldsymbol{\theta}^{+,t}, \boldsymbol{\theta}^{-,t} = \begin{cases} -2, \ 2, \ -2, \ -1, \ 2, \ 1, & t \in \mathcal{A}_1 \cup \mathcal{A}_3, \\ -1.5, \ 1, \ -1, \ 2, \ 1, \ 1.5, & t \in \mathcal{A}_2 \cup \mathcal{A}_4. \end{cases}$$

Finally, we include the homophily for gender, an attribute assigned to each node, in both formation and dissolution models to let $p = 8$. The parameters are

$$\boldsymbol{\theta}^{+,t}, \boldsymbol{\theta}^{-,t} = \begin{cases} -2, \ 2, \ -2, \ -1, \ -1, \ 2, \ 1, \ 1, & t \in \mathcal{A}_1 \cup \mathcal{A}_3, \\ -1.5, \ 1, \ -1, \ 1, \ 2, \ 1, \ 1.5, \ 2, & t \in \mathcal{A}_2 \cup \mathcal{A}_4. \end{cases}$$

The nodal attributes, $\boldsymbol{x}_i \in \{\text{Female}, \text{Male}\}$ for $i \in [n]$, are fixed across time $t$ in the generation process.

Figure 3 exhibits examples of generated networks at particular time points. Specifically, Scenario 2 produces adjacency matrices that are sparse, which is often the case in reality. For comparison, to detect change points with our method, we use the network statistics that generate the networks in both formation and dissolution models. For gSeg and kerSeg methods, besides using the networks as one specification, we also

use the same network statistics that generate the simulated networks as another specification. The CPDrdpg and CPDnbs methods directly use the networks as input data. Tables 4, 5, and 6 display the means and standard deviations of evaluation metrics for different specifications.

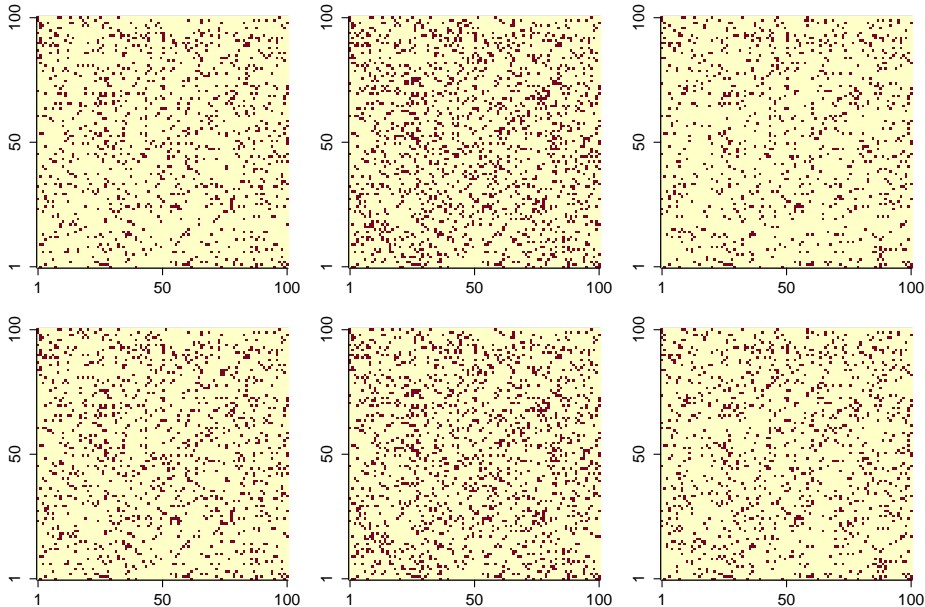

Figure 3: Examples of adjacency matrices generated from STERGM with $p = 6$ and $n = 100$. In the first row, from left to right, each plot corresponds to the network at $t = 25, 50, 75$ respectively. In the second row, from left to right, each plot corresponds to the network at $t = 26, 51, 76$ respectively (the change points). In each display, a red dot indicates one and zero otherwise.

For $p = 4$, the performance of the kerSeg method in terms of the covering metric $C(\mathcal{G}, \mathcal{G}')$ improves significantly when we substitute the input data from networks to network statistics. The CPDrdpg and CPDnbs methods can also achieve good performance when the dyadic dependency is weak. However, for $p = 6$, the competitor methods tend to detect excessive number of change points, when the simulated networks are highly dyadic dependent due to the inclusion of the transitivity. Using network statistics as input can no longer improve the performance of gSeg and kerSeg, in contrast to Scenario 1 where the dyadic dependency is relatively mild. Nevertheless, our CPDstergm method, which dissects the network evolution using formation and dissolution processes, can achieve a good performance when the simulated networks are both temporal and dyadic dependent. Lastly, for $p = 8$, the performance of CPDrdpg, CPDnbs, and CPDker methods does not deteriorate, and that of the CPDrdpg method improves as the number of nodes increases. Notably, our method permits the inclusion of nodal attributes to facilitate change point detection, a feature that many existing methods for dynamic graphs do not offer. On average, our method produces smaller absolute error, smaller one-sided Hausdorff distances, and greater coverage of interval partitions, outperforming the competitor methods at different levels of dyadic dependency.

### Scenario 3: Random Dot Product Graph Model (RDPGM)

In this scenario, we simulate dynamic networks using the Random Dot Product Graph Model (Young & Scheinerman, 2007; Athreya et al., 2018). At time point $t = 1$, we generate two latent positions $\boldsymbol{X}_i^t, \boldsymbol{Z}_i^t \in \mathbb{R}^d$ for each node $i \in [n]$ from a multivariate Normal distribution as $\boldsymbol{X}_i^1, \boldsymbol{Z}_i^1 \sim \mathcal{N}(\mathbf{1}, \boldsymbol{I}_d)$. For subsequent time points, the latent positions evolve according to the following autoregressive process:

$$\boldsymbol{X}_i^{t+1} = \rho \boldsymbol{X}_i^t + (1 - \rho)\boldsymbol{\epsilon}_i^t, \quad \boldsymbol{Z}_i^{t+1} = \rho \boldsymbol{Z}_i^t + (1 - \rho)\boldsymbol{\epsilon}_i^t, \quad t = 1, \ldots, T - 1,$$

where $\boldsymbol{\epsilon}_i^t \sim \mathcal{N}(\mathbf{0}, \boldsymbol{I}_d)$. Throughout, we set $\rho = 0.9$ to induce the temporal dependence, and we normalize the resulting latent positions at each time point. Next, to incorporate structural patterns in the dynamic

Table 4: Means (standard deviations) of evaluation metrics for dynamic networks simulated from the STERGM with $p = 4$. The best coverage metric is bolded.

| $p$ | $n$ | Method | $|\hat{K} - K| \downarrow$ | $d(\hat{\mathcal{C}}|\mathcal{C}) \downarrow$ | $d(\mathcal{C}|\hat{\mathcal{C}}) \downarrow$ | $C(\mathcal{G}, \mathcal{G}') \uparrow$ |
|---|---|---|---|---|---|---|
| 4 | 50 | CPDstergm | 0.0 (0.0) | 0.0 (0.0) | 0.0 (0.0) | **100**% |
| | | CPDrdpg | 0.0 (0.0) | 1.5 (0.7) | 1.5 (0.7) | 92.48% |
| | | CPDnbs | 0.1 (0.3) | 3.7 (5.7) | 2.3 (1.0) | 87.67% |
| | | CPDker | 0.3 (0.5) | 19.7 (13.1) | 10.7 (2.3) | 61.61% |
| | | gSeg (nets.) | 1.6 (1.0) | 23.1 (7.0) | 13.1 (8.3) | 45.62% |
| | | kerSeg (nets.) | 2.6 (0.7) | 0.0 (0.0) | 13.5 (4.3) | 80.53% |
| | | gSeg (stats.) | 2.0 (0.4) | 48.1 (13.4) | 4.2 (7.0) | 50.26% |
| | | kerSeg (stats.) | 0.0 (0.0) | 0.0 (0.0) | 0.0 (0.0) | **100**% |
| 4 | 100 | CPDstergm | 0.0 (0.0) | 0.0 (0.0) | 0.0 (0.0) | **100**% |
| | | CPDrdpg | 0.1 (0.3) | 1.0 (0.0) | 1.3 (1.3) | 92.86% |
| | | CPDnbs | 0.1 (0.3) | 3.5 (5.7) | 2.0 (0.0) | 88.13% |
| | | CPDker | 0.3 (0.5) | 17.7 (6.5) | 9.5 (3.4) | 60.44% |
| | | gSeg (nets.) | 1.5 (1.2) | 20.5 (6.5) | 15.5 (7.3) | 45.21% |
| | | kerSeg (nets.) | 2.7 (0.5) | 0.0 (0.0) | 16.2 (2.9) | 76.87% |
| | | gSeg (stats.) | 2.3 (0.5) | Inf (na) | Inf (na) | 40.42% |
| | | kerSeg (stats.) | 0.1 (0.4) | 0.2 (0.4) | 0.6 (1.1) | 99.21% |
| 4 | 200 | CPDstergm | 0.0 (0.0) | 0.3 (0.5) | 0.3 (0.5) | **99.48**% |
| | | CPDrdpg | 0.0 (0.0) | 1.0 (0.0) | 1.0 (0.0) | 93.19% |
| | | CPDnbs | 0.1 (0.3) | 4.1 (6.1) | 2.7 (2.6) | 87.08% |
| | | CPDker | 0.1 (0.3) | 26.5 (11.2) | 11.7 (2.3) | 53.58% |
| | | gSeg (nets.) | 1.5 (1.4) | 22.7 (6.3) | 15.6 (7.3) | 46.82% |
| | | kerSeg (nets.) | 2.5 (0.6) | 0.0 (0.0) | 15.3 (3.0) | 76.60% |
| | | gSeg (stats.) | 2.0 (0.9) | Inf (na) | Inf (na) | 37.35% |
| | | kerSeg (stats.) | 0.1 (0.3) | 0.0 (0.0) | 1.0 (2.6) | 99.33% |

Table 5: Means (standard deviations) of evaluation metrics for dynamic networks simulated from the STERGM with $p = 6$. The best coverage metric is bolded.

| $p$ | $n$ | Method | $|\hat{K} - K| \downarrow$ | $d(\hat{\mathcal{C}}|\mathcal{C}) \downarrow$ | $d(\mathcal{C}|\hat{\mathcal{C}}) \downarrow$ | $C(\mathcal{G}, \mathcal{G}') \uparrow$ |
|---|---|---|---|---|---|---|
| 6 | 50 | CPDstergm | 0.0 (0.0) | 1.1 (0.3) | 1.1 (0.3) | **94.20**% |
| | | CPDrdpg | 1.2 (1.7) | 6.0 (5.5) | 8.0 (4.9) | 75.61% |
| | | CPDnbs | 1.5 (0.5) | 4.3 (2.3) | 11.3 (1.0) | 75.64% |
| | | CPDker | 0.0 (0.0) | 17.9 (4.9) | 11.1 (2.6) | 54.90% |
| | | gSeg (nets.) | 13.1 (1.2) | 0.0 (0.0) | 19.4 (1.1) | 27.47% |
| | | kerSeg (nets.) | 10.1 (1.0) | 1.5 (1.1) | 18.5 (1.5) | 35.82% |
| | | gSeg (stats.) | 15.3 (1.2) | 1.5 (0.6) | 20.6 (0.6) | 25.18% |
| | | kerSeg (stats.) | 9.7 (1.2) | 3.3 (1.3) | 19.4 (1.8) | 35.28% |
| 6 | 100 | CPDstergm | 0.0 (0.0) | 1.1 (0.4) | 1.1 (0.4) | **93.95**% |
| | | CPDrdpg | 0.9 (1.0) | 4.9 (1.7) | 8.6 (3.8) | 77.59% |
| | | CPDnbs | 1.4 (0.5) | 4.9 (2.2) | 10.9 (1.0) | 73.84% |
| | | CPDker | 0.0 (0.0) | 29.9 (9.4) | 10.7 (2.2) | 50.48% |
| | | gSeg (nets.) | 12.0 (0.0) | 0.0 (0.0) | 19.0 (0.0) | 28.00% |
| | | kerSeg (nets.) | 10.2 (0.9) | 0.5 (0.5) | 18.1 (1.1) | 36.13% |
| | | gSeg (stats.) | 14.9 (3.2) | 2.8 (4.9) | 20.5 (0.6) | 25.09% |
| | | kerSeg (stats.) | 8.1 (1.2) | 5.2 (1.6) | 17.9 (1.8) | 38.32% |
| 6 | 200 | CPDstergm | 0.1 (0.3) | 1.0 (0.0) | 2.1 (4.4) | **97.06**% |
| | | CPDrdpg | 0.3 (0.6) | 2.4 (0.8) | 3.3 (2.3) | 91.08% |
| | | CPDnbs | 1.4 (0.5) | 4.3 (1.9) | 11.9 (0.5) | 75.58% |
| | | CPDker | 0.0 (0.0) | 35.2 (4.3) | 10.7 (1.7) | 48.77% |
| | | gSeg (nets.) | 12.0 (0.0) | 0.0 (0.0) | 19.0 (0.0) | 28.00% |
| | | kerSeg (nets.) | 10.0 (0.7) | 0.9 (0.4) | 18.1 (0.9) | 36.17% |
| | | gSeg (stats.) | 5.5 (6.6) | 28.6 (22.2) | 20.4 (0.9) | 31.91% |
| | | kerSeg (stats.) | 8.7 (1.0) | 3.4 (0.8) | 18.9 (0.4) | 42.86% |

Table 6: Means (standard deviations) of evaluation metrics for dynamic networks simulated from the STERGM with $p = 8$. The best coverage metric is bolded.

| $p$ | $n$ | Method | $|\hat{K} - K| \downarrow$ | $d(\hat{\mathcal{C}}|\mathcal{C}) \downarrow$ | $d(\mathcal{C}|\hat{\mathcal{C}}) \downarrow$ | $C(\mathcal{G}, \mathcal{G}') \uparrow$ |
|---|---|---|---|---|---|---|
| 8 | 50 | CPDstergm | 0.4 (0.6) | 3.7 (5.5) | 5.1 (3.8) | **86.32**% |
| | | CPDrdpg | 1.3 (1.1) | 9.4 (7.1) | 11.3 (5.4) | 70.72% |
| | | CPDnbs | 1.3 (0.6) | 4.9 (4.2) | 11.8 (0.6) | 75.57% |
| | | CPDker | 0.1 (0.3) | 20.5 (8.7) | 12.6 (2.1) | 51.15% |
| | | gSeg (nets.) | 13.5 (1.2) | 0.0 (0.0) | 19.7 (1.2) | 27.07% |
| | | kerSeg (nets.) | 10.4 (1.5) | 1.5 (1.2) | 19.1 (1.8) | 36.58% |
| | | gSeg (stats.) | 14.6 (2.2) | 2.7 (1.7) | 19.9 (1.2) | 27.10% |
| | | kerSeg (stats.) | 9.3 (1.3) | 4.8 (1.9) | 18.5 (1.8) | 36.23% |
| 8 | 100 | CPDstergm | 0.0 (0.0) | 1.9 (1.2) | 1.9 (1.2) | **92.65**% |
| | | CPDrdpg | 0.8 (1.3) | 7.1 (3.1) | 9.0 (3.6) | 73.13% |
| | | CPDnbs | 1.3 (0.5) | 5.1 (3.2) | 12.0 (0.0) | 75.51% |
| | | CPDker | 0.1 (0.3) | 29.8 (9.5) | 12.0 (1.8) | 49.71% |
| | | gSeg (nets.) | 12.0 (0.0) | 0.0 (0.0) | 19.0 (0.0) | 28.00% |
| | | kerSeg (nets.) | 9.6 (1.2) | 1.3 (1.3) | 18.0 (1.1) | 37.31% |
| | | gSeg (stats.) | 12.9 (1.3) | 3.3 (2.4) | 19.9 (0.7) | 28.56% |
| | | kerSeg (stats.) | 8.7 (1.4) | 5.9 (2.2) | 18.4 (1.7) | 35.78% |
| 8 | 200 | CPDstergm | 0.0 (0.0) | 1.0 (0.0) | 1.0 (0.0) | **94.45**% |
| | | CPDrdpg | 0.7 (1.0) | 4.2 (2.0) | 5.7 (2.2) | 85.59% |
| | | CPDnbs | 1.4 (0.5) | 5.3 (3.1) | 11.1 (1.0) | 73.38% |
| | | CPDker | 0.1 (0.3) | 33.3 (5.9) | 10.5 (2.0) | 49.89% |
| | | gSeg (nets.) | 12.0 (0.0) | 0.0 (0.0) | 19.0 (0.0) | 28.00% |
| | | kerSeg (nets.) | 9.1 (0.6) | 0.0 (0.0) | 16.9 (1.0) | 38.13% |
| | | gSeg (stats.) | 14.1 (0.9) | 1.1 (0.8) | 19.6 (0.5) | 26.15% |
| | | kerSeg (stats.) | 8.8 (1.0) | 2.9 (0.4) | 17.6 (0.5) | 38.60% |

networks, we generate two weight matrices $\boldsymbol{U}, \boldsymbol{V} \in \mathbb{R}^{d \times d}$, where each entry is sampled independently as

$$\boldsymbol{U}_{r,s} \sim \text{Uniform}(0, 1/16), \quad \boldsymbol{V}_{r,s} \sim \text{Uniform}(1/16, 2/16), \quad \forall \, r, s \in \{1, \ldots, d\},$$

and the weight matrices remain fixed within the corresponding time segments between consecutive change points. Lastly, at each time point $t$, the adjacency matrix $\boldsymbol{y}^t \in \{0, 1\}^{n \times n}$ is generated as

$$\boldsymbol{y}_{ij}^t \sim \begin{cases} \text{Bernoulli}(\boldsymbol{X}_i^{t\top} \boldsymbol{U} \boldsymbol{Z}_j^t), & t \in \mathcal{A}_1 \cup \mathcal{A}_3, \\ \text{Bernoulli}(\boldsymbol{X}_i^{t\top} \boldsymbol{V} \boldsymbol{Z}_j^t), & t \in \mathcal{A}_2 \cup \mathcal{A}_4. \end{cases}$$

The latent dimension $d$ is varied across simulation settings with $d \in \{10, 15, 20\}$.

Figure 4 presents examples of generated networks at specific time points. In particular, Scenario 3 produces adjacency matrices that are sparse, and no specific network pattern can be noticed. To detect change points with the proposed method, we use two network statistics, edge count and mutuality, in both formation and dissolution models, since the networks are generated based on the similarity in latent positions between nodes. For gSeg and kerSeg methods, besides dynamic networks, we also use these two network statistics as another specification. The CPDrdpg and CPDnbs methods directly use the networks as input data. Tables 7, 8, and 9 display the means and standard deviations of evaluation metrics for different specifications.

The weight matrices, generated from the Uniform distributions with small support, induce sparsity in the simulated networks, posing challenges for both the proposed and competitor methods. Although most of the methods are able to detect the true change points, they also tend to identify additional change points that deviate substantially from the ground truth, as reflected by high absolute errors and large one-sided Hausdorff distances. Moreover, as the networks are generated from latent positions that differ by nodes, we observe a degradation in the performance of gSeg and CPDnbs when the latent dimension $d$ grows, partly due to the increased complexity of the underlying network structures. In contrast, the performance of kerSeg, which uses network statistics as input, and that of CPDrdpg, which aligns closely with the data generating mechanism, improve when the latent dimension $d$ increases. Similarly, the proposed CPDstergm

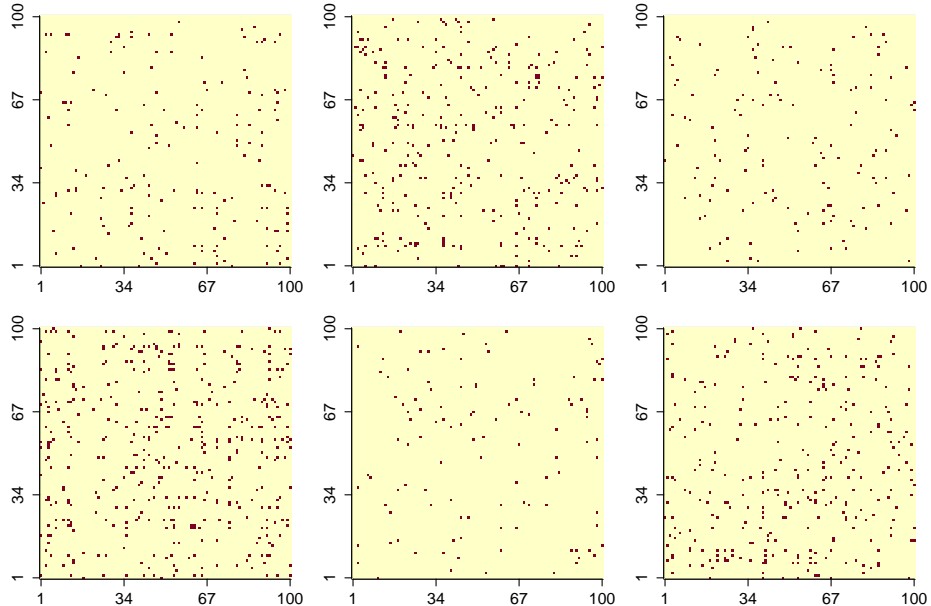

Figure 4: Examples of adjacency matrices generated from RDPGM with $d = 15$ and $n = 100$. In the first row, from left to right, each plot corresponds to the network at $t = 25, 50, 75$ respectively. In the second row, from left to right, each plot corresponds to the network at $t = 26, 51, 76$ respectively (the change points). In each display, a red dot indicates one and zero otherwise.

method demonstrates robustness to different latent dimensions and the performance improves with larger network sizes. On the other hand, the gSeg and kerSeg methods show substantial gains when applied to network statistics instead of dynamic networks. The performance of CPDker remains consistent across varied latent dimensions. The proposed CPDstergm method, which also utilizes network statistics, outperforms the competitor methods, indicating its adaptability to sparse and temporal dependent networks.

### 5.1.1 Time Comparison

We compare the computation times of different methods across varying node sizes, and the results are provided in Table 10. We focus on three representative configurations where the change point detection performance is comparable across methods, as shown in Section 5.1. The reported computation times in seconds reflect the total runtime over three simulated network time series. It is worth noting that some competing methods exhibit shorter runtime when they fail to detect the correct change points and terminate early. Although the proposed method requires more computation time, it achieves better performance on average compared to the competitors.

### 5.2 MIT Cellphone Data

The Massachusetts Institute of Technology (MIT) cellphone data (Eagle & Pentland, 2006) consists of human interactions via cellphone activity, among $n = 96$ participants for a duration of $T = 232$ days. The data were taken from 2004-09-15 to 2005-05-04, which covers the winter and spring vacations in the MIT academic calendar. For participants $i$ and $j$, a connected edge $\boldsymbol{y}_{ij}^t = 1$ indicates that they had made at least one phone call on day $t$, and $\boldsymbol{y}_{ij}^t = 0$ otherwise.

As the data portrays human interactions, we use the number of edges, isolates, and triangles to represent the occurrence of connections, the sparsity of social networks, and the transitive association of friendship, respectively. For gSeg, and kerSeg methods, we use these three network statistics $\{\boldsymbol{g}(\boldsymbol{y}^t)\}_{t=1}^T$ as input data, since they produce more informative results than using the dynamic networks $\{\boldsymbol{y}^t\}_{t=1}^T$. The CPDrdpg and CPDnbs methods directly use the dynamic networks, and CPDker method uses network statistics.

Table 7: Means (standard deviations) of evaluation metrics for dynamic networks simulated from the RDPGM with $d = 10$. The best coverage metric is bolded.

| $d$ | $n$ | Method | $|\hat{K} - K| \downarrow$ | $d(\hat{\mathcal{C}}|\mathcal{C}) \downarrow$ | $d(\mathcal{C}|\hat{\mathcal{C}}) \downarrow$ | $C(\mathcal{G}, \mathcal{G}') \uparrow$ |
|---|---|---|---|---|---|---|
| 10 | 50 | CPDstergm | 1.8 (1.1) | 1.1 (0.9) | 13.1 (5.5) | **81.61**% |
| | | CPDrdpg | 2.0 (1.1) | 5.7 (5.0) | 18.3 (2.2) | 72.82% |
| | | CPDnbs | 1.0 (0.5) | 5.2 (3.4) | 10.2 (3.3) | 76.71% |
| | | CPDker | 0.1 (0.3) | 20.0 (8.3) | 9.7 (3.1) | 64.49% |
| | | gSeg (nets.) | 3.0 (0.0) | Inf (na) | Inf (na) | 0.00% |
| | | kerSeg (nets.) | 4.2 (1.4) | 3.3 (4.4) | 16.0 (3.1) | 61.06% |
| | | gSeg (stats.) | 2.4 (1.3) | 16.9 (17.3) | 19.7 (1.1) | 64.26% |
| | | kerSeg (stats.) | 2.5 (1.5) | 6.0 (5.5) | 19.5 (2.0) | 73.06% |
| 10 | 100 | CPDstergm | 0.4 (0.5) | 0.9 (0.4) | 6.5 (7.5) | **93.34**% |
| | | CPDrdpg | 1.9 (1.0) | 6.6 (4.6) | 17.7 (2.3) | 71.83% |
| | | CPDnbs | 1.0 (0.8) | 5.3 (4.1) | 9.5 (4.1) | 77.52% |
| | | CPDker | 0.3 (0.5) | 26.7 (9.4) | 9.9 (2.3) | 57.39% |
| | | gSeg (nets.) | 2.9 (0.4) | Inf (na) | Inf (na) | 5.87% |
| | | kerSeg (nets.) | 7.1 (1.1) | 1.5 (3.8) | 17.8 (2.0) | 46.90% |
| | | gSeg (stats.) | 4.1 (2.2) | 10.5 (10.9) | 20.4 (1.0) | 60.88% |
| | | kerSeg (stats.) | 4.9 (1.5) | 0.8 (1.7) | 19.3 (2.3) | 68.43% |
| 10 | 200 | CPDstergm | 0.0 (0.0) | 1.0 (0.0) | 1.0 (0.0) | **96.36**% |
| | | CPDrdpg | 2.3 (0.7) | 5.6 (2.5) | 18.7 (2.2) | 71.59% |
| | | CPDnbs | 1.5 (0.7) | 6.5 (3.4) | 11.4 (2.7) | 70.36% |
| | | CPDker | 0.0 (0.0) | 24.4 (8.5) | 10.4 (1.7) | 57.14% |
| | | gSeg (nets.) | 2.7 (0.6) | Inf (na) | Inf (na) | 13.66% |
| | | kerSeg (nets.) | 8.1 (1.8) | 0.1 (0.4) | 18.5 (1.4) | 43.41% |
| | | gSeg (stats.) | 4.9 (1.3) | 7.3 (3.9) | 20.1 (0.3) | 62.78% |
| | | kerSeg (stats.) | 5.4 (1.3) | 2.2 (3.9) | 20.1 (1.5) | 61.71% |

Table 8: Means (standard deviations) of evaluation metrics for dynamic networks simulated from the RDPGM with $d = 15$. The best coverage metric is bolded.

| $d$ | $n$ | Method | $|\hat{K} - K| \downarrow$ | $d(\hat{\mathcal{C}}|\mathcal{C}) \downarrow$ | $d(\mathcal{C}|\hat{\mathcal{C}}) \downarrow$ | $C(\mathcal{G}, \mathcal{G}') \uparrow$ |
|---|---|---|---|---|---|---|
| 15 | 50 | CPDstergm | 1.7 (1.0) | 1.9 (3.9) | 14.7 (2.7) | **80.38**% |
| | | CPDrdpg | 2.4 (0.6) | 1.6 (0.6) | 16.9 (1.9) | 75.90% |
| | | CPDnbs | 1.5 (0.7) | 6.0 (4.1) | 11.3 (1.7) | 70.88% |
| | | CPDker | 0.2 (0.4) | 18.6 (10.0) | 7.3 (2.5) | 66.67% |
| | | gSeg (nets.) | 3.0 (0.0) | Inf (na) | Inf (na) | 0.00% |
| | | kerSeg (nets.) | 5.3 (1.4) | 3.3 (4.5) | 18.7 (1.8) | 56.01% |
| | | gSeg (stats.) | 2.6 (1.8) | 25.8 (27.7) | 19.9 (0.5) | 54.73% |
| | | kerSeg (stats.) | 3.2 (1.1) | 2.7 (5.2) | 20.3 (2.1) | 76.76% |
| 15 | 100 | CPDstergm | 0.9 (0.6) | 1.0 (0.0) | 11.1 (6.5) | **88.32**% |
| | | CPDrdpg | 1.1 (0.8) | 4.0 (5.6) | 15.5 (1.8) | 79.18% |
| | | CPDnbs | 1.6 (0.8) | 7.1 (3.6) | 10.8 (2.0) | 68.99% |
| | | CPDker | 0.1 (0.3) | 22.3 (7.5) | 10.3 (2.5) | 60.73% |
| | | gSeg (nets.) | 3.0 (0.0) | Inf (na) | Inf (na) | 0.00% |
| | | kerSeg (nets.) | 6.5 (1.6) | 2.7 (5.7) | 17.9 (2.0) | 47.57% |
| | | gSeg (stats.) | 4.0 (2.2) | 10.4 (15.8) | 20.0 (1.2) | 63.45% |
| | | kerSeg (stats.) | 4.3 (1.3) | 0.1 (0.3) | 20.6 (2.4) | 74.23% |
| 15 | 200 | CPDstergm | 0.1 (0.3) | 1.0 (0.0) | 2.1 (4.4) | **95.65**% |
| | | CPDrdpg | 1.7 (0.9) | 4.3 (4.7) | 15.6 (1.4) | 76.17% |
| | | CPDnbs | 1.9 (0.7) | 7.9 (2.8) | 11.8 (0.6) | 66.32% |
| | | CPDker | 0.1 (0.3) | 24.2 (7.5) | 10.3 (1.4) | 56.99% |
| | | gSeg (nets.) | 2.8 (0.4) | Inf (na) | Inf (na) | 8.91% |
| | | kerSeg (nets.) | 7.9 (1.9) | 3.0 (5.3) | 18.8 (1.7) | 41.86% |
| | | gSeg (stats.) | 4.2 (2.0) | 9.1 (16.5) | 18.9 (0.7) | 61.83% |
| | | kerSeg (stats.) | 4.5 (1.2) | 0.2 (0.6) | 20.0 (2.6) | 72.44% |

Table 9: Means (standard deviations) of evaluation metrics for dynamic networks simulated from the RDPGM with $d = 20$. The best coverage metric is bolded.

| $d$ | $n$ | Method | $|\hat{K} - K| \downarrow$ | $d(\hat{\mathcal{C}}|\mathcal{C}) \downarrow$ | $d(\mathcal{C}|\hat{\mathcal{C}}) \downarrow$ | $C(\mathcal{G}, \mathcal{G}') \uparrow$ |
|---|---|---|---|---|---|---|
| 20 | 50 | CPDstergm | 1.1 (0.7) | 5.6 (8.4) | 15.9 (2.1) | **79.76**% |
| | | CPDrdpg | 2.6 (1.0) | 1.5 (0.8) | 16.8 (1.3) | 73.88% |
| | | CPDnbs | 1.5 (0.8) | 8.4 (2.3) | 11.8 (0.6) | 66.97% |
| | | CPDker | 0.3 (0.5) | 16.2 (10.0) | 7.5 (3.7) | 71.19% |
| | | gSeg (nets.) | 2.9 (0.3) | Inf (na) | Inf (na) | 3.19% |
| | | kerSeg (nets.) | 5.3 (1.4) | 2.8 (4.5) | 16.7 (2.1) | 58.03% |
| | | gSeg (stats.) | 3.1 (1.8) | 21.8 (30.1) | 19.9 (0.5) | 58.70% |
| | | kerSeg (stats.) | 3.0 (1.2) | 0.4 (1.1) | 17.3 (3.1) | 79.34% |
| 20 | 100 | CPDstergm | 1.2 (0.7) | 0.9 (0.3) | 15.7 (4.4) | **87.37**% |
| | | CPDrdpg | 1.6 (0.9) | 3.0 (4.9) | 16.3 (1.0) | 78.92% |
| | | CPDnbs | 1.7 (0.8) | 8.3 (3.0) | 11.6 (1.2) | 66.22% |
| | | CPDker | 0.0 (0.0) | 19.3 (8.9) | 9.5 (2.0) | 64.06% |
| | | gSeg (nets.) | 2.9 (0.3) | Inf (na) | Inf (na) | 6.18% |
| | | kerSeg (nets.) | 7.5 (1.3) | 1.0 (2.6) | 18.3 (1.6) | 44.46% |
| | | gSeg (stats.) | 2.9 (2.8) | 26.4 (25.4) | 19.7 (0.6) | 50.98% |
| | | kerSeg (stats.) | 3.1 (1.1) | 0.0 (0.0) | 18.2 (3.1) | 81.13% |
| 20 | 200 | CPDstergm | 0.7 (0.5) | 1.0 (0.0) | 11.3 (6.9) | **89.83**% |
| | | CPDrdpg | 1.8 (0.4) | 1.2 (0.6) | 16.9 (1.0) | 80.81% |
| | | CPDnbs | 1.7 (0.7) | 8.3 (3.5) | 11.8 (0.4) | 67.29% |
| | | CPDker | 0.2 (0.4) | 24.7 (9.2) | 10.9 (2.0) | 54.76% |
| | | gSeg (nets.) | 2.8 (0.4) | Inf (na) | Inf (na) | 9.54% |
| | | kerSeg (nets.) | 8.9 (0.7) | 0.0 (0.0) | 18.3 (2.0) | 39.80% |
| | | gSeg (stats.) | 3.9 (2.3) | 16.8 (21.0) | 20.1 (1.3) | 54.09% |
| | | kerSeg (stats.) | 4.1 (0.9) | 0.0 (0.0) | 17.0 (3.0) | 73.33% |

Table 10: Time comparison in seconds across different methods and node sizes.

| | SBM ($\rho = 0$) | | STERGM ($p = 4$) | | RDPGM ($d = 20$) | |
|---|---|---|---|---|---|---|
| Method & Node | 50 | 100 | 50 | 100 | 50 | 100 |
| CPDstergm | 15.033 | 39.833 | 17.085 | 39.293 | 29.607 | 40.065 |
| CPDrdpg | 2.899 | 6.289 | 2.961 | 6.271 | 3.352 | 6.690 |
| CPDnbs | 0.308 | 1.124 | 0.310 | 1.078 | 0.338 | 1.436 |
| CPDker | 9.622 | 12.062 | 12.105 | 12.692 | 7.749 | 7.418 |
| gSeg (nets.) | 0.112 | 0.238 | 0.183 | 0.509 | 0.088 | 0.221 |
| kerSeg (nets.) | 3.971 | 4.303 | 4.707 | 4.922 | 5.957 | 7.877 |
| gSeg (stats.) | 2.698 | 7.244 | 2.321 | 6.071 | 1.276 | 1.960 |
| kerSeg (stats.) | 6.290 | 10.893 | 6.207 | 10.066 | 6.161 | 7.119 |

In addition, as the proposed CPDstergm method supports the usage of node level covariates, we define a nodal attribute $\boldsymbol{x}_i \in \{\text{High-Activity}, \text{Low-Activity}\}$ for each node $i \in [n]$, based on its cumulative activity over time. Specifically, we calculate the total degree of each node across the time span, and classify nodes as 'High-Activity' if their cumulative degree exceeds the median, and 'Low-Activity' otherwise. The four network statistics, edges, isolates, triangles, and homophily for activeness, are used in both formation and dissolution models of our method. Figure 5 displays the change magnitude $\Delta\hat{\boldsymbol{\zeta}}$ of Equation (17) and the detected change points from the proposed and competitor methods. Moreover, Table 11 provides a list of potential nearby events that align with the detected change points, and Figure 6 displays a raster plot of active edges over time, as a visual validation to the detected change points.

The two shaded areas in Figure 5 correspond to the winter and spring vacations in the MIT 2004-2005 academic calendar, and our method can punctually detect the pattern change in the contact behaviors. The competitor methods can also detect the beginning of the winter vacation, but their results on the spring vacation are deviated. Visually, the substantial reduction of interactions in Region I and the subtle shifts in Region II of Figure 6 support the change points detected by the proposed method, aligning with the

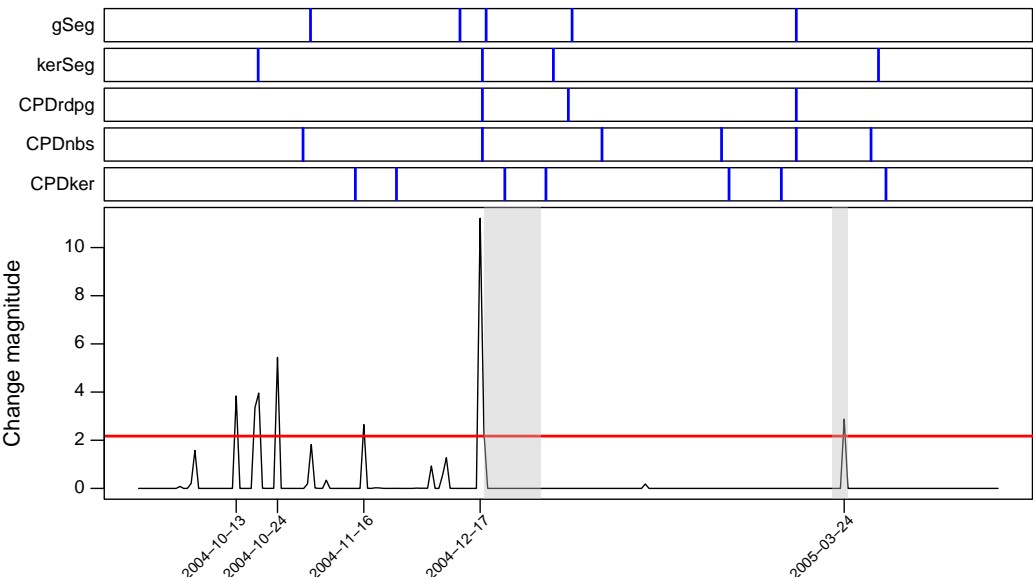

Figure 5: Visualization of the change magnitude $\Delta\hat{\boldsymbol{\zeta}}$ and the detected change points from our method for the MIT cellphone data. The detected change points from the competitor methods (blue bars) are also displayed. The two shaded areas correspond to the winter and spring vacations in the MIT 2004-2005 academic calendar. The data-driven threshold (red horizontal line) is calculated by (18) with $\mathcal{Z}_{0.975}$.

largest spike for the winter vacation (2004-12-17) and the smaller spike for the spring vacation (2005-03-24) in Figure 5. Furthermore, we detect a few spikes in the middle of October 2004, which correspond to the annual sponsor meeting that happened on 2004-10-21. About two-thirds of the participants have prepared and attended the annual sponsor meeting, and the majority of their time has contributed to achieve project goals throughout the week (Eagle & Pentland, 2006). The drastic shifts in Region III of Figure 6 also reveal the frequent changes in the network patterns during the sponsor meeting week.

Table 11: Potential nearby events that align with the detected change points (CP) of our method.

| Detected CP | Potential nearby events |
|---|---|
| 2004-10-13 | Preparation for the sponsor meeting |
| 2004-10-24 | 2004-10-21 (Sponsor meeting) |
| 2004-11-16 | 2004-11-17 (Last day to cancel subjects) |
| 2004-12-17 | 2004-12-18 to 2005-01-02 (Winter vacation) |
| 2005-03-24 | 2005-03-21 to 2005-03-25 (Spring vacation) |

## 5.3 Stock Market Data

The stock market data consists of the weekly log returns of 29 stocks included in the Dow Jones Industrial Average (DJIA) index, and it is available in the R package `ecp` (James & Matteson, 2015). We consider the data from 2007-01-01 to 2010-01-04, which covers the 2008 worldwide economic crisis. Moreover, we focus on using the negative correlations among stock returns to detect the systematic anomalies in the financial market. First, we use a sliding window of width 4 to calculate the correlation matrices of the weekly log returns. We then truncate the correlation matrices by setting those entries which have negative values as 1, and the remaining as 0. In the $T = 158$ constructed networks, a connected edge $\boldsymbol{y}_{ij}^{t} = 1$ indicates the log returns of stocks $i$ and $j$ are negatively correlated over the four-week period that ends at week $t$.

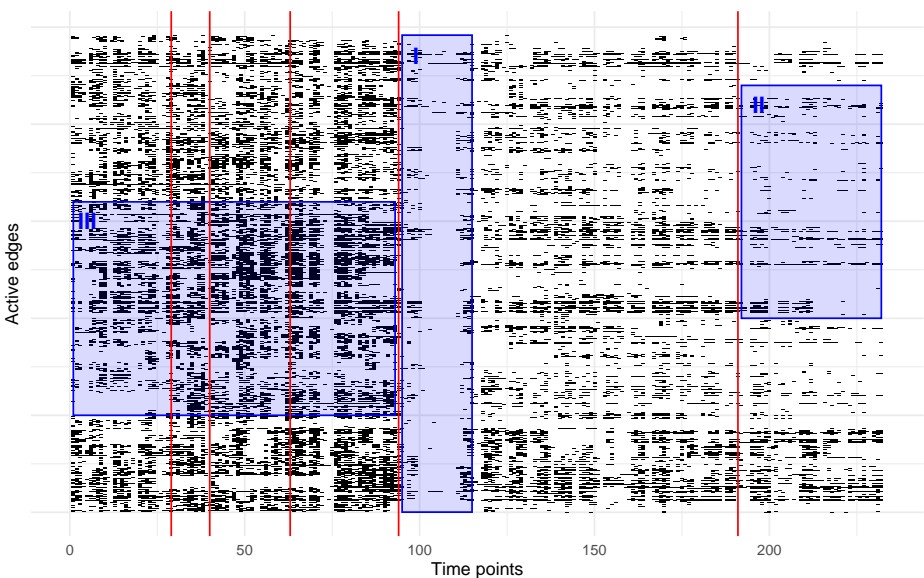

Figure 6: Raster plot of the MIT cellphone data. Each black tile represents an active edge between a pair of nodes at a given time point. The red vertical lines indicate change points detected by the proposed method, and the blue shaded regions highlight notable changes in network interaction patterns.

As a network statistic, the number of triangles signifies the volatility of the stock market, since the three stocks are mutually negatively correlated. In other words, the more triangles in a network, the more opposite movements among the stock returns, suggesting a large fluctuation in the financial market. On the contrary, when the number of triangles is low, the majority of the stock returns either increase or decrease at the same time, suggesting a stable trend in the market. In addition, we define a nodal attribute $\boldsymbol{x}_i \in \{\text{Hedging-Prone}, \text{Market-Following}\}$ for each stock $i \in [n]$, based on it cumulative degree over time. Specifically, stocks with total degree above the median are labeled as 'Hedging-Prone', reflecting defensive behavior relative to the market, while the remaining stocks are labeled as 'Market-Following', indicating synchronized movement with the broader market. For the proposed method, we use three network statistics, edges, triangles, and homophily for risk orientation, in both formation and dissolution models. For the competitor methods, we use dynamic networks as input data, since they produce more reasonable results than using the network statistics. For CPDker, we use network statistics as input. Figure 7 displays the change magnitude $\Delta \hat{\boldsymbol{\zeta}}$ of Equation (17) and the detected change points from the proposed and competitor methods. Table 12 provides a list of potential nearby events that align with the detected change points, and Figure 8 displays a raster plot of active edges over time, as visual validation to the detected change points.

As expected, the stock market is highly volatile between 2007 and 2009. The competitors have detected excessive number of change points, aligning with the smaller spikes in $\Delta \hat{\boldsymbol{\zeta}}$ of Figure 7. Those change points could be detected by our method, if we manually lower the threshold to adjust the sensitivity. Since the networks are constructed using a sliding window, a detected change point indicates the pattern change occurs amid the four-week time horizon. As supporting evidence to the three detected change points in Table 12, the New Century Financial Corporation was the largest U.S. subprime mortgage lender in 2007, and the Lehman Brothers was one of the largest investment banks. Their bankruptcies caused by the collapse of the mortgage industry severely fueled the worldwide financial crisis, which also led the Dow Jones Industrial Average to the bottom. In Figure 8, Region I shows a substantial increase in edge density following the collapse of New Century Financial Corporation, indicating heightened volatility in the stock market. As fewer edges suggest stable trend, Region II captures the market downturn following the bankruptcy of Lehman Brothers. Finally, Region III shows a rebound in market activity, as edge density rises after the Dow Jones Industrial Average reaches its lowest point, suggesting the beginning of a recovery phase.

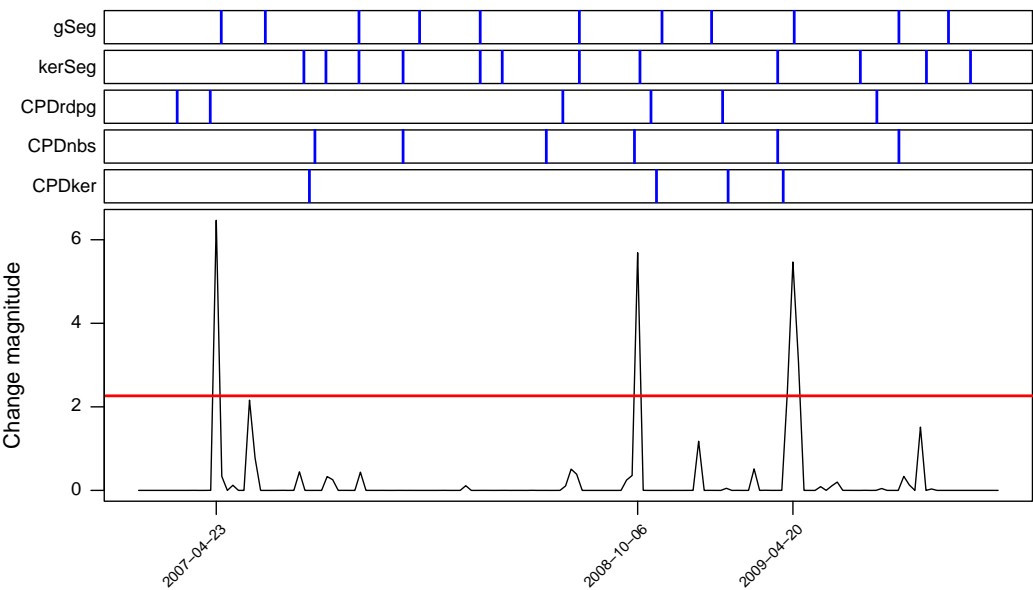

Figure 7: Visualization of the change magnitude $\Delta\hat{\boldsymbol{\zeta}}$ and the detected change points from our method for the stock market data. The detected change points from the competitor methods (blue bars) are also displayed. The data-driven threshold (red horizontal line) is calculated by (18) with $\mathcal{Z}_{0.975}$.

Table 12: Potential nearby events that align with the detected change points (CP) of our method.

| Detected CP | Potential nearby events |
|---|---|
| 2007-04-23 | 2007-04-02 (New Century Financial Corporation filed for bankruptcy) |
| 2008-10-06 | 2008-09-15 (Lehman Brothers filed for bankruptcy) |
| 2009-04-20 | 2009-03-09 (Dow Jones Industrial Average bottomed) |

## 6 Discussion

In this work, we study the change point detection problem in time series of networks, which serves as a prerequisite for dynamic network analysis. Essentially, we fit a time-heterogeneous STERGM while penalizing the sum of Euclidean norms of the parameter differences between consecutive time steps. The objective function with Group Fused Lasso penalty is solved via Alternating Direction Method of Multipliers, and we adopt the pseudo-likelihood of STERGM to expedite parameter estimation.

The STERGM (Krivitsky & Handcock, 2014) used in our method is a flexible model to fit dynamic networks with both dyadic and temporal dependence. It manages dyad formation and dissolution separately, as the underlying reasons that induce the two processes are usually different in reality. Furthermore, the ERGM suite (Handcock et al., 2022) provides an extensive list of network statistics to capture the structural changes, and we develop an R package `CPDstergm` to implement the proposed method.

Several improvements to our change point detection framework are possible. Relational phenomena by nature have degrees of strength, and dichotomizing valued networks into binary networks may introduce biases for analysis (Thomas & Blitzstein, 2011). To this end, we can extend the STERGM with a valued ERGM (Krivitsky, 2012; Desmarais & Cranmer, 2012a; Caimo & Gollini, 2020) to facilitate change point detection in dynamic valued networks. Moreover, the number of participants and their attributes are subject to change over time. It is necessary for a change point detection method to adjust the network sizes as in Krivitsky et al. (2011), and to adapt the time-evolving nodal attributes by incorporating the Exponential-

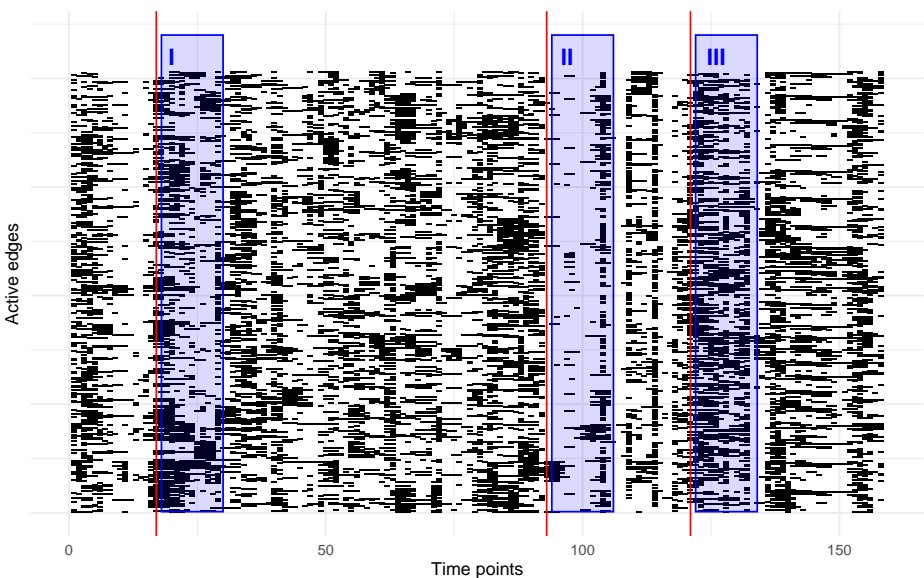

Figure 8: Raster plot of the stock market data. Each black tile represents an active edge between a pair of nodes at a given time point. The red vertical lines indicate change points detected by the proposed method, and the blue shaded regions highlight notable changes in network interaction patterns.

family Random Network Model (ERNM) as in Fellows & Handcock (2012) and Fellows & Handcock (2013). In addition, the proposed framework can be extended to detect change points in dynamic multilayer networks (Sohn & Park, 2017; Wang et al., 2025), where different types of interactions for a fixed set of nodes are observed simultaneously. Lastly, recent advances in contrastive learning (Ermshaus et al., 2023; Puchkin & Shcherbakova, 2023) present a novel and promising direction for developing adaptive and data-driven approaches to change point detection in dynamic networks.

## Code and Data Availability

The R package `CPDstergm` is available at `https://github.com/allenkei/CPDstergm`. The code to reproduce the results in the manuscript using the R package is available at `https://github.com/allenkei/CPDstergm_demo`. The MIT cellphone data is available in the R package `CPDstergm`, and the stock market data is available in the R package `ecp`.

## Acknowledgments

We thank Mark S. Handcock for helpful comments on this work. Also, we are extremely grateful for the constructive comments provided by the anonymous reviewers and Action Editors. Yik Lun Kei and Oscar Hernan Madrid Padilla were partially funded by NSF DMS-2015489.

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

## A  ADMM Convergence

In this section, we provide the proof for Proposition 1. Consider the constrained optimization problem

$$\hat{\boldsymbol{\theta}} = \arg\min_{\boldsymbol{\theta}} -l(\boldsymbol{\theta}) + \lambda \sum_{i=1}^{\tau-1} \frac{\|\boldsymbol{z}_{i+1,\cdot} - \boldsymbol{z}_{i,\cdot}\|_2}{\boldsymbol{d}_i}$$
$$\text{subject to } \boldsymbol{\theta} - \boldsymbol{z} = \boldsymbol{0}.$$

The Lagrangian can be expressed as

$$\mathcal{L}(\boldsymbol{\theta}, \boldsymbol{z}, \boldsymbol{\rho}) = f(\boldsymbol{\theta}) + g(\boldsymbol{z}) + \text{tr}[\boldsymbol{\rho}^\top(\boldsymbol{\theta} - \boldsymbol{z})] \tag{19}$$

and the augmented Lagrangian can be expressed as

$$\mathcal{L}_\alpha(\boldsymbol{\theta}, \boldsymbol{z}, \boldsymbol{\rho}) = f(\boldsymbol{\theta}) + g(\boldsymbol{z}) + \text{tr}[\boldsymbol{\rho}^\top(\boldsymbol{\theta} - \boldsymbol{z})] + \frac{\alpha}{2}\|\boldsymbol{\theta} - \boldsymbol{z}\|_F^2 \tag{20}$$

where

$$f(\boldsymbol{\theta}) = -l(\boldsymbol{\theta}) \geq 0, \quad g(\boldsymbol{z}) = \lambda \sum_{i=1}^{\tau-1} \frac{\|\boldsymbol{z}_{i+1,\cdot} - \boldsymbol{z}_{i,\cdot}\|_2}{\boldsymbol{d}_i} \geq 0,$$

and $\boldsymbol{\rho} \in \mathbb{R}^{\tau \times p}$ is the Lagrange multiplier.

Let $(\boldsymbol{\theta}^*, \boldsymbol{z}^*, \boldsymbol{\rho}^*)$ be the optimal point of the Lagrangian $\mathcal{L}(\boldsymbol{\theta}, \boldsymbol{z}, \boldsymbol{\rho})$ in Equation (19). Also, let

$$p^k = f(\boldsymbol{\theta}^k) + g(\boldsymbol{z}^k)$$

where $k$ is the current ADMM iteration. Denote $\boldsymbol{r}^{k+1} = \boldsymbol{\theta}^{k+1} - \boldsymbol{z}^{k+1} \in \mathbb{R}^{\tau \times p}$ as the primal residual in the matrix form. Since $\mathcal{L}(\boldsymbol{\theta}^{k+1}, \boldsymbol{z}^{k+1}, \boldsymbol{\rho}^*) = p^{k+1} + \text{tr}[(\boldsymbol{\rho}^*)^\top \boldsymbol{r}^{k+1}]$, and $\mathcal{L}(\boldsymbol{\theta}^*, \boldsymbol{z}^*, \boldsymbol{\rho}^*) \leq \mathcal{L}(\boldsymbol{\theta}^{k+1}, \boldsymbol{z}^{k+1}, \boldsymbol{\rho}^*)$, we have

$$p^* \leq p^{k+1} + \text{tr}[(\boldsymbol{\rho}^*)^\top \boldsymbol{r}^{k+1}]$$
$$p^* - p^{k+1} \leq \text{tr}[(\boldsymbol{\rho}^*)^\top \boldsymbol{r}^{k+1}]. \tag{21}$$

Note that the dual update of ADMM procedure for the optimization problem in (20) is $\boldsymbol{\rho}^{k+1} = \boldsymbol{\rho}^k + \alpha \boldsymbol{r}^{k+1}$, so $\boldsymbol{\rho}^k = \boldsymbol{\rho}^{k+1} - \alpha \boldsymbol{r}^{k+1}$. Since $\boldsymbol{\theta}^{k+1} = \arg\min_{\boldsymbol{\theta}} \mathcal{L}_\alpha(\boldsymbol{\theta}, \boldsymbol{z}^k, \boldsymbol{\rho}^k)$, the optimal condition gives

$$\boldsymbol{0} \in \partial f(\boldsymbol{\theta}^{k+1}) + \boldsymbol{\rho}^k + \alpha(\boldsymbol{\theta}^{k+1} - \boldsymbol{z}^k). \tag{22}$$

Substituting the $\boldsymbol{\rho}^k$ in (22) with $\boldsymbol{\rho}^{k+1} - \alpha \boldsymbol{r}^{k+1}$, and then adding and subtracting $\alpha \boldsymbol{z}^{k+1}$ to the right hand side of (22), we have

$$\boldsymbol{0} \in \partial f(\boldsymbol{\theta}^{k+1}) + (\boldsymbol{\rho}^{k+1} - \alpha \boldsymbol{r}^{k+1}) + \alpha(\boldsymbol{\theta}^{k+1} - \boldsymbol{z}^k) + \alpha \boldsymbol{z}^{k+1} - \alpha \boldsymbol{z}^{k+1}$$
$$\boldsymbol{0} \in \partial f(\boldsymbol{\theta}^{k+1}) + \boldsymbol{\rho}^{k+1} - \alpha \boldsymbol{r}^{k+1} + \alpha\big(\boldsymbol{\theta}^{k+1} - \boldsymbol{z}^{k+1} + (\boldsymbol{z}^{k+1} - \boldsymbol{z}^k)\big)$$
$$\boldsymbol{0} \in \partial f(\boldsymbol{\theta}^{k+1}) + \boldsymbol{\rho}^{k+1} - \alpha \boldsymbol{r}^{k+1} + \alpha\big(\boldsymbol{r}^{k+1} + (\boldsymbol{z}^{k+1} - \boldsymbol{z}^k)\big)$$
$$\boldsymbol{0} \in \partial f(\boldsymbol{\theta}^{k+1}) + \boldsymbol{\rho}^{k+1} + \alpha(\boldsymbol{z}^{k+1} - \boldsymbol{z}^k).$$

This implies that $\boldsymbol{\theta}^{k+1}$ minimizes the objective function

$$f(\boldsymbol{\theta}) + \operatorname{tr}\big[\big(\boldsymbol{\rho}^{k+1} + \alpha(\boldsymbol{z}^{k+1} - \boldsymbol{z}^k)\big)^\top \boldsymbol{\theta}\big]$$

and

$$f(\boldsymbol{\theta}^{k+1}) + \operatorname{tr}\big[\big(\boldsymbol{\rho}^{k+1} + \alpha(\boldsymbol{z}^{k+1} - \boldsymbol{z}^k)\big)^\top \boldsymbol{\theta}^{k+1}\big] \le f(\boldsymbol{\theta}^*) + \operatorname{tr}\big[\big(\boldsymbol{\rho}^{k+1} + \alpha(\boldsymbol{z}^{k+1} - \boldsymbol{z}^k)\big)^\top \boldsymbol{\theta}^*\big]. \tag{23}$$

Similarly, since $\boldsymbol{z}^{k+1} = \arg\min_{\boldsymbol{z}} \mathcal{L}_\alpha(\boldsymbol{\theta}^{k+1}, \boldsymbol{z}, \boldsymbol{\rho}^k)$, the optimal condition gives

$$\boldsymbol{0} \in \partial g(\boldsymbol{z}^{k+1}) - \boldsymbol{\rho}^k - \alpha(\boldsymbol{\theta}^{k+1} - \boldsymbol{z}^{k+1}). \tag{24}$$

Substituting the $\boldsymbol{\rho}^k$ in (24) with $\boldsymbol{\rho}^{k+1} - \alpha \boldsymbol{r}^{k+1}$, we have

$$\boldsymbol{0} \in \partial g(\boldsymbol{z}^{k+1}) - (\boldsymbol{\rho}^{k+1} - \alpha \boldsymbol{r}^{k+1}) - \alpha \boldsymbol{r}^{k+1}$$
$$\boldsymbol{0} \in \partial g(\boldsymbol{z}^{k+1}) - \boldsymbol{\rho}^{k+1}.$$

This implies that $\boldsymbol{z}^{k+1}$ minimizes the objective function

$$g(\boldsymbol{z}) - \operatorname{tr}[(\boldsymbol{\rho}^{k+1})^\top \boldsymbol{z}]$$

and

$$g(\boldsymbol{z}^{k+1}) - \operatorname{tr}[(\boldsymbol{\rho}^{k+1})^\top \boldsymbol{z}^{k+1}] \le g(\boldsymbol{z}^*) - \operatorname{tr}[(\boldsymbol{\rho}^{k+1})^\top \boldsymbol{z}^*]. \tag{25}$$

Adding the two Inequalities (23) and (25) together, we have

$$f(\boldsymbol{\theta}^{k+1}) + g(\boldsymbol{z}^{k+1}) - f(\boldsymbol{\theta}^*) - g(\boldsymbol{z}^*) \le \operatorname{tr}\big[\big(\boldsymbol{\rho}^{k+1} + \alpha(\boldsymbol{z}^{k+1} - \boldsymbol{z}^k)\big)^\top (\boldsymbol{\theta}^* - \boldsymbol{\theta}^{k+1})\big] + \operatorname{tr}[(\boldsymbol{\rho}^{k+1})^\top (\boldsymbol{z}^{k+1} - \boldsymbol{z}^*)]$$
$$p^{k+1} - p^* \le \operatorname{tr}\big[\big(\boldsymbol{\rho}^{k+1} + \alpha(\boldsymbol{z}^{k+1} - \boldsymbol{z}^k)\big)^\top (\boldsymbol{\theta}^* - \boldsymbol{\theta}^{k+1})\big] + \operatorname{tr}[(\boldsymbol{\rho}^{k+1})^\top (\boldsymbol{z}^{k+1} - \boldsymbol{z}^*)]. \tag{26}$$

Separating and grouping the terms on the right hand side in (26), we have

$$p^{k+1} - p^* \le \operatorname{tr}\big[(\boldsymbol{\rho}^{k+1})^\top (\boldsymbol{\theta}^* - \boldsymbol{\theta}^{k+1})\big] + \operatorname{tr}\big[\alpha(\boldsymbol{z}^{k+1} - \boldsymbol{z}^k)^\top (\boldsymbol{\theta}^* - \boldsymbol{\theta}^{k+1})\big] + \operatorname{tr}[(\boldsymbol{\rho}^{k+1})^\top (\boldsymbol{z}^{k+1} - \boldsymbol{z}^*)]$$
$$\le \operatorname{tr}\big[(\boldsymbol{\rho}^{k+1})^\top \big(\boldsymbol{\theta}^* - \boldsymbol{z}^* - (\boldsymbol{\theta}^{k+1} - \boldsymbol{z}^{k+1})\big)\big] + \operatorname{tr}\big[\alpha(\boldsymbol{z}^{k+1} - \boldsymbol{z}^k)^\top (\boldsymbol{\theta}^* - \boldsymbol{\theta}^{k+1})\big]$$
$$\le -\operatorname{tr}\big[(\boldsymbol{\rho}^{k+1})^\top \boldsymbol{r}^{k+1}\big] + \operatorname{tr}\big[\alpha(\boldsymbol{z}^{k+1} - \boldsymbol{z}^k)^\top (\boldsymbol{\theta}^* - \boldsymbol{\theta}^{k+1})\big] \tag{27}$$

where $\boldsymbol{\theta}^* - \boldsymbol{z}^* = \boldsymbol{0}$ and $\boldsymbol{\theta}^{k+1} - \boldsymbol{z}^{k+1} = \boldsymbol{r}^{k+1}$. Moreover, note that

$$\boldsymbol{r}^{k+1} = \boldsymbol{\theta}^{k+1} - \boldsymbol{z}^{k+1} - (\boldsymbol{\theta}^* - \boldsymbol{z}^*)$$
$$\boldsymbol{r}^{k+1} = (\boldsymbol{\theta}^{k+1} - \boldsymbol{\theta}^*) - (\boldsymbol{z}^{k+1} - \boldsymbol{z}^*)$$
$$-\boldsymbol{r}^{k+1} = (\boldsymbol{\theta}^* - \boldsymbol{\theta}^{k+1}) + (\boldsymbol{z}^{k+1} - \boldsymbol{z}^*)$$
$$(\boldsymbol{\theta}^* - \boldsymbol{\theta}^{k+1}) = -\boldsymbol{r}^{k+1} - (\boldsymbol{z}^{k+1} - \boldsymbol{z}^*).$$

Substituting the term $(\boldsymbol{\theta}^* - \boldsymbol{\theta}^{k+1})$ in Inequality (27), we have

$$p^{k+1} - p^* \le -\operatorname{tr}\big[(\boldsymbol{\rho}^{k+1})^\top \boldsymbol{r}^{k+1}\big] + \operatorname{tr}\big[\alpha(\boldsymbol{z}^{k+1} - \boldsymbol{z}^k)^\top \big(-\boldsymbol{r}^{k+1} - (\boldsymbol{z}^{k+1} - \boldsymbol{z}^*)\big)\big]$$
$$\le -\operatorname{tr}\big[(\boldsymbol{\rho}^{k+1})^\top \boldsymbol{r}^{k+1}\big] - \operatorname{tr}\big[\alpha(\boldsymbol{z}^{k+1} - \boldsymbol{z}^k)^\top \boldsymbol{r}^{k+1}\big] - \operatorname{tr}\big[\alpha(\boldsymbol{z}^{k+1} - \boldsymbol{z}^k)^\top (\boldsymbol{z}^{k+1} - \boldsymbol{z}^*)\big]. \tag{28}$$

Then adding Inequalities (21) and (28), we have

$$0 \le \operatorname{tr}[(\boldsymbol{\rho}^*)^\top \boldsymbol{r}^{k+1}] - \operatorname{tr}\big[(\boldsymbol{\rho}^{k+1})^\top \boldsymbol{r}^{k+1}\big] - \operatorname{tr}\big[\alpha(\boldsymbol{z}^{k+1} - \boldsymbol{z}^k)^\top \boldsymbol{r}^{k+1}\big] - \operatorname{tr}\big[\alpha(\boldsymbol{z}^{k+1} - \boldsymbol{z}^k)^\top (\boldsymbol{z}^{k+1} - \boldsymbol{z}^*)\big]$$
$$0 \le \operatorname{tr}[(\boldsymbol{\rho}^* - \boldsymbol{\rho}^{k+1})^\top \boldsymbol{r}^{k+1}] - \operatorname{tr}\big[\alpha(\boldsymbol{z}^{k+1} - \boldsymbol{z}^k)^\top \boldsymbol{r}^{k+1}\big] - \operatorname{tr}\big[\alpha(\boldsymbol{z}^{k+1} - \boldsymbol{z}^k)^\top (\boldsymbol{z}^{k+1} - \boldsymbol{z}^*)\big]$$
$$0 \ge 2\operatorname{tr}[(\boldsymbol{\rho}^{k+1} - \boldsymbol{\rho}^*)^\top \boldsymbol{r}^{k+1}] + 2\operatorname{tr}\big[\alpha(\boldsymbol{z}^{k+1} - \boldsymbol{z}^k)^\top \boldsymbol{r}^{k+1}\big] + 2\operatorname{tr}\big[\alpha(\boldsymbol{z}^{k+1} - \boldsymbol{z}^k)^\top (\boldsymbol{z}^{k+1} - \boldsymbol{z}^*)\big] \tag{29}$$

where (29) followed as we multiply the inequality by two and change the sign on both sides.

Now we consider the expansion of the first term on the right hand side in Inequality (29). Substituting the $\boldsymbol{\rho}^{k+1}$ with the dual update $\boldsymbol{\rho}^{k+1} = \boldsymbol{\rho}^k + \alpha \boldsymbol{r}^{k+1}$, we have

$$2\mathrm{tr}[(\boldsymbol{\rho}^{k+1} - \boldsymbol{\rho}^*)^\top \boldsymbol{r}^{k+1}] \tag{30}$$
$$= 2\mathrm{tr}[(\boldsymbol{\rho}^k + \alpha \boldsymbol{r}^{k+1} - \boldsymbol{\rho}^*)^\top \boldsymbol{r}^{k+1}]$$
$$= 2\mathrm{tr}[(\boldsymbol{\rho}^k - \boldsymbol{\rho}^*)^\top \boldsymbol{r}^{k+1}] + 2\mathrm{tr}[\alpha (\boldsymbol{r}^{k+1})^\top (\boldsymbol{r}^{k+1})]$$
$$= 2\mathrm{tr}[(\boldsymbol{\rho}^k - \boldsymbol{\rho}^*)^\top \boldsymbol{r}^{k+1}] + \alpha \|\boldsymbol{r}^{k+1}\|_F^2 + \alpha \|\boldsymbol{r}^{k+1}\|_F^2. \tag{31}$$

Since $\boldsymbol{\rho}^{k+1} = \boldsymbol{\rho}^k + \alpha \boldsymbol{r}^{k+1}$, we also have $\boldsymbol{r}^{k+1} = \frac{1}{\alpha}(\boldsymbol{\rho}^{k+1} - \boldsymbol{\rho}^k)$. Substituting the $\boldsymbol{r}^{k+1}$ in the first two terms of (31) and expanding the matrix multiplications, the expression in (30) proceeds as

$$2\mathrm{tr}[(\boldsymbol{\rho}^{k+1} - \boldsymbol{\rho}^*)^\top \boldsymbol{r}^{k+1}]$$
$$= \frac{1}{\alpha}\mathrm{tr}[2(\boldsymbol{\rho}^k - \boldsymbol{\rho}^*)^\top (\boldsymbol{\rho}^{k+1} - \boldsymbol{\rho}^k)] + \alpha \frac{1}{\alpha^2}\mathrm{tr}[(\boldsymbol{\rho}^{k+1} - \boldsymbol{\rho}^k)^\top (\boldsymbol{\rho}^{k+1} - \boldsymbol{\rho}^k)] + \alpha \|\boldsymbol{r}^{k+1}\|_F^2$$
$$= \frac{1}{\alpha}\mathrm{tr}[(\boldsymbol{\rho}^{k+1})^\top \boldsymbol{\rho}^{k+1} - 2(\boldsymbol{\rho}^*)^\top \boldsymbol{\rho}^{k+1} + 2(\boldsymbol{\rho}^*)^\top \boldsymbol{\rho}^k - (\boldsymbol{\rho}^k)^\top \boldsymbol{\rho}^k] + \alpha \|\boldsymbol{r}^{k+1}\|_F^2$$
$$= \frac{1}{\alpha}\mathrm{tr}[(\boldsymbol{\rho}^{k+1})^\top \boldsymbol{\rho}^{k+1} - 2(\boldsymbol{\rho}^*)^\top \boldsymbol{\rho}^{k+1} + 2(\boldsymbol{\rho}^*)^\top \boldsymbol{\rho}^k - (\boldsymbol{\rho}^k)^\top \boldsymbol{\rho}^k + (\boldsymbol{\rho}^*)^\top \boldsymbol{\rho}^* - (\boldsymbol{\rho}^*)^\top \boldsymbol{\rho}^*] + \alpha \|\boldsymbol{r}^{k+1}\|_F^2$$
$$= \frac{1}{\alpha}\mathrm{tr}\Big\{ \big[(\boldsymbol{\rho}^{k+1})^\top \boldsymbol{\rho}^{k+1} - 2(\boldsymbol{\rho}^*)^\top \boldsymbol{\rho}^{k+1} + (\boldsymbol{\rho}^*)^\top \boldsymbol{\rho}^*\big] - \big[(\boldsymbol{\rho}^k)^\top \boldsymbol{\rho}^k - 2(\boldsymbol{\rho}^*)^\top \boldsymbol{\rho}^k + (\boldsymbol{\rho}^*)^\top \boldsymbol{\rho}^*\big] \Big\} + \alpha \|\boldsymbol{r}^{k+1}\|_F^2$$
$$= \frac{1}{\alpha}\big( \|\boldsymbol{\rho}^{k+1} - \boldsymbol{\rho}^*\|_F^2 - \|\boldsymbol{\rho}^k - \boldsymbol{\rho}^*\|_F^2 \big) + \alpha \|\boldsymbol{r}^{k+1}\|_F^2. \tag{32}$$

Next we consider the expansion of the second and third terms on the right hand side in Inequality (29), with an additional term $\alpha \|\boldsymbol{r}^{k+1}\|_F^2$. By adding and subtracting $\alpha \|\boldsymbol{z}^{k+1} - \boldsymbol{z}^k\|_F^2$, we have

$$\alpha \|\boldsymbol{r}^{k+1}\|_F^2 + 2\mathrm{tr}\big[\alpha (\boldsymbol{z}^{k+1} - \boldsymbol{z}^k)^\top \boldsymbol{r}^{k+1}\big] + 2\mathrm{tr}\big[\alpha (\boldsymbol{z}^{k+1} - \boldsymbol{z}^k)^\top (\boldsymbol{z}^{k+1} - \boldsymbol{z}^*)\big] \tag{33}$$
$$= \alpha \|\boldsymbol{r}^{k+1}\|_F^2 + \alpha \mathrm{tr}\big[2(\boldsymbol{z}^{k+1} - \boldsymbol{z}^k)^\top \boldsymbol{r}^{k+1}\big] + 2\mathrm{tr}\big[\alpha (\boldsymbol{z}^{k+1} - \boldsymbol{z}^k)^\top (\boldsymbol{z}^{k+1} - \boldsymbol{z}^*)\big] + \alpha \|\boldsymbol{z}^{k+1} - \boldsymbol{z}^k\|_F^2 - \alpha \|\boldsymbol{z}^{k+1} - \boldsymbol{z}^k\|_F^2$$
$$= \alpha \Big\{ \|\boldsymbol{r}^{k+1}\|_F^2 + \mathrm{tr}\big[2(\boldsymbol{z}^{k+1} - \boldsymbol{z}^k)^\top \boldsymbol{r}^{k+1}\big] + \|\boldsymbol{z}^{k+1} - \boldsymbol{z}^k\|_F^2 \Big\} - \alpha \|\boldsymbol{z}^{k+1} - \boldsymbol{z}^k\|_F^2 + 2\mathrm{tr}\big[\alpha (\boldsymbol{z}^{k+1} - \boldsymbol{z}^k)^\top (\boldsymbol{z}^{k+1} - \boldsymbol{z}^*)\big]$$
$$= \alpha \|\boldsymbol{r}^{k+1} + (\boldsymbol{z}^{k+1} - \boldsymbol{z}^k)\|_F^2 - \alpha \|\boldsymbol{z}^{k+1} - \boldsymbol{z}^k\|_F^2 + 2\mathrm{tr}\big[\alpha (\boldsymbol{z}^{k+1} - \boldsymbol{z}^k)^\top (\boldsymbol{z}^{k+1} - \boldsymbol{z}^*)\big] \tag{34}$$

Since $\boldsymbol{z}^{k+1} - \boldsymbol{z}^* = (\boldsymbol{z}^{k+1} - \boldsymbol{z}^k) + (\boldsymbol{z}^k - \boldsymbol{z}^*)$, we substitute the $\boldsymbol{z}^{k+1} - \boldsymbol{z}^*$ in the third term of (34) so that (33) proceeds as

$$\alpha \|\boldsymbol{r}^{k+1}\|_F^2 + 2\mathrm{tr}\big[\alpha (\boldsymbol{z}^{k+1} - \boldsymbol{z}^k)^\top \boldsymbol{r}^{k+1}\big] + 2\mathrm{tr}\big[\alpha (\boldsymbol{z}^{k+1} - \boldsymbol{z}^k)^\top (\boldsymbol{z}^{k+1} - \boldsymbol{z}^*)\big]$$
$$= \alpha \|\boldsymbol{r}^{k+1} + (\boldsymbol{z}^{k+1} - \boldsymbol{z}^k)\|_F^2 - \alpha \|\boldsymbol{z}^{k+1} - \boldsymbol{z}^k\|_F^2 + 2\mathrm{tr}\Big\{ \alpha (\boldsymbol{z}^{k+1} - \boldsymbol{z}^k)^\top [(\boldsymbol{z}^{k+1} - \boldsymbol{z}^k) + (\boldsymbol{z}^k - \boldsymbol{z}^*)] \Big\}$$
$$= \alpha \|\boldsymbol{r}^{k+1} + (\boldsymbol{z}^{k+1} - \boldsymbol{z}^k)\|_F^2 - \alpha \|\boldsymbol{z}^{k+1} - \boldsymbol{z}^k\|_F^2 + \alpha \mathrm{tr}\Big\{ (\boldsymbol{z}^{k+1} - \boldsymbol{z}^k)^\top [2(\boldsymbol{z}^{k+1} - \boldsymbol{z}^k) + 2(\boldsymbol{z}^k - \boldsymbol{z}^*)] \Big\}$$
$$= \alpha \|\boldsymbol{r}^{k+1} + (\boldsymbol{z}^{k+1} - \boldsymbol{z}^k)\|_F^2 + \alpha \mathrm{tr}\Big\{ (\boldsymbol{z}^{k+1} - \boldsymbol{z}^k)^\top [(\boldsymbol{z}^{k+1} - \boldsymbol{z}^k) + 2(\boldsymbol{z}^k - \boldsymbol{z}^*)] \Big\}. \tag{35}$$

Since $\boldsymbol{z}^{k+1} - \boldsymbol{z}^k = (\boldsymbol{z}^{k+1} - \boldsymbol{z}^*) - (\boldsymbol{z}^k - \boldsymbol{z}^*)$, we sequentially substitute the $\boldsymbol{z}^{k+1} - \boldsymbol{z}^k$ in (35) so that (33) proceeds as

$$\alpha \|\boldsymbol{r}^{k+1}\|_F^2 + 2\mathrm{tr}\big[\alpha (\boldsymbol{z}^{k+1} - \boldsymbol{z}^k)^\top \boldsymbol{r}^{k+1}\big] + 2\mathrm{tr}\big[\alpha (\boldsymbol{z}^{k+1} - \boldsymbol{z}^k)^\top (\boldsymbol{z}^{k+1} - \boldsymbol{z}^*)\big]$$
$$= \alpha \|\boldsymbol{r}^{k+1} + (\boldsymbol{z}^{k+1} - \boldsymbol{z}^k)\|_F^2 + \alpha \mathrm{tr}\Big\{ (\boldsymbol{z}^{k+1} - \boldsymbol{z}^k)^\top [(\boldsymbol{z}^{k+1} - \boldsymbol{z}^*) - (\boldsymbol{z}^k - \boldsymbol{z}^*) + 2(\boldsymbol{z}^k - \boldsymbol{z}^*)] \Big\}$$
$$= \alpha \|\boldsymbol{r}^{k+1} + (\boldsymbol{z}^{k+1} - \boldsymbol{z}^k)\|_F^2 + \alpha \mathrm{tr}\Big\{ (\boldsymbol{z}^{k+1} - \boldsymbol{z}^k)^\top [(\boldsymbol{z}^{k+1} - \boldsymbol{z}^*) + (\boldsymbol{z}^k - \boldsymbol{z}^*)] \Big\}$$
$$= \alpha \|\boldsymbol{r}^{k+1} + (\boldsymbol{z}^{k+1} - \boldsymbol{z}^k)\|_F^2 + \alpha \mathrm{tr}\Big\{ [(\boldsymbol{z}^{k+1} - \boldsymbol{z}^*) - (\boldsymbol{z}^k - \boldsymbol{z}^*)]^\top [(\boldsymbol{z}^{k+1} - \boldsymbol{z}^*) + (\boldsymbol{z}^k - \boldsymbol{z}^*)] \Big\}$$
$$= \alpha \|\boldsymbol{r}^{k+1} + (\boldsymbol{z}^{k+1} - \boldsymbol{z}^k)\|_F^2 + \alpha \big[ \|\boldsymbol{z}^{k+1} - \boldsymbol{z}^*\|_F^2 - \|\boldsymbol{z}^k - \boldsymbol{z}^*\|_F^2 \big]. \tag{36}$$

Combining the results from (32) and (36) to substitute the corresponding terms on the right hand side of Inequality (29), we have

$$2\text{tr}[(\boldsymbol{\rho}^{k+1} - \boldsymbol{\rho}^*)^\top \boldsymbol{r}^{k+1}] + 2\text{tr}[\alpha(\boldsymbol{z}^{k+1} - \boldsymbol{z}^k)^\top \boldsymbol{r}^{k+1}] + 2\text{tr}[\alpha(\boldsymbol{z}^{k+1} - \boldsymbol{z}^k)^\top (\boldsymbol{z}^{k+1} - \boldsymbol{z}^*)] \le 0$$

$$\frac{1}{\alpha}\big(\|\boldsymbol{\rho}^{k+1} - \boldsymbol{\rho}^*\|_F^2 - \|\boldsymbol{\rho}^k - \boldsymbol{\rho}^*\|_F^2\big) + \alpha\|\boldsymbol{r}^{k+1} + (\boldsymbol{z}^{k+1} - \boldsymbol{z}^k)\|_F^2 + \alpha\big[\|\boldsymbol{z}^{k+1} - \boldsymbol{z}^*\|_F^2 - \|\boldsymbol{z}^k - \boldsymbol{z}^*\|_F^2\big] \le 0$$

$$\frac{1}{\alpha}\|\boldsymbol{\rho}^{k+1} - \boldsymbol{\rho}^*\|_F^2 + \alpha\|\boldsymbol{z}^{k+1} - \boldsymbol{z}^*\|_F^2 - \frac{1}{\alpha}\|\boldsymbol{\rho}^k - \boldsymbol{\rho}^*\|_F^2 - \alpha\|\boldsymbol{z}^k - \boldsymbol{z}^*\|_F^2 + \alpha\|\boldsymbol{r}^{k+1} + (\boldsymbol{z}^{k+1} - \boldsymbol{z}^k)\|_F^2 \le 0. \quad (37)$$

Define

$$V^k = \frac{1}{\alpha}\|\boldsymbol{\rho}^k - \boldsymbol{\rho}^*\|_F^2 + \alpha\|\boldsymbol{z}^k - \boldsymbol{z}^*\|_F^2 \ge 0.$$

Then Inequality (37) can be expressed as

$$\begin{aligned}
V^{k+1} - V^k &\le -\alpha\|\boldsymbol{r}^{k+1} + (\boldsymbol{z}^{k+1} - \boldsymbol{z}^k)\|_F^2 \\
&\le -\alpha\big(\|\boldsymbol{r}^{k+1}\|_F^2 + 2\text{tr}[(\boldsymbol{r}^{k+1})^\top (\boldsymbol{z}^{k+1} - \boldsymbol{z}^k)] + \|\boldsymbol{z}^{k+1} - \boldsymbol{z}^k\|_F^2\big) \\
&\le -\alpha\|\boldsymbol{r}^{k+1}\|_F^2 - \alpha\|\boldsymbol{z}^{k+1} - \boldsymbol{z}^k\|_F^2 - 2\alpha\text{tr}[(\boldsymbol{r}^{k+1})^\top (\boldsymbol{z}^{k+1} - \boldsymbol{z}^k)]. 
\end{aligned} \quad (38)$$

Recall that $\boldsymbol{z}^{k+1}$ minimizes $g(\boldsymbol{z}) - \text{tr}[(\boldsymbol{\rho}^{k+1})^\top \boldsymbol{z}]$ and $\boldsymbol{z}^k$ minimizes $g(\boldsymbol{z}) - \text{tr}[(\boldsymbol{\rho}^k)^\top \boldsymbol{z}]$. Then

$$g(\boldsymbol{z}^{k+1}) - \text{tr}[(\boldsymbol{\rho}^{k+1})^\top \boldsymbol{z}^{k+1}] \le g(\boldsymbol{z}^k) - \text{tr}[(\boldsymbol{\rho}^{k+1})^\top \boldsymbol{z}^k]$$

and

$$g(\boldsymbol{z}^k) - \text{tr}[(\boldsymbol{\rho}^k)^\top \boldsymbol{z}^k] \le g(\boldsymbol{z}^{k+1}) - \text{tr}[(\boldsymbol{\rho}^k)^\top \boldsymbol{z}^{k+1}].$$

Adding the above two inequalities, we have

$$\begin{aligned}
\text{tr}[(\boldsymbol{\rho}^k)^\top \boldsymbol{z}^{k+1}] + \text{tr}[(\boldsymbol{\rho}^{k+1})^\top \boldsymbol{z}^k] - \text{tr}[(\boldsymbol{\rho}^{k+1})^\top \boldsymbol{z}^{k+1}] - \text{tr}[(\boldsymbol{\rho}^k)^\top \boldsymbol{z}^k] &\le 0 \\
\text{tr}[(\boldsymbol{\rho}^k)^\top (\boldsymbol{z}^{k+1} - \boldsymbol{z}^k)] + \text{tr}[(\boldsymbol{\rho}^{k+1})^\top (\boldsymbol{z}^k - \boldsymbol{z}^{k+1})] &\le 0 \\
-\text{tr}[(\boldsymbol{\rho}^{k+1} - \boldsymbol{\rho}^k)^\top (\boldsymbol{z}^{k+1} - \boldsymbol{z}^k)] &\le 0.
\end{aligned}$$

Recall $\boldsymbol{\rho}^{k+1} - \boldsymbol{\rho}^k = \alpha\boldsymbol{r}^{k+1}$. Then

$$-\alpha\text{tr}[(\boldsymbol{r}^{k+1})^\top (\boldsymbol{z}^{k+1} - \boldsymbol{z}^k)] \le 0.$$

Back to Inequality (38), we can see that

$$V^{k+1} - V^k \le -\alpha\|\boldsymbol{r}^{k+1}\|_F^2 - \alpha\|\boldsymbol{z}^{k+1} - \boldsymbol{z}^k\|_F^2$$

also hold. Since $V^k \ge 0$ and

$$V^{k+1} \le V^k - \alpha\big(\|\boldsymbol{r}^{k+1}\|_F^2 + \|\boldsymbol{z}^{k+1} - \boldsymbol{z}^k\|_F^2\big), \quad (39)$$

we know that $V^k$ is a bounded by below decreasing sequence.

By iterating (39), we have

$$\alpha\sum_{k=0}^{\infty}\big(\|\boldsymbol{r}^{k+1}\|_F^2 + \|\boldsymbol{z}^{k+1} - \boldsymbol{z}^k\|_F^2\big) \le V^0$$

which implies the primal residual $\|\boldsymbol{r}^{k+1}\|_F^2 = \|\boldsymbol{\theta}^{k+1} - \boldsymbol{z}^{k+1}\|_F^2 \to 0$ and dual residual $\|\boldsymbol{z}^{k+1} - \boldsymbol{z}^k\|_F^2 \to 0$ as $k \to \infty$. Similarly, for

$$r_{\text{primal}}^k = \sqrt{\frac{1}{\tau \times p}\sum_{i=1}^{\tau}\sum_{j=1}^{p}(\theta_{ij}^k - z_{ij}^k)^2} \quad \text{and} \quad r_{\text{dual}}^k = \sqrt{\frac{1}{\tau \times p}\sum_{i=1}^{\tau}\sum_{j=1}^{p}(z_{ij}^k - z_{ij}^{k-1})^2},$$

we have $r_{\text{primal}}^k \to 0$ and $r_{\text{dual}}^k \to 0$ as $k \to \infty$. This concludes the proof for Proposition 1.

## B    Newton-Raphson Method for Updating $\theta$

In this section, we derive the gradient and Hessian for the Newton-Raphson method to update $\boldsymbol{\theta}$. The first-order derivative of $l(\boldsymbol{\theta})$ with respect to $\boldsymbol{\theta}^{+,t} \in \mathbb{R}^{p_1}$, the parameter in the formation model at a particular time point $t$, is

$$
\begin{aligned}
\nabla_{\boldsymbol{\theta}^{+,t}} \, l(\boldsymbol{\theta}) &= \sum_{(i,j) \in \mathbb{Y}} \left\{ \boldsymbol{y}_{ij}^{+,t} \Delta \boldsymbol{g}_{ij}^{+}(\boldsymbol{y}^{+,t}) - \frac{\exp[\boldsymbol{\theta}^{+,t} \cdot \Delta \boldsymbol{g}_{ij}^{+}(\boldsymbol{y}^{+,t})]}{1 + \exp[\boldsymbol{\theta}^{+,t} \cdot \Delta \boldsymbol{g}_{ij}^{+}(\boldsymbol{y}^{+,t})]} \Delta \boldsymbol{g}_{ij}^{+}(\boldsymbol{y}^{+,t}) \right\} \\
&= \sum_{(i,j) \in \mathbb{Y}} (\boldsymbol{y}_{ij}^{+,t} - \boldsymbol{\mu}_{ij}^{+,t}) \cdot \Delta \boldsymbol{g}_{ij}^{+}(\boldsymbol{y}^{+,t})
\end{aligned}
$$

where $\boldsymbol{\mu}_{ij}^{+,t} = h(\boldsymbol{\theta}^{+,t} \cdot \Delta \boldsymbol{g}_{ij}^{+}(\boldsymbol{y}^{+,t})) \in (0,1)$. The $h(x) = 1/(1 + \exp(-x))$ is the element-wise sigmoid function. Likewise, the first-order derivative of $l(\boldsymbol{\theta})$ with respect to $\boldsymbol{\theta}^{-,t} \in \mathbb{R}^{p_2}$, the parameter in the dissolution model at a particular time point $t$, is similar except for notational difference.

Denote the objective function in Equation (10) as $\mathcal{L}_\alpha(\boldsymbol{\theta})$. To update the parameters $\boldsymbol{\theta} \in \mathbb{R}^{\tau \times p}$ in a compact form, we first vectorize it as $\vec{\boldsymbol{\theta}} = \mathrm{vec}_{\tau p}(\boldsymbol{\theta}) \in \mathbb{R}^{\tau p \times 1}$. The matrices $\boldsymbol{z} \in \mathbb{R}^{\tau \times p}$ and $\boldsymbol{u} \in \mathbb{R}^{\tau \times p}$ are also vectorized as $\vec{\boldsymbol{z}} = \mathrm{vec}_{\tau p}(\boldsymbol{z}) \in \mathbb{R}^{\tau p \times 1}$ and $\vec{\boldsymbol{u}} = \mathrm{vec}_{\tau p}(\boldsymbol{u}) \in \mathbb{R}^{\tau p \times 1}$. With the constructed matrices $\boldsymbol{H} \in \mathbb{R}^{2\tau E \times \tau p}$, the gradient of $\mathcal{L}_\alpha(\boldsymbol{\theta})$ with respect to $\vec{\boldsymbol{\theta}}$ is

$$
\nabla_{\vec{\boldsymbol{\theta}}} \, \mathcal{L}_\alpha(\boldsymbol{\theta}) = -\boldsymbol{H}^\top (\vec{\boldsymbol{y}} - \vec{\boldsymbol{\mu}}) + \alpha(\vec{\boldsymbol{\theta}} - \vec{\boldsymbol{z}}^{(a)} + \vec{\boldsymbol{u}}^{(a)})
$$

where $\vec{\boldsymbol{\mu}} = h(\boldsymbol{H} \cdot \vec{\boldsymbol{\theta}}) \in (0,1)^{2\tau E \times 1}$.

Furthermore, the second order derivative of $l(\boldsymbol{\theta})$ with respect to $\boldsymbol{\theta}^{+,t} \in \mathbb{R}^{p_1}$ is

$$
\nabla_{\boldsymbol{\theta}^{+,t}}^2 \, l(\boldsymbol{\theta}) = \sum_{(i,j) \in \mathbb{Y}} -\boldsymbol{\mu}_{ij}^{+,t}(1 - \boldsymbol{\mu}_{ij}^{+,t}) [\Delta \boldsymbol{g}_{ij}^{+}(\boldsymbol{y}^{+,t}) \Delta \boldsymbol{g}_{ij}^{+}(\boldsymbol{y}^{+,t})^\top]
$$

and the second order derivative of $l(\boldsymbol{\theta})$ with respect to $\boldsymbol{\theta}^{-,t} \in \mathbb{R}^{p_2}$ is similar except for notational difference. Thus, with the constructed matrices $\boldsymbol{H} \in \mathbb{R}^{2\tau E \times \tau p}$ and $\boldsymbol{W} \in (0, 1/4)^{2\tau E \times 2\tau E}$, the Hessian of $\mathcal{L}_\alpha(\boldsymbol{\theta})$ with respect to $\vec{\boldsymbol{\theta}} \in \mathbb{R}^{\tau p \times 1}$ is

$$
\nabla_{\vec{\boldsymbol{\theta}}}^2 \, \mathcal{L}_\alpha(\boldsymbol{\theta}) = \boldsymbol{H}^\top \boldsymbol{W} \boldsymbol{H} + \alpha \boldsymbol{I}_{\tau p}
$$

where $\boldsymbol{I}_{\tau p}$ is the identity matrix. By using the Newton-Raphson method, the $\vec{\boldsymbol{\theta}} \in \mathbb{R}^{\tau p \times 1}$ is updated as

$$
\vec{\boldsymbol{\theta}}_{c+1} = \vec{\boldsymbol{\theta}}_c - \left(\boldsymbol{H}^\top \boldsymbol{W} \boldsymbol{H} + \alpha \boldsymbol{I}_{\tau p}\right)^{-1} \cdot \left(-\boldsymbol{H}^\top (\vec{\boldsymbol{y}} - \vec{\boldsymbol{\mu}}) + \alpha(\vec{\boldsymbol{\theta}}_c - \vec{\boldsymbol{z}}^{(a)} + \vec{\boldsymbol{u}}^{(a)})\right)
$$

where $c$ denotes the current Newton-Raphson iteration. Note that both $\boldsymbol{W}$ and $\vec{\boldsymbol{\mu}}$ are also calculated based on $\vec{\boldsymbol{\theta}}_c$. The constructed formation and dissolution network data is vectorized in the form of $\vec{\boldsymbol{y}} \in \{0,1\}^{2\tau E \times 1}$ to align with the dyad order of the matrix $\boldsymbol{H} \in \mathbb{R}^{2\tau E \times \tau p}$.

## C    Group Lasso for Updating $\beta$

In this section, we provide the derivation to update $\boldsymbol{\beta}$, which is equivalent to solving a Group Lasso problem (Yuan & Lin, 2006). Denote the objective function in (11) as $\mathcal{L}_\alpha(\boldsymbol{\gamma}, \boldsymbol{\beta})$. When $\boldsymbol{\beta}_{i,\cdot} \neq \boldsymbol{0}$, the first-order derivative of $\mathcal{L}_\alpha(\boldsymbol{\gamma}, \boldsymbol{\beta})$ with respect to $\boldsymbol{\beta}_{i,\cdot}$ is

$$
\frac{\partial}{\partial \boldsymbol{\beta}_{i,\cdot}} \mathcal{L}_\alpha(\boldsymbol{\gamma}, \boldsymbol{\beta}) = \lambda \frac{\boldsymbol{\beta}_{i,\cdot}}{\|\boldsymbol{\beta}_{i,\cdot}\|_2} - \alpha \boldsymbol{X}_{\cdot,i}^\top (\boldsymbol{\theta}^{(a+1)} + \boldsymbol{u}^{(a)} - \boldsymbol{1}_{\tau,1} \boldsymbol{\gamma} - \boldsymbol{X}_{\cdot,i} \boldsymbol{\beta}_{i,\cdot} - \boldsymbol{X}_{\cdot,-i} \boldsymbol{\beta}_{-i,\cdot})
$$

where $\boldsymbol{X}_{\cdot,i} \in \mathbb{R}^{\tau \times 1}$ is the $i$th column of matrix $\boldsymbol{X} \in \mathbb{R}^{\tau \times (\tau-1)}$ and $\boldsymbol{\beta}_{i,\cdot} \in \mathbb{R}^{1 \times p}$ is the $i$th row of matrix $\boldsymbol{\beta} \in \mathbb{R}^{(\tau-1) \times p}$. Setting the gradient to $\boldsymbol{0}$, we have

$$
\boldsymbol{\beta}_{i,\cdot} = \left(\alpha \boldsymbol{X}_{\cdot,i}^\top \boldsymbol{X}_{\cdot,i} + \frac{\lambda}{\|\boldsymbol{\beta}_{i,\cdot}\|_2}\right)^{-1} \boldsymbol{s}_i \tag{40}
$$

where

$$\boldsymbol{s}_i = \alpha \boldsymbol{X}_{\cdot,i}^\top (\boldsymbol{\theta}^{(a+1)} + \boldsymbol{u}^{(a)} - \boldsymbol{1}_{\tau,1}\boldsymbol{\gamma} - \boldsymbol{X}_{\cdot,-i}\boldsymbol{\beta}_{-i,\cdot}).$$

Taking the Euclidean norm of (40) on both sides and rearrange the terms, we have

$$\|\boldsymbol{\beta}_{i,\cdot}\|_2 = (\alpha \boldsymbol{X}_{\cdot,i}^\top \boldsymbol{X}_{\cdot,i})^{-1} (\|\boldsymbol{s}_i\|_2 - \lambda).$$

Plugging $\|\boldsymbol{\beta}_{i,\cdot}\|_2$ into (40), the solution of $\boldsymbol{\beta}_{i,\cdot}$ is

$$\boldsymbol{\beta}_{i,\cdot} = \frac{1}{\alpha \boldsymbol{X}_{\cdot,i}^\top \boldsymbol{X}_{\cdot,i}} \left(1 - \frac{\lambda}{\|\boldsymbol{s}_i\|_2}\right) \boldsymbol{s}_i.$$

When $\boldsymbol{\beta}_{i,\cdot} = \boldsymbol{0}$, the subgradient $\boldsymbol{v}$ of $\|\boldsymbol{\beta}_{i,\cdot}\|_2$ needs to satisfy $\|\boldsymbol{v}\|_2 \leq 1$. Since

$$\boldsymbol{0} \in \lambda \boldsymbol{v} - \alpha \boldsymbol{X}_{\cdot,i}^\top (\boldsymbol{\theta}^{(a+1)} + \boldsymbol{u}^{(a)} - \boldsymbol{1}_{\tau,1}\boldsymbol{\gamma} - \boldsymbol{X}_{\cdot,-i}\boldsymbol{\beta}_{-i,\cdot}),$$

we have $\boldsymbol{v} = \lambda^{-1}\boldsymbol{s}_i$ and we obtain the condition that $\boldsymbol{\beta}_{i,\cdot}$ becomes $\boldsymbol{0}$ if $\|\boldsymbol{s}_i\|_2 \leq \lambda$. Therefore, to update $\boldsymbol{\beta}_{i,\cdot}$ for each $i = 1, \ldots, \tau - 1$, we can iteratively apply the following equation:

$$\boldsymbol{\beta}_{i,\cdot} \leftarrow \frac{1}{\alpha \boldsymbol{X}_{\cdot,i}^\top \boldsymbol{X}_{\cdot,i}} \left(1 - \frac{\lambda}{\|\boldsymbol{s}_i\|_2}\right)_+ \boldsymbol{s}_i$$

where $(\cdot)_+ = \max(\cdot, 0)$. The matrix $\boldsymbol{X} \in \mathbb{R}^{\tau \times (\tau-1)}$ is constructed from the position dependent weight $\boldsymbol{d} \in \mathbb{R}^{\tau-1}$.

## D    Error Bound under Structured Sparsity

In this section, we provide the proof for Proposition 2. Denote $\boldsymbol{\theta}^* \in \mathbb{R}^{\tau \times p}$ as the true parameter and suppose that $\|\boldsymbol{\theta}^*\|_\infty \leq M/2$ for some $M > 0$. Let $\hat{\boldsymbol{\theta}} \in \mathbb{R}^{\tau \times p}$ be the minimizer of the objective function in (5), subject to the constraint $\|\boldsymbol{\theta}\|_\infty \leq M/2$. Since $\hat{\boldsymbol{\theta}}$ minimizes the penalized objective, we have

$$L(\hat{\boldsymbol{\theta}}) + \lambda \sum_{i=1}^{\tau-1} \frac{\|\hat{\boldsymbol{\theta}}_{i+1,\cdot} - \hat{\boldsymbol{\theta}}_{i,\cdot}\|_2}{\boldsymbol{d}_i} \leq L(\boldsymbol{\theta}^*) + \lambda \sum_{i=1}^{\tau-1} \frac{\|\boldsymbol{\theta}_{i+1,\cdot}^* - \boldsymbol{\theta}_{i,\cdot}^*\|_2}{\boldsymbol{d}_i}$$

$$L(\hat{\boldsymbol{\theta}}) - L(\boldsymbol{\theta}^*) \leq \lambda \sum_{i=1}^{\tau-1} \frac{\|\boldsymbol{\theta}_{i+1,\cdot}^* - \boldsymbol{\theta}_{i,\cdot}^*\|_2 - \|\hat{\boldsymbol{\theta}}_{i+1,\cdot} - \hat{\boldsymbol{\theta}}_{i,\cdot}\|_2}{\boldsymbol{d}_i} \tag{41}$$

where $L(\boldsymbol{\theta}) := -l(\boldsymbol{\theta})$.

Define the estimation error as $\hat{\boldsymbol{\Delta}} := \hat{\boldsymbol{\theta}} - \boldsymbol{\theta}^* \in \mathbb{R}^{\tau \times p}$. Using the triangle inequality $\|\boldsymbol{a} + \boldsymbol{b}\|_2 \geq \|\boldsymbol{b}\|_2 - \|\boldsymbol{a}\|_2$ with $\boldsymbol{a} = \hat{\boldsymbol{\Delta}}_{i+1,\cdot} - \hat{\boldsymbol{\Delta}}_{i,\cdot}$ and $\boldsymbol{b} = \boldsymbol{\theta}_{i+1,\cdot}^* - \boldsymbol{\theta}_{i,\cdot}^*$ such that $\boldsymbol{a} + \boldsymbol{b} = \hat{\boldsymbol{\theta}}_{i+1,\cdot} - \hat{\boldsymbol{\theta}}_{i,\cdot}$, we obtain:

$$\|\hat{\boldsymbol{\theta}}_{i+1,\cdot} - \hat{\boldsymbol{\theta}}_{i,\cdot}\|_2 \geq \|\boldsymbol{\theta}_{i+1,\cdot}^* - \boldsymbol{\theta}_{i,\cdot}^*\|_2 - \|\hat{\boldsymbol{\Delta}}_{i+1,\cdot} - \hat{\boldsymbol{\Delta}}_{i,\cdot}\|_2$$

$$\|\hat{\boldsymbol{\Delta}}_{i+1,\cdot} - \hat{\boldsymbol{\Delta}}_{i,\cdot}\|_2 \geq \|\boldsymbol{\theta}_{i+1,\cdot}^* - \boldsymbol{\theta}_{i,\cdot}^*\|_2 - \|\hat{\boldsymbol{\theta}}_{i+1,\cdot} - \hat{\boldsymbol{\theta}}_{i,\cdot}\|_2 \tag{42}$$

for $i = 1, \cdots, \tau - 1$. Applying Inequality (42) to the right-hand side of (41), we obtain

$$L(\hat{\boldsymbol{\theta}}) - L(\boldsymbol{\theta}^*) \leq \lambda \sum_{i=1}^{\tau-1} \frac{\|\hat{\boldsymbol{\Delta}}_{i+1,\cdot} - \hat{\boldsymbol{\Delta}}_{i,\cdot}\|_2}{\boldsymbol{d}_i}. \tag{43}$$

Now, we define the set of true change points as

$$S = \left\{ i \in \{1, \ldots, \tau - 1\} : \boldsymbol{\theta}_{i+1,\cdot}^* \neq \boldsymbol{\theta}_{i,\cdot}^* \right\}.$$

Suppose the loss function $L(\boldsymbol{\theta})$ satisfies the Restricted Strong Convexity condition:

$$L(\boldsymbol{\theta}^* + \boldsymbol{\Delta}) \geq L(\boldsymbol{\theta}^*) + \langle \nabla L(\boldsymbol{\theta}^*), \boldsymbol{\Delta} \rangle + \frac{k}{2} \|\boldsymbol{\Delta}\|_F^2 \tag{44}$$

for all perturbations $\boldsymbol{\Delta} \in \mathbb{R}^{\tau \times p}$ that satisfy the structured sparsity condition:

$$\sum_{i \notin S} \frac{\|\boldsymbol{\Delta}_{i+1,\cdot} - \boldsymbol{\Delta}_{i,\cdot}\|_2}{\boldsymbol{d}_i} \leq \alpha \sum_{i \in S} \frac{\|\boldsymbol{\Delta}_{i+1,\cdot} - \boldsymbol{\Delta}_{i,\cdot}\|_2}{\boldsymbol{d}_i} \tag{45}$$

for some constants $k > 0$ and $\alpha > 0$.

Assume the estimation error $\hat{\boldsymbol{\Delta}} := \hat{\boldsymbol{\theta}} - \boldsymbol{\theta}^* \in \mathbb{R}^{\tau \times p}$ satisfies (45). Then from Inequality (43), we have

$$L(\hat{\boldsymbol{\theta}}) - L(\boldsymbol{\theta}^*) \leq \lambda \left( \sum_{i \in S} \frac{\|\hat{\boldsymbol{\Delta}}_{i+1,\cdot} - \hat{\boldsymbol{\Delta}}_{i,\cdot}\|_2}{\boldsymbol{d}_i} + \sum_{i \notin S} \frac{\|\hat{\boldsymbol{\Delta}}_{i+1,\cdot} - \hat{\boldsymbol{\Delta}}_{i,\cdot}\|_2}{\boldsymbol{d}_i} \right)$$

$$L(\hat{\boldsymbol{\theta}}) - L(\boldsymbol{\theta}^*) \leq \lambda \cdot (1 + \alpha) \sum_{i \in S} \frac{\|\hat{\boldsymbol{\Delta}}_{i+1,\cdot} - \hat{\boldsymbol{\Delta}}_{i,\cdot}\|_2}{\boldsymbol{d}_i} \tag{46}$$

where (46) follows by the condition in (45).

Moreover, since $\langle \nabla L(\boldsymbol{\theta}^*), \hat{\boldsymbol{\Delta}} \rangle \geq -|\langle \nabla L(\boldsymbol{\theta}^*), \hat{\boldsymbol{\Delta}} \rangle|$ and $\boldsymbol{\theta}^* + \hat{\boldsymbol{\Delta}} = \hat{\boldsymbol{\theta}}$, it follows from the Restricted Strong Convexity condition in (44) that:

$$L(\boldsymbol{\theta}^* + \hat{\boldsymbol{\Delta}}) - L(\boldsymbol{\theta}^*) \geq \frac{k}{2} \|\hat{\boldsymbol{\Delta}}\|_F^2 + \langle \nabla L(\boldsymbol{\theta}^*), \hat{\boldsymbol{\Delta}} \rangle$$

$$L(\hat{\boldsymbol{\theta}}) - L(\boldsymbol{\theta}^*) \geq \frac{k}{2} \|\hat{\boldsymbol{\Delta}}\|_F^2 - |\langle \nabla L(\boldsymbol{\theta}^*), \hat{\boldsymbol{\Delta}} \rangle|$$

$$\lambda \cdot (1 + \alpha) \sum_{i \in S} \frac{\|\hat{\boldsymbol{\Delta}}_{i+1,\cdot} - \hat{\boldsymbol{\Delta}}_{i,\cdot}\|_2}{\boldsymbol{d}_i} \geq \frac{k}{2} \|\hat{\boldsymbol{\Delta}}\|_F^2 - |\langle \nabla L(\boldsymbol{\theta}^*), \hat{\boldsymbol{\Delta}} \rangle| \tag{47}$$

where (47) follows by (46).

Next, we denote the Total Variation (TV) norm $\|\cdot\|_{\mathrm{TV}}$, for a matrix $\boldsymbol{A} \in \mathbb{R}^{\tau \times p}$, as

$$\|\boldsymbol{A}\|_{\mathrm{TV}} := \sum_{i=1}^{\tau-1} \frac{\|\boldsymbol{A}_{i+1,\cdot} - \boldsymbol{A}_{i,\cdot}\|_2}{\boldsymbol{d}_i} \geq 0$$

where $\{\boldsymbol{d}_i\}_{i=1}^{\tau-1} > 0$ are the position dependent weights. Denote a constraint set $\mathcal{C}$ as

$$\mathcal{C} = \left\{ \boldsymbol{A} \in \mathbb{R}^{\tau \times p} : \|\boldsymbol{A}\|_{\mathrm{TV}} \leq 1, \ \|\boldsymbol{A}\|_\infty \leq 1 \right\}.$$

Then we define a restricted dual norm $\|\cdot\|_*$, for a matrix $\boldsymbol{B} \in \mathbb{R}^{\tau \times p}$, as

$$\|\boldsymbol{B}\|_* := \sup_{\boldsymbol{A} \in \mathcal{C}} \langle \boldsymbol{B}, \boldsymbol{A} \rangle.$$

Let

$$c := \max \left\{ \|\hat{\boldsymbol{\Delta}}\|_{\mathrm{TV}}, \|\hat{\boldsymbol{\Delta}}\|_\infty \right\} \geq 0 \quad \text{and} \quad \widetilde{\boldsymbol{\Delta}} = \frac{1}{c} \hat{\boldsymbol{\Delta}}.$$

Note that

$$\|\widetilde{\boldsymbol{\Delta}}\|_{\mathrm{TV}} = \frac{1}{c} \|\hat{\boldsymbol{\Delta}}\|_{\mathrm{TV}} \leq \frac{1}{c} \cdot c = 1,$$

$$\|\widetilde{\boldsymbol{\Delta}}\|_\infty = \frac{1}{c} \|\hat{\boldsymbol{\Delta}}\|_\infty \leq \frac{1}{c} \cdot c = 1,$$

so $\widetilde{\boldsymbol{\Delta}} \in \mathcal{C}$ and

$$\langle \nabla L(\boldsymbol{\theta}^*), \widetilde{\boldsymbol{\Delta}} \rangle \leq \sup_{\widetilde{\boldsymbol{\Delta}} \in \mathcal{C}} \langle \nabla L(\boldsymbol{\theta}^*), \widetilde{\boldsymbol{\Delta}} \rangle = \|\nabla L(\boldsymbol{\theta}^*)\|_*.$$

Furthermore, we have

$$\begin{aligned}
|\langle \nabla L(\boldsymbol{\theta}^*), \hat{\boldsymbol{\Delta}} \rangle| &= c \cdot |\langle \nabla L(\boldsymbol{\theta}^*), \widetilde{\boldsymbol{\Delta}} \rangle| \\
&\leq c \cdot \|\nabla L(\boldsymbol{\theta}^*)\|_* \\
&\leq \|\nabla L(\boldsymbol{\theta}^*)\|_* \cdot \max\{\|\hat{\boldsymbol{\Delta}}\|_{\mathrm{TV}}, \ \|\hat{\boldsymbol{\Delta}}\|_\infty\} \\
&\leq \|\nabla L(\boldsymbol{\theta}^*)\|_* \cdot \max\{\|\hat{\boldsymbol{\Delta}}\|_{\mathrm{TV}}, \ M\}
\end{aligned} \tag{48}$$

where $\|\hat{\boldsymbol{\Delta}}\|_\infty \leq \|\boldsymbol{\theta}^*\|_\infty + \|\hat{\boldsymbol{\theta}}\|_\infty \leq M/2 + M/2 = M$.

We now assume that

$$\|\nabla L(\boldsymbol{\theta}^*)\|_* \leq \frac{\lambda}{2}$$

holds with probability at least $1 - \delta$. Then from (48), we obtain

$$\begin{aligned}
|\langle \nabla L(\boldsymbol{\theta}^*), \hat{\boldsymbol{\Delta}} \rangle| &\leq \frac{\lambda}{2} \cdot \max\{\|\hat{\boldsymbol{\Delta}}\|_{\mathrm{TV}}, \ M\} \\
&\leq \frac{\lambda}{2} \cdot \max\left\{ \sum_{i=1}^{\tau-1} \frac{\|\hat{\boldsymbol{\Delta}}_{i+1,\cdot} - \hat{\boldsymbol{\Delta}}_{i,\cdot}\|_2}{\boldsymbol{d}_i}, M \right\} \\
&\leq \frac{\lambda}{2} \cdot \max\left\{ \sum_{i \in S} \frac{\|\hat{\boldsymbol{\Delta}}_{i+1,\cdot} - \hat{\boldsymbol{\Delta}}_{i,\cdot}\|_2}{\boldsymbol{d}_i} + \sum_{i \notin S} \frac{\|\hat{\boldsymbol{\Delta}}_{i+1,\cdot} - \hat{\boldsymbol{\Delta}}_{i,\cdot}\|_2}{\boldsymbol{d}_i}, M \right\} \\
&\leq \frac{\lambda}{2} \cdot \max\left\{ (1 + \alpha) \sum_{i \in S} \frac{\|\hat{\boldsymbol{\Delta}}_{i+1,\cdot} - \hat{\boldsymbol{\Delta}}_{i,\cdot}\|_2}{\boldsymbol{d}_i}, M \right\} \\
&\leq \frac{\lambda}{2} \cdot \left\{ (1 + \alpha) \sum_{i \in S} \frac{\|\hat{\boldsymbol{\Delta}}_{i+1,\cdot} - \hat{\boldsymbol{\Delta}}_{i,\cdot}\|_2}{\boldsymbol{d}_i} + M \right\}
\end{aligned}$$

$$\tag{49}$$
$$\tag{50}$$

where (49) follows by the condition in (45), and (50) follows by $\max(a, b) \leq a + b$ for $a, b \geq 0$.

Substituting (50) into (47), we have

$$\begin{aligned}
\frac{k}{2}\|\hat{\boldsymbol{\Delta}}\|_F^2 - \frac{\lambda}{2} \cdot \left\{ (1 + \alpha) \sum_{i \in S} \frac{\|\hat{\boldsymbol{\Delta}}_{i+1,\cdot} - \hat{\boldsymbol{\Delta}}_{i,\cdot}\|_2}{\boldsymbol{d}_i} + M \right\} &\leq \lambda \cdot (1 + \alpha) \sum_{i \in S} \frac{\|\hat{\boldsymbol{\Delta}}_{i+1,\cdot} - \hat{\boldsymbol{\Delta}}_{i,\cdot}\|_2}{\boldsymbol{d}_i} \\
\frac{k}{2}\|\hat{\boldsymbol{\Delta}}\|_F^2 - \frac{\lambda}{2} \cdot (1 + \alpha) \sum_{i \in S} \frac{\|\hat{\boldsymbol{\Delta}}_{i+1,\cdot} - \hat{\boldsymbol{\Delta}}_{i,\cdot}\|_2}{\boldsymbol{d}_i} - \frac{\lambda}{2}M &\leq \lambda \cdot (1 + \alpha) \sum_{i \in S} \frac{\|\hat{\boldsymbol{\Delta}}_{i+1,\cdot} - \hat{\boldsymbol{\Delta}}_{i,\cdot}\|_2}{\boldsymbol{d}_i} \\
\frac{k}{2}\|\hat{\boldsymbol{\Delta}}\|_F^2 &\leq \frac{3}{2}\lambda(1 + \alpha) \sum_{i \in S} \frac{\|\hat{\boldsymbol{\Delta}}_{i+1,\cdot} - \hat{\boldsymbol{\Delta}}_{i,\cdot}\|_2}{\boldsymbol{d}_i} + \frac{\lambda}{2}M.
\end{aligned} \tag{51}$$

Note that $\sum_{i \in S} \|\hat{\boldsymbol{\Delta}}_{i+1,\cdot} - \hat{\boldsymbol{\Delta}}_{i,\cdot}\|_2^2 \leq \|\hat{\boldsymbol{\Delta}}\|_F^2$. Then with Cauchy-Schwarz inequality, we have

$$\begin{aligned}
\sum_{i \in S} \frac{\|\hat{\boldsymbol{\Delta}}_{i+1,\cdot} - \hat{\boldsymbol{\Delta}}_{i,\cdot}\|_2}{\boldsymbol{d}_i} &\leq \sqrt{\sum_{i \in S} \frac{1}{\boldsymbol{d}_i^2}} \cdot \sqrt{\sum_{i \in S} \|\hat{\boldsymbol{\Delta}}_{i+1,\cdot} - \hat{\boldsymbol{\Delta}}_{i,\cdot}\|_2^2} \\
&\leq \sqrt{\sum_{i \in S} \boldsymbol{d}_i^{-2}} \cdot \|\hat{\boldsymbol{\Delta}}\|_F.
\end{aligned} \tag{52}$$

Substituting (52) into (51), we have

$$\frac{k}{2}\|\hat{\boldsymbol{\Delta}}\|_F^2 \leq \frac{3}{2}\lambda(1+\alpha) \cdot \left[\sqrt{\sum_{i \in S} \boldsymbol{d}_i^{-2}} \cdot \|\hat{\boldsymbol{\Delta}}\|_F\right] + \frac{\lambda}{2}M$$

$$k\|\hat{\boldsymbol{\Delta}}\|_F^2 \leq 3\lambda(1+\alpha) \cdot \sqrt{\sum_{i \in S} \boldsymbol{d}_i^{-2}} \cdot \|\hat{\boldsymbol{\Delta}}\|_F + \lambda M. \tag{53}$$

Note that for $a, b > 0$, we have $a + b \leq \max(2a, 2b)$. Then from (53), we have

$$k\|\hat{\boldsymbol{\Delta}}\|_F^2 \leq \max\left\{6\lambda(1+\alpha) \cdot \sqrt{\sum_{i \in S} \boldsymbol{d}_i^{-2}} \cdot \|\hat{\boldsymbol{\Delta}}\|_F, \ 2\lambda M\right\}.$$

This implies that either

$$k\|\hat{\boldsymbol{\Delta}}\|_F^2 \leq 6\lambda(1+\alpha) \cdot \sqrt{\sum_{i \in S} \boldsymbol{d}_i^{-2}} \cdot \|\hat{\boldsymbol{\Delta}}\|_F \tag{54}$$

or

$$k\|\hat{\boldsymbol{\Delta}}\|_F^2 \leq 2\lambda M. \tag{55}$$

In the first case with (54), we have

$$\|\hat{\boldsymbol{\Delta}}\|_F^2 \leq \left[\frac{6\lambda}{k}(1+\alpha) \cdot \sqrt{\sum_{i \in S} \boldsymbol{d}_i^{-2}}\right]^2.$$

In the second case with (55), we have

$$\|\hat{\boldsymbol{\Delta}}\|_F^2 \leq \frac{2\lambda M}{k}.$$

Combining the two cases, we have

$$\|\hat{\boldsymbol{\Delta}}\|_F^2 \leq \max\left\{\left[\frac{6\lambda}{k}(1+\alpha) \cdot \sqrt{\sum_{i \in S} \boldsymbol{d}_i^{-2}}\right]^2, \ \frac{2\lambda M}{k}\right\}.$$

Thus,

$$\frac{1}{\tau p}\|\hat{\boldsymbol{\theta}} - \boldsymbol{\theta}^*\|_F^2 \leq \frac{1}{\tau p}\max\left\{\left[\frac{6\lambda}{k}(1+\alpha) \cdot \sqrt{\sum_{i \in S} \boldsymbol{d}_i^{-2}}\right]^2, \ \frac{2\lambda M}{k}\right\}.$$

This concludes the proof for Proposition 2.

## E Practical Guidelines

As in Boyd et al. (2011), we also update the penalty parameter $\alpha$ for the augmentation term to improve convergence and to reduce reliance on its initial choice. Specifically, after the completion of an ADMM iteration, we calculate the respective primal and dual residuals:

$$r_{\text{primal}}^{(a)} = \sqrt{\frac{1}{\tau \times p} \sum_{i=1}^{\tau} \sum_{j=1}^{p} (\boldsymbol{\theta}_{ij}^{(a)} - \boldsymbol{z}_{ij}^{(a)})^2} \ \text{ and } \ r_{\text{dual}}^{(a)} = \sqrt{\frac{1}{\tau \times p} \sum_{i=1}^{\tau} \sum_{j=1}^{p} (\boldsymbol{z}_{ij}^{(a)} - \boldsymbol{z}_{ij}^{(a-1)})^2}$$

at the $a$th ADMM iteration. We update the penalty parameter $\alpha$ and the scaled dual variable $\boldsymbol{u}$ with the following schedule:

$$\alpha^{(a+1)} = 2\alpha^{(a)}, \quad \boldsymbol{u}^{(a+1)} = \frac{1}{2}\boldsymbol{u}^{(a)} \quad \text{if } r_{\text{primal}}^{(a)} > 10 \times r_{\text{dual}}^{(a)},$$

$$\alpha^{(a+1)} = \frac{1}{2}\alpha^{(a)}, \quad \boldsymbol{u}^{(a+1)} = 2\boldsymbol{u}^{(a)} \quad \text{if } r_{\text{dual}}^{(a)} > 10 \times r_{\text{primal}}^{(a)}.$$

Moreover, since STERGM is a probability distribution for the dynamic networks, in this work we stop ADMM learning until

$$\left| \frac{l(\boldsymbol{\theta}^{(a+1)}) - l(\boldsymbol{\theta}^{(a)})}{l(\boldsymbol{\theta}^{(a)})} \right| \leq \epsilon_{\text{tol}} \tag{56}$$

where $\epsilon_{\text{tol}}$ is a tolerance for the stopping criteria. By convention, we also implement two post-processing steps to finalize the detected change points $\{\hat{B}_k\}_{k=1}^K$. When the spacing between consecutive change points is less than a threshold or $\hat{B}_k - \hat{B}_{k-1} < \delta_{\text{spc}}$, we keep the detected change point with greater $\Delta\hat{\boldsymbol{\zeta}}$ value to avoid clusters of nearby change points. Furthermore, as the endpoints of a time span are usually not of interest, we discard the change point $\hat{B}_k$ that is less than a threshold $\delta_{\text{end}}$ and greater than $T - \delta_{\text{end}}$. In Section 5, we set $\delta_{\text{spc}} = 5$, and we set $\delta_{\text{end}} = 5$ and $\delta_{\text{end}} = 10$ for the simulated and real data experiments, respectively.

The algorithm to solve (6) via ADMM is presented in Algorithm 1. The complexity of an iteration for the Newton-Raphson method is $O(\tau^2 p^2)$ and that for the block coordinate descent method is $O(\tau(\tau-1)p)$. In general, the complexity of Algorithm 1 is at least of order $O(A[C\tau^2 p^2 + D\tau(\tau-1)p])$, where $A$, $C$, and $D$ are the respective numbers of iterations for ADMM, Newton-Raphson, and Group Lasso.

---

**Algorithm 1** Group Fused Lasso STERGM

1: **Input**: initialized parameters $\boldsymbol{\theta}^{(1)}, \boldsymbol{\gamma}^{(1)}, \boldsymbol{\beta}^{(1)}, \boldsymbol{u}^{(1)}$, tuning parameter $\lambda$, penalty parameter $\alpha$, number of iterations for ADMM, Newton-Raphson, and Group Lasso $A, C, D$, vectorized network data $\vec{\boldsymbol{y}}$, network change statistics $\boldsymbol{H}$

2: **for** $a = 1, \cdots, A$ **do**

3:     $\vec{\boldsymbol{\theta}} = \text{vec}_{\tau p}(\boldsymbol{\theta}^{(a)})$, $\vec{\boldsymbol{z}}^{(a)} = \text{vec}_{\tau p}(\mathbf{1}_{\tau,1}\boldsymbol{\gamma}^{(a)} + X\boldsymbol{\beta}^{(a)})$, $\vec{\boldsymbol{u}}^{(a)} = \text{vec}_{\tau p}(\boldsymbol{u}^{(a)})$

4:     **for** $c = 1, \cdots, C$ **do**

5:         Let $\vec{\boldsymbol{\theta}}_{c+1}$ be updated according to (13)

6:     **end for**

7:     $\boldsymbol{\theta}^{(a+1)} = \text{vec}_{\tau,p}^{-1}(\vec{\boldsymbol{\theta}}_{c+1})$

8:     Set $\tilde{\boldsymbol{\gamma}} = \boldsymbol{\gamma}^{(a)}$ and $\tilde{\boldsymbol{\beta}} = \boldsymbol{\beta}^{(a)}$

9:     **for** $d = 1, \cdots, D$ **do**

10:        Let $\tilde{\boldsymbol{\beta}}_{i,\cdot}^{d+1}$ be updated according to (14) for $i = 1, \cdots, \tau - 1$

11:        $\tilde{\boldsymbol{\gamma}}^{d+1} = (1/\tau)\mathbf{1}_{1,\tau} \cdot (\boldsymbol{\theta}^{(a+1)} + \boldsymbol{u}^{(a)} - \boldsymbol{X}\tilde{\boldsymbol{\beta}}^{d+1})$

12:     **end for**

13:     $\boldsymbol{\gamma}^{(a+1)} = \tilde{\boldsymbol{\gamma}}^{d+1}$, $\boldsymbol{\beta}^{(a+1)} = \tilde{\boldsymbol{\beta}}^{d+1}$

14:     $\boldsymbol{z}^{(a+1)} = \mathbf{1}_{\tau,1}\boldsymbol{\gamma}^{(a+1)} + \boldsymbol{X}\boldsymbol{\beta}^{(a+1)}$

15:     $\boldsymbol{u}^{(a+1)} = \boldsymbol{\theta}^{(a+1)} - \boldsymbol{z}^{(a+1)} + \boldsymbol{u}^{(a)}$

16: **end for**

17: $\hat{\boldsymbol{\theta}} \leftarrow \boldsymbol{\theta}^{(a+1)}$

18: **Output**: learned parameters $\hat{\boldsymbol{\theta}}$

---

To understand the role of two specific components in our framework, we conduct an ablation study on (1) the adaptive update of the penalty parameter $\alpha \in \mathbb{R}_+$ for the augmentation term, and (2) the use of position dependent weights $\boldsymbol{d} \in \mathbb{R}_+^{\tau-1}$ in the Group Fused Lasso penalty. Under the same settings as in Section 5.1, Table 13 reports the performance across different scenarios. While enabling either adaptive $\alpha$ or weighted $\boldsymbol{d}$ alone yields good performance, enabling both simultaneously produces similar results. Given that the differences in performance across configurations are modest, it is recommended to enable both components for the proposed method. This choice provides a robust and automated approach that leverages the benefits of adaptive optimization and time-aware regularization, without requiring further determination.

Table 13: Means (standard deviations) of evaluation metrics for dynamic networks across different scenario with $n = 100$, varying the use of position dependent weights and adaptive penalty updates.

| Scenario | Weighted $\boldsymbol{d}$ | Adaptive $\alpha$ | $\lvert\hat{K} - K\rvert \downarrow$ | $d(\hat{\mathcal{C}}\lvert\mathcal{C}) \downarrow$ | $d(\mathcal{C}\lvert\hat{\mathcal{C}}) \downarrow$ | $C(\mathcal{G}, \mathcal{G}') \uparrow$ |
|---|---|---|---|---|---|---|
| SBM $(\rho = 0.0)$ | ✗ | ✓ | 0.7 (0.6) | 0.9 (0.3) | 7.7 (6.3) | 91.45% |
| | ✓ | ✗ | 0.7 (1.3) | 0.8 (0.4) | 5.0 (6.4) | 91.34% |
| | ✓ | ✓ | 0.7 (1.3) | 0.8 (0.4) | 5.0 (6.4) | 91.34% |
| STERGM $(p = 6)$ | ✗ | ✓ | 0.0 (0.0) | 1.3 (0.6) | 1.3 (0.6) | 93.71% |
| | ✓ | ✗ | 0.0 (0.0) | 1.1 (0.3) | 1.1 (0.3) | 93.95% |
| | ✓ | ✓ | 0.0 (0.0) | 1.1 (0.4) | 1.1 (0.4) | 93.95% |
| RDPGM $(d = 10)$ | ✗ | ✓ | 0.2 (0.4) | 0.9 (0.3) | 3.9 (6.3) | 94.78% |
| | ✓ | ✗ | 0.2 (0.4) | 0.9 (0.3) | 4.1 (6.8) | 95.37% |
| | ✓ | ✓ | 0.4 (0.5) | 0.9 (0.0) | 6.5 (7.5) | 93.34% |

## F  Network Statistics in Experiments

In this section, we provide the formulations of the network statistics used in the simulation and real data experiments. The network statistics of interest are chosen from the extensive list in `ergm` (Handcock et al., 2022), an R library for network analysis. Tables 14 displays the formulations of network statistics used in the respective formation and dissolution models of our method for $t = 2, \ldots, T$ and the formulations of network statistics used in the competitor methods for $t = 1, \ldots, T$. The formulations are referred to directed networks, and those for undirected networks are similar.

Table 14: Network statistics and formulations.

| | Network Statistics | Formulation |
|---|---|---|
| $\boldsymbol{g}^{+}(\boldsymbol{y}^{+,t})$ | Edge Count | $\sum_{ij} \boldsymbol{y}_{ij}^{+,t}$ |
| | Mutuality | $\sum_{i<j} \boldsymbol{y}_{ij}^{+,t} \boldsymbol{y}_{ji}^{+,t}$ |
| | Triangles | $\sum_{ijk} \boldsymbol{y}_{ij}^{+,t} \boldsymbol{y}_{jk}^{+,t} \boldsymbol{y}_{ik}^{+,t} + \sum_{ij<k} \boldsymbol{y}_{ij}^{+,t} \boldsymbol{y}_{jk}^{+,t} \boldsymbol{y}_{ki}^{+,t}$ |
| | Homophily | $\sum_{ij} \boldsymbol{y}_{ij}^{+,t} \times \mathbb{1}(\boldsymbol{x}_i = \boldsymbol{x}_j)$ |
| | Isolates | $\sum_i \mathbb{1}\big(\deg_{\text{in}}(\boldsymbol{y}^{+,t}, i) = 0 \wedge \deg_{\text{out}}(\boldsymbol{y}^{+,t}, i) = 0\big)$ |
| $\boldsymbol{g}^{-}(\boldsymbol{y}^{-,t})$ | Edge Count | $\sum_{ij} \boldsymbol{y}_{ij}^{-,t}$ |
| | Mutuality | $\sum_{i<j} \boldsymbol{y}_{ij}^{-,t} \boldsymbol{y}_{ji}^{-,t}$ |
| | Triangles | $\sum_{ijk} \boldsymbol{y}_{ij}^{-,t} \boldsymbol{y}_{jk}^{-,t} \boldsymbol{y}_{ik}^{-,t} + \sum_{ij<k} \boldsymbol{y}_{ij}^{-,t} \boldsymbol{y}_{jk}^{-,t} \boldsymbol{y}_{ki}^{-,t}$ |
| | Homophily | $\sum_{ij} \boldsymbol{y}_{ij}^{-,t} \times \mathbb{1}(\boldsymbol{x}_i = \boldsymbol{x}_j)$ |
| | Isolates | $\sum_i \mathbb{1}\big(\deg_{\text{in}}(\boldsymbol{y}^{-,t}, i) = 0 \wedge \deg_{\text{out}}(\boldsymbol{y}^{-,t}, i) = 0\big)$ |
| $\boldsymbol{g}(\boldsymbol{y}^{t})$ | Edge Count | $\sum_{ij} \boldsymbol{y}_{ij}^{t}$ |
| | Mutuality | $\sum_{i<j} \boldsymbol{y}_{ij}^{t} \boldsymbol{y}_{ji}^{t}$ |
| | Triangles | $\sum_{ijk} \boldsymbol{y}_{ij}^{t} \boldsymbol{y}_{jk}^{t} \boldsymbol{y}_{ik}^{t} + \sum_{ij<k} \boldsymbol{y}_{ij}^{t} \boldsymbol{y}_{jk}^{t} \boldsymbol{y}_{ki}^{t}$ |
| | Homophily | $\sum_{ij} \boldsymbol{y}_{ij}^{t} \times \mathbb{1}(\boldsymbol{x}_i = \boldsymbol{x}_j)$ |
| | Isolates | $\sum_i \mathbb{1}\big(\deg_{\text{in}}(\boldsymbol{y}^{t}, i) = 0 \wedge \deg_{\text{out}}(\boldsymbol{y}^{t}, i) = 0\big)$ |

