# OpenReview forum: "Change Point Detection on A Separable Model for Dynamic Networks"
_TMLR — Accepted by TMLR_

### Review · Reviewer_CGpB · 2025-04-03

**Summary Of Contributions:**

This paper describes a methodology for detecting changes in the parameters generating data for dynamic networks. The method is based on solving an optimization problem, constructed from a Separable Temporal Exponential-family Random Graph Model and a group norm regularization. A specialized ADMM method for the problem is formulated and analyzed. The method is tested on simulated and real-world data, and compared with existing approaches to network data change detection. On simulation data, at least, it appears to give improved performance over competitors, likely due to the ability to model richer temporal evolution patterns and handle more complex random behaviors.

**Audience:**

Yes

**Claims And Evidence:**

Yes

**Requested Changes:**

# Critical for Securing Recommendation
Do a bit of work to improve the explanation of the performance on real data. Plots, as suggested above, would really help. It would be really helpful to understand what was happening with all of the methods.

# Helpful to Strengthen the Work
Improve the wording in Propositions 2 and 3.

**Strengths And Weaknesses:**

# Strengths:
- The problem itself is interesting
- The simulation experiments are fairly convincing about the merits compared to competitors.
- The writing is fairly clear:
    * It is well-structured
    * The methodology is explained well. Both the experimental setups and the algorithm itself were fairly easy to follow.
- The analysis of the ADMM method is nice to have.

# Weaknesses
- The applications to real data are difficult to interpret.
    * The authors do a reasonable job discussing events that coincide with the detected changes. However, the algorithms are detecting changes that don't always line up with the explanations.
    * Ideally, however, there would be some way to display the data in a way that would make it clearer what was causing the changes to occur:
        - Maybe display some snapshots of the adjacency matrices near the change points
        - Alternatively, something like a raster plot might help (where each pair is its own row)
        - Additionally, since the matrices for the stock data were computed after some preprocessing, it might be useful to alternatively plot the raw data and line it up with the detected change points.
- Propositions 2 and 3 are stated awkwardly.
    * For Proposition 2, it would probably be better to say something like "A step of Newton-Raphson can be implemented by..."
    * For Proposition 3, it would probably be better to say something like "Block coordinate descent can be implemented by..."

---

> ### Author Response · Authors · 2025-07-10
> **Responses to Review Comments**
>
> We are very grateful for the constructive comments, especially for pointing out the interpretation on real data experiments. The revised texts are colored in blue in the updated manuscript. Additionally, we provide a finite-sample error bound in Section 4.2.
>
> \textbf{Q1:} Besides aligning the detected change points with real world events, we have included the raster plots obtained from the networks (adjacency matrices), which indeed provide a better visualization in interpreting the structural changes as recommended by the reviewer. Specifically, the raster plot for the MIT cellphone data is provided in Figure 6, and the raster plot for the stock market data is provided in Figure 8. Moreover, we use red lines to indicate the detected change points, and we use blue regions to highlight the substantial structural changes.
>
>
> For the MIT cellphone data, we observe notable changes in the volume of interaction before and after the detected change points from the raster plot. In Lines 423-430 on Page 21, we write:
>
> ``Visually, the substantial reduction of interactions in Region I and the subtle shifts in Region II of Figure 6 support the change points detected by the proposed method, aligning with the largest spike for the winter vacation (2004-12-17) and the smaller spike for the spring vacation (2005-03-24) in Figure 5. Furthermore, we detect a few spikes in the middle of October 2004, which correspond to the annual sponsor meeting that happened on 2004-10-21. About two-thirds of the participants have prepared and attended the annual sponsor meeting, and the majority of their time has contributed to achieve project goals throughout the week. The drastic shifts in Region III of Figure 6 also reveal the frequent changes in the network patterns during the sponsor meeting week."
>
> For the stock market data, we can also see clear change in the volume of interaction before and after the detected change points from the raster plot. In Lines 463-467 on Page 23, we write:
>
> ``In Figure 8, Region I shows a substantial increase in edge density following the collapse of New Century Financial Corporation, indicating heightened volatility in the stock market. As fewer edges suggest stable trend, Region II captures the market downturn following the bankruptcy of Lehman Brothers. Finally, Region III shows a rebound in market activity, as edge density rises after the Dow Jones Industrial Average reaches its lowest point, suggesting the beginning of a recovery phase."
>
>
> \textbf{Q2:} Thanks for the comments! Since we have added the error bound as a Proposition, we no longer called the Newton-Raphson and Group Lasso updates as Propositions. Also, we have modified the wording for the two updates. Specifically, in Lines 206-208 on Pages 7-8, we write:
>
> ``The Newton-Raphson method to iteratively update the parameter $\theta \in \mathbb{R}^{\tau p \times 1}$ for the proposed framework is implemented as follows:
> $$\theta_{c+1} = \theta_{c} - \big( H^\top W H + \alpha I_{\tau p}\big)^{-1} \cdot \big( -H^\top (y - \mu) + \alpha  (\theta_{c} - z^{(a)} + u^{(a)})\big)$$
> where $c$ denotes the current Newton-Raphson iteration."
>
>
> In Lines 222-224 on Page 8, we write:
>
> ``Specifically, the block coordinate descent method to update $\beta_{i,\cdot}$ for each block $i = 1, \dots, \tau-1$ is implemented as follows:
>
> $$\beta_{i,\cdot} = \frac{1}{\alpha X_{\cdot,i}^{\top} X_{\cdot,i}} \Big( 1 - \frac{\lambda}{||s_{i} ||_2} \Big)\ _{+} s_i $$
>
> where $(\cdot)_+ = \max(\cdot,0)$ and
>
> $$s_{i} = \alpha X_{\cdot,i}^\top (\theta^{(a+1)} + u^{(a)} - 1_{\tau,1} \gamma - X_{\cdot,-i} \beta_{-i,\cdot} ).$$"

---

### Review · Reviewer_pZTb · 2025-04-30

**Summary Of Contributions:**

In this paper, the authors propose a method for change point detection of graphs, specifically when the graphs stem from a Separable Temporal Exponential-family Random Graph Model. The method is based on the estimation of the parameters for each graph, exploiting the fused lasso to impose continuity in the parameters for graphs belonging to the same model. The change points are then determined with a simple threshold in the norm of the difference for continuous parameters. The method is compared with two other R packages, both on simulated and real data. The code is provided along the paper.

**Audience:**

Yes

**Broader Impact Concerns:**

No concerns

**Claims And Evidence:**

Yes

**Requested Changes:**

- I would've liked a better explanation of the exponential model and its true and surrogate likelihoods.

 - The experimental section needs a lot of work to gain the protagonism that I expect in this kind of papers. In particular, the code of several graph CPD methods cited in the paper is available, and therefore, they should be included in the comparison. I would also like to see how this method performs with graphs stemming from other dynamic network models.

 - The description of the methods (in particular, the methods with which the authors compare) is quite poor.

 - There's no time comparison or mention of the complexity of any of the methods.

**Strengths And Weaknesses:**

Although the reading flow is not particularly a strength, the paper is well-written in general.
The model formulation and the optimization seem to be correct.

To me, the main weaknesses are related to the focus of the paper, in the following sense.
As far as I understand, the model (all the Section 2) is not new. Or is there a contribution in this section that I'm missing?
On the other hand, the use of fused lasso for change point detection is also not new. Moreover, the optimization (formulation and resolution through ADMM with its subproblems) is essentially the same as in the paper by Bleakley and Vert.
Therefore, the contribution of the paper is to use these two tools toghether. Which is a valid contribution to me.
However, if this is the case (I might be wrong), I would've expected a more profound experimental section.
The quality of the experimental section is ok for a completely new method with new ideas, where one wants to show that it makes sense in specific scenarios.
In this case, I expect the experimental section to be the core of the paper.

---

> ### Author Response · Authors · 2025-07-10
> **Responses to Review Comments**
>
> We are extremely grateful for the constructive comments, especially for pointing out the focus of the paper. The revised texts are colored in blue in the updated manuscript. For the simulation study, we have added one more scenario and two more competitor methods in Section 5.1. Moreover, we provide a finite-sample error bound in Section 4.2.
>
>
>
>
> \textbf{Q1:} We have enhanced the introduction of ERGM and STERGM in Section 2.1 and 2.2, respectively. Moreover, for the comparison of the exponential model between its true and surrogate likelihoods, in Lines 137-155 on Pages 4-5, we write:
>
> ``Empirically, for change point detection, the pseudo-likelihood of STERGM is preferable to the true likelihood for the following three reasons. First, the dyadic dependency in $y^{+,t}$ and $y^{-,t}$ is mitigated by conditioning on the previous network $y^{t-1}$ and by separately modeling through the formation and dissolution processes. Specifically, conditional on $y_{ij}^{t-1} = 1$, the $y_{ij}^{+,t} = \max(y_{ij}^{t-1},y_{ij}^{t})$ can only be $1$; while conditional on $y_{ij}^{t-1} = 0$, the $y_{ij}^{-,t} = \min(y_{ij}^{t-1},y_{ij}^{t})$ can only be $0$. This design explicitly restricts the states of dyads by partitioning network evolution into formation and dissolution processes, thereby reducing the dyadic dependence within $y^{+,t}$ and $y^{-,t}$. Hence, conditioning on the previous network which already captures the structural dependencies, the pseudo-likelihood of STERGM becomes a reasonable surrogate. Second, the primary objective in change point detection is to localize substantial structural changes over time, rather than to recover the coefficient estimates for network effect interpretation. In the former case, the pseudo-likelihood of STERGM remains adequate, as large parameter shifts can still be reliably identified, even if the estimates are subject to mild bias by using a common approximation to the true likelihood. Third, we adopt the logarithm of the pseudo-likelihood to particularly avoid using MCMC sampling or Bayesian inference, which is computationally challenging for the optimization problem defined in Section 3. Instead, the pseudo-likelihood of STERGM improves the scalability of the estimation procedure, and the sigmoid function that involves the pre-computed change statistics permits efficient calculation of the gradients and Hessians for iterative parameter updates. In summary, the pseudo-likelihood of STERGM provides both computational feasibility and effectiveness to facilitate change point detection in dynamic networks, an advantage that the true likelihood cannot offer at scale."
>
> \textbf{Q2:} We have added one more scenario in Section 5.1 to have a total of three scenarios for our simulation study. Specifically, in Scenario 3 on Page 15, we use the Random Dot Product Graph Model [1,2] to generate dynamic graphs, and our proposed method outperforms the competitor methods. The corresponding results are displayed in Tables 7, 8, 9 on Pages 19-20. Furthermore, we have added two more competitor methods, namely CPDrdgp [3] and CPDnbs [4], for all three scenarios in the simulation study and real data experiments. Our proposed method outperforms the competitors in the simulation study, and the corresponding results are displayed in Tables 1-9 in Section 5.1.

---

> ### Author Response · Authors · 2025-07-10
> **Response to Review Comments**
>
> \textbf{Q3:} We have modified the description and provided more details about the competitor methods. Specifically, in Lines 304-326 on Pages 11-12, we write:
>
> ``We compare our proposed method against four competing methods: gSeg [5], kerSeg [6], CPDrdpg [3], and CPDnbs [4]. The gSeg method utilizes a graph-based scan statistic and the kerSeg method employs a kernel-based scan statistic to assess distributional differences between two segments before and after a candidate change point. The CPDrdpg method identifies change points by estimating the latent positions of nodes using a Random Dot Product Graph Model (RDPGM) and by constructing a nonparametric CUSUM statistic that accounts for temporal dependence. The CPDnbs method integrates sample splitting with wild binary segmentation (WBS) and detects change points by maximizing the inner product between two CUSUM statistics derived from the split samples. In particular, the gSeg and kerSeg methods are available in the respective R libraries \texttt{gSeg} and \texttt{kerSeg}. The CPDrdpg and CPDnbs methods are available in the R library \texttt{changepoints}."
>
> ``For gSeg, we use the minimum spanning tree to construct the similarity graph, and we use the original edge-count scan statistic to test the null hypothesis that there is no change point within a time span. The significance level is set to $\alpha=0.05$. For kerSeg, we use the kernel-based scan statistic fGKCP$_1$ and we set the significance level $\alpha=0.001$. Since gSeg and kerSeg are general methods for change point detection, we use networks (nets.) and network statistics (stats.) as two types of input data for comparison. For CPDrdpg, we let the number of intervals for wild binary segmentation (WBS) be $W = 50$, and we let the number of leading singular values of an adjacency matrix in the scaled PCA algorithm be $d = 5$ to fit a RDPGM. For CPDnbs, we let the number of intervals for WBS be $W=15$ and we set the threshold for detection to the order of $n\log^2(T)$ as suggested by [2]. Throughout, we use these chosen settings, since they produce higher coverage metrics $C(\mathcal{G},\mathcal{G}')$ for the competitors across different scenarios on average. Changing the above settings can improve their performance on some specifications, while severely jeopardizing their performance on other specifications."
>
>
> \textbf{Q4:} The time comparison of the proposed and competitor methods are provided in Table 12 in Appendix E on Page 39. We have also discussed the complexity of the proposed ADMM algorithm. Specifically, in Lines 750-759 on Page 39, we write:
>
> ``The complexity of an iteration for the Newton-Raphson method is $O(\tau^2 p^2)$ and that for the block coordinate descent method is $O(\tau (\tau-1) p )$. In general, the complexity of Algorithm 1 is at least of order $O(A[C\tau^2 p^2 + D \tau (\tau-1) p ])$, where $A$, $C$, and $D$ are the respective numbers of iterations for ADMM, Newton-Raphson, and Group Lasso. Table 12 compares the computation times of different methods across varying node sizes. We focus on three representative configurations where the change point detection performance, as shown in Section 5.1, is comparable across methods. The reported computation times in seconds reflect the total runtime over three simulated network time series. It is worth noting that some competing methods exhibit shorter runtime when they fail to detect the correct change points and terminate early. Though the proposed method requires more computation time, it achieves better performance on average compared to the competitors."
>
>
> [1] Stephen Young and Edward Scheinerman. Random dot product graph models for social networks. In International Workshop on Algorithms and Models for the Web-Graph, 2007
>
> [2] Avanti Athreya, Donniell Fishkind, Minh Tang, Carey Priebe, Youngser Park, Joshua Vogelstein, Keith Levin, Vince Lyzinski, Yichen Qin, and Daniel Sussman. Statistical inference on random dot product graphs: a survey. Journal of Machine Learning Research, 2018.
>
>
> [3] Oscar Hernan Madrid Padilla, Yi Yu, and Carey E Priebe. Change point localization in dependent dynamic nonparametric random dot product graphs. Journal of Machine Learning Research, 2022.
>
> [4] Daren Wang, Yi Yu, and Alessandro Rinaldo. Optimal change point detection and localization in sparse dynamic networks. The Annals of Statistics, 2021.
>
> [5] Hao Chen and Nancy Zhang. Graph-based change-point detection. The Annals of Statistics, 2015.
>
> [6] Hoseung Song and Hao Chen. Practical and powerful kernel-based change-point detection. IEEE Transactions on Signal Processing, 2024.

---

> > ### Comment · Reviewer_pZTb · 2025-07-28
> >
> > Thank you for the revision and for addressing my questions and concerns. I find the changes satisfactory. However, I would suggest moving the time comparison from the appendix to the main text.

---

> > > ### Author Response · Authors · 2025-07-29
> > > **Response to Review Comments**
> > >
> > > We greatly appreciate the positive feedback. We have moved the time comparison from the appendix to the main text in Section 5.2.

---

### Review · Reviewer_XU3V · 2025-07-01

**Summary Of Contributions:**

This paper presents a novel framework for unsupervised change point detection in time series of dynamic networks by leveraging a time-heterogeneous Separable Temporal Exponential-family Random Graph Model (STERGM). To effectively capture shifts in the underlying data-generating processes, the proposed method fits a sequence of STERGM models while enforcing smoothness via a Group Fused Lasso (GFL) penalty on the parameter differences across time. The resulting optimization problem is efficiently solved using the Alternating Direction Method of Multipliers (ADMM), and parameter estimation is expedited through the use of the STERGM pseudo-likelihood. This approach addresses the challenges posed by temporal and dyadic dependencies in evolving networks and allows for the detection of multiple structural changes.

**Audience:**

Yes

**Claims And Evidence:**

Yes

**Requested Changes:**

Check weaknesses. Especially new baselines needed and details on fair parameter tuning.

**Strengths And Weaknesses:**

Strengths

- Change point detection needs substantially more attention as a critical task for analytics
- Solid and sound solution for the particular problem
- Results support the overall claims and shows the strengths of the solution

Weaknesses

- Tuning of parameters for the proposed method and baselines requires more details. It's unclear how fair is the comparison currently when multiple parameters require tuning

- Only two baselines are used. Several of the datasets used are time-series based and hence baselines for that community could serve as potential baselines. For example [a] and relevant works cited there.

Ermshaus, Arik, Patrick Schäfer, and Ulf Leser. "ClaSP: parameter-free time series segmentation." Data Mining and Knowledge Discovery 37.3 (2023): 1262-1300.

- Runtime performance is missing, the method seems computationally heavy

- An ablation study can help understand strengths of sub-components of the solution

---

> ### Author Response · Authors · 2025-07-10
> **Responses to Review Comments**
>
> We are extremely grateful for the constructive comments, especially for pointing out the discussion on tuning parameters and ablation study. The revised texts are colored in blue in the updated manuscript. For the simulation study, we have added one more scenario and two more competing methods [1,2] in Section 5.1. Moreover, we provide a finite-sample error bound in Section 4.2.
>
>
>
> \textbf{Q1:} There are two major tuning parameters for the proposed method, the $\lambda$ for the Group Fused Lasso regularization and the $\alpha$ for the augmentation term. For the tuning parameter $\lambda$, we use BIC to perform model selection, which is a common procedure in choosing its optimal value. The details are provided in Section 4.3. Moreover, for the augmentation term, we update the penalty parameter $\alpha$ along with the scaled dual variable, according to the schedule that is recommended by [3], which is also described in our Appendix E on Page 38. For the competitor methods, we follow the recommendation provided by the papers of the respective methods. In cases where tuning is necessary, we choose the value from a list of candidates, such that it produces higher coverage metrics $C(G, G')$ on average across different scenarios. In other words, we fine-tune the competitor methods and report better performance for fair evaluation. Specifically, in Lines 315-326 on Pages 11-12, we write,
>
> ``For gSeg, we use the minimum spanning tree to construct the similarity graph, and we use the original edge-count scan statistic to test the null hypothesis that there is no change point within a time span. The significance level is set to $\alpha=0.05$. For kerSeg, we use the kernel-based scan statistic fGKCP$_1$ and we set the significance level $\alpha=0.001$. Since gSeg and kerSeg are general methods for change point detection, we use networks (nets.) and network statistics (stats.) as two types of input data for comparison. For CPDrdpg, we let the number of intervals for wild binary segmentation (WBS) be $W = 50$, and we let the number of leading singular values of an adjacency matrix in the scaled PCA algorithm be $d = 5$ to fit a RDPGM. For CPDnbs, we let the number of intervals for WBS be $W=15$ and we set the threshold for detection to the order of $n\log^2(T)$ as suggested by [2]. Throughout, we use these chosen settings, since they produce higher coverage metrics $C(\mathcal{G},\mathcal{G}')$ for the competitors across different scenarios on average. Changing the above settings can improve their performance on some specifications, while severely jeopardizing their performance on other specifications."
>
>
> \textbf{Q2:} We have added one more scenario in Section 5.1 to have a total of three scenarios for our simulation study. Specifically, in Scenario 3 on Page 15, we use the Random Dot Product Graph Model [5,6] to generate dynamic graphs, and our proposed method outperforms the competitor methods. The corresponding results are displayed in Tables 7, 8, 9 on Pages 19-20. Furthermore, we have added two more competitor methods, namely CPDrdgp [1] and CPDnbs [2], for all three scenarios in our simulation study and real data experiments. Our proposed method outperforms the competitors in the simulation study, and the corresponding results are displayed in Tables 1-9 in Section 5.1. Furthermore, we have attempted the ClaSP [4] method to the simulated networks. However, since ClaSP is not designed for dynamic networks and ClaSP focuses on time series with periodic structure, it does not detect any change points in our piece-wise constant sequence with T=100 time points. Given that the simulated network time series violate the assumptions of ClaSP, it may not be fair to compare our proposed method with ClaSP. Yet, the contrastive learning framework proposed by ClaSP presents a promising extension to the proposed method, as we mention in the discussion section in Lines 488-491 on Page 24.

---

> ### Author Response · Authors · 2025-07-10
> **Responses to Review Comments**
>
> \textbf{Q3:} The time comparison of the proposed and competitor methods are provided in Table 12 in Appendix E on Page 39. We have also discussed the complexity of the proposed ADMM algorithm. Specifically, in Lines 745-754 on Page 39, we write:
>
> ``The complexity of an iteration for the Newton-Raphson method is $O(\tau^2 p^2)$ and that for the block coordinate descent method is $O(\tau (\tau-1) p )$. In general, the complexity of Algorithm 1 is at least of order $O(A[C\tau^2 p^2 + D \tau (\tau-1) p ])$, where $A$, $C$, and $D$ are the respective numbers of iterations for ADMM, Newton-Raphson, and Group Lasso. Table 12 compares the computation times of different methods across varying node sizes. We focus on three representative configurations where the change point detection performance, as shown in Section 5.1, is comparable across methods. The reported computation times in seconds reflect the total runtime over three simulated network time series. It is worth noting that some competing methods exhibit shorter runtime when they fail to detect the correct change points and terminate early. Though the proposed method requires more computation time, it achieves better performance on average compared to the competitors."
>
>
> \textbf{Q4:} Since we do not have the ground truth for each sub-routine and the ADMM implements the three updates recursively, it could be difficult to evaluate the performance of individual sub-routine in terms of change point detection. Instead, we conduct an ablation study on two components of the proposed framework: the adaptive update of the penalty parameter $\alpha$ for the augmentation term, and (2) the use of position dependent weights $\bf{d}$ in the Group Fused Lasso penalty. These two components are also related to the \textbf{Q1} raised by the reviewer regarding the tuning parameter, and we greatly appreciate the comment on conducting an ablation study, which helps us understand the proposed framework better. Specifically, in Lines 760-767 on Pages 39-40, we write:
>
> ``To understand the role of two specific components in our framework, we conduct an ablation study on (1) the adaptive update of the penalty parameter $\alpha$ for the augmentation term, and (2) the use of position dependent weights $\bf{d}$ in the Group Fused Lasso penalty. Under the same settings as in Section 5.1, Table 13 reports the performance across different scenarios. While enabling either adaptive $\alpha$ or weighted $\bf{d}$ alone yields good performance, enabling both simultaneously produces similar results. Given that the differences in performance across configurations are modest, it is recommended to enable both components for the proposed method. This choice provides a robust and automated approach that leverages the benefits of adaptive optimization and time-aware regularization, without requiring further determination."
>
>
>
>
>
> [1] Oscar Hernan Madrid Padilla, Yi Yu, and Carey E Priebe. Change point localization in dependent dynamic nonparametric random dot product graphs. Journal of Machine Learning Research, 2022.
>
> [2] Daren Wang, Yi Yu, and Alessandro Rinaldo. Optimal change point detection and localization in sparse dynamic networks. The Annals of Statistics, 2021.
>
> [3] Stephen Boyd, Neal Parikh, Eric Chu, Borja Peleato, Jonathan Eckstein. Distributed optimization and statistical learning via the alternating direction method of multipliers. Foundations and Trends in Machine Learning, 2011.
>
> [4] Ermshaus, Arik, Patrick Schäfer, and Ulf Leser. ClaSP: parameter-free time series segmentation. Data Mining and Knowledge Discovery, 2023
>
>
> [5] Stephen Young and Edward Scheinerman. Random dot product graph models for social networks. In International Workshop on Algorithms and Models for the Web-Graph, 2007
>
> [6] Avanti Athreya, Donniell Fishkind, Minh Tang, Carey Priebe, Youngser Park, Joshua Vogelstein, Keith Levin, Vince Lyzinski, Yichen Qin, and Daniel Sussman. Statistical inference on random dot product graphs: a survey. Journal of Machine Learning Research, 2018.

---

### Decision · Action_Editor_PaRM · 2025-08-15

**Recommendation:** Accept as is

**Audience:**

Yes

**Audience Explanation:**

This work would be of interest to those who work on change point detection and its applications across domains.

**Claims And Evidence:**

Yes

**Claims Explanation:**

The claims are generally well-supported. There are some concerns regarding the baselines and the datasets. For the final version, it would be great if the authors can add more baselines and release the datasets.

---

> ### Author Response · Authors · 2025-08-29
> **Response to Comments**
>
> We have added one more baseline method, namely CPDker[1], to both simulated and real data experiments. The results are displayed in Tables 1-10 in Section 5.1 and Figures 5 and 7 in Section 5.2. Additionally, the code and data source are provided in the 'Code and Data Availability' section on Page 25. We greatly appreciate the review comments.
>
> [1] Madrid Padilla, Oscar Hernan, Yi Yu, Daren Wang, and Alessandro Rinaldo. "Optimal nonparametric multivariate change point detection and localization." IEEE Transactions on Information Theory, 2021